# Divergent metabolic programmes control two populations of MAIT cells that protect the lung

Thomas Riffelmacher [1,2,5] ✉, Mallory Paynich Murray[1,5], Chantal Wientjens[1], Shilpi Chandra[1], Viankail Cedillo-Castelán[1], Ting-Fang Chou[1], Sara McArdle[1], Christopher Dillingham[1], Jordan Devereaux[3], Aaron Nilsen[3], Simon Brunel[1], David M. Lewinsohn[3], Jeff Hasty[4], Gregory Seumois[1], Christopher A. Benedict[1], Pandurangan Vijayanand[1] & Mitchell Kronenberg [1,4] ✉

Although mucosal-associated invariant T (MAIT) cells provide rapid, innate-like responses, they are not pre-set, and memory-like responses have been described for MAIT cells following infections. The importance of metabolism for controlling these responses, however, is unknown. Here, following pulmonary immunization with a *Salmonella* vaccine strain, mouse MAIT cells expanded as separate CD127−Klrg1+ and CD127+Klrg1− antigen-adapted populations that differed in terms of their transcriptome, function and localization in lung tissue. These populations remained altered from steady state for months as stable, separate MAIT cell lineages with enhanced effector programmes and divergent metabolism. CD127+ MAIT cells engaged in an energetic, mitochondrial metabolic programme, which was critical for their maintenance and IL-17A synthesis. This programme was supported by high fatty acid uptake and mitochondrial oxidation and relied on highly polarized mitochondria and autophagy. After vaccination, CD127+ MAIT cells protected mice against *Streptococcus pneumoniae* infection. In contrast, Klrg1+ MAIT cells had dormant but ready-to-respond mitochondria and depended instead on Hif1a-driven glycolysis to survive and produce IFN-γ. They responded antigen independently and participated in protection from influenza virus. These metabolic dependencies may enable tuning of memory-like MAIT cell responses for vaccination and immunotherapies.

Mucosal-associated invariant T (MAIT) cells are found in humans, mice and many other mammals[1,2]. They recognize MR1, a major histocompatibility complex-class I-like protein. MR1 presents to MAIT cells riboflavin-derived metabolites produced by microbes, that when combined with methylglyoxal, form 5-(2-oxopropylideneamino)-6-D-ribitylaminouracil (5-OP-RU)[3,4]. MAIT cells are abundant in humans in blood, lung, liver and other tissues, and probably are critical for many types of immune response. Because MR1 is highly conserved, allogeneic MAIT cells are unlikely to cause substantial graft-versus-host disease. Therefore, MAIT cells are an attractive lymphocyte type for the development of cell therapies.

[1]La Jolla Institute for Immunology, La Jolla, CA, USA. [2]Kennedy Institute of Rheumatology, University of Oxford, Oxford, UK. [3]Oregon Health and Science University, Portland, OR, USA. [4]Department of Molecular Biology, University of California San Diego, La Jolla, CA, USA. [5]These authors contributed equally: Thomas Riffelmacher, Mallory Paynich Murray. ✉e-mail: triffelmacher@lji.org; mitch@lji.org

Several reports have described memory-like responses of natural killer (NK) cells and innate lymphoid cells, which has been referred to as trained immunity[5–9]. Similar long-term changes have been observed for innate-like T cells, including γδ T cells and invariant natural killer T (iNKT) cells[10–12]. MAIT cells exhibit immediate effector functions, and they also express markers of antigen experience, even as they differentiate in the thymus[1,13–15]. They therefore might be considered to be natural memory cells. MAIT cells can undergo further antigen-dependent differentiation with long-term increases in their population size and activation state. This occurs following infection with some of the microbes that express antigens for MAIT cells, including pathogens *Legionella pneumophila* and *Francisella tularensis*, as well as BRD509, an attenuated, vaccine strain of *Salmonella enterica* serovar Typhimurium[4,16–18]. After these infections, the memory-like MAIT cells provide increased protective capacity.

MAIT cell heterogeneity at steady state has been demonstrated by several reports[1,15,19,20], with the predominant mouse MAIT cell subset characterized by the expression of RORγt and the ability to secrete IL-17, named MAIT17 cells on the basis of similarity to IL-17-producing CD4+ T cells. A T-bet-expressing population that secretes IFN-γ, called MAIT1 cells, has also been characterized[13,20,21]. These functions are preserved in memory-like MAIT cells following vaccination or infection, but a detailed characterization of memory-like MAIT cells and the factors that lead to their formation is needed[16–18,22,23]. Human MAIT cell functional subsets have not been defined as clearly, but human MAIT cells capable of secreting IFN-γ and/or IL-17 have been observed[1,19,20,24]. Because IFN-γ and IL-17 stimulate very different types of immune response, therapeutic approaches based on activating MAIT cells will need to consider factors that impact the balance of these types of MAIT cell response.

One key factor that influences CD4 and CD8 T cell memory is the distinct metabolic programmes that regulate naïve and effector versus memory states[25–27]. The metabolism of memory-like MAIT cells has not been investigated, and furthermore, the role of metabolism in forming the functional subsets of these cells has not been determined. In this article, we characterized two states of antigen-adapted or memory-like MAIT cells and showed they are both highly different from steady-state MAIT cells for their gene expression programmes. We defined the phenotypes of these antigen-adapted MAIT (*aa*MAIT) cell subsets, and their distribution, protective capacities and differing metabolic states. We analysed their stability and the role of metabolism in regulating their immune responsiveness and prevalence.

## Results

### Two *aa*MAIT cell subsets

We used the rapidly cleared vaccine strain of *S.* Typhimurium, BRD509, to challenge mice with the goal of assessing MAIT cell responses to subsequent bacterial or viral pulmonary infections. We concentrated on MAIT cells in the lung because of their critical role in controlling respiratory pathogens[17,18,22,23,28–30]. In agreement with previous results[16], retropharyngeal vaccination with *S.* Typhimurium led to a >50-fold increase in pulmonary MAIT cells, gated as in Extended Data Fig. 1a, which occurred within days and was maintained beyond 8 weeks (Fig. 1a, Extended Data Fig. 1a). This increase was accompanied by transient dendritic cell (DC) activation, gated as in Extended Data Fig. 1b, characterized by increased MR-1, CD86 and CD80 expression by CD11b+ CD103– conventional type-2 DC (cDC2) (Fig. 1b). The increase in MR1 was selective in that cDC1 were not affected (Extended Data Fig. 1b). An invasion mutant strain of *S.* Typhimurium (Δ*InvA*Δ*SpiB*) did not increase expression of MR1 (Fig. 1b) or increase MAIT cells (Extended Data Fig. 1c,d), indicating that cell invasion was required[31]. Complementation of a mutant *S.* Typhimurium strain that lacks MAIT cell antigen (Δ*RibD*) with the synthetic MAIT cell antigen 5-OP-RU confirmed antigen dependence (Extended Data Fig. 1e,f). Importantly, the lung MAIT cell increase did not require persistence of bacteria beyond 1 week, as demonstrated with a *Salmonella* synchronized lysis mutant (Extended Data Fig. 1g,h)[32].

As the existence of memory in a cell type capable of immediate effector function might be controversial, we refer to MAIT cell populations altered long term by bacterial exposure as *aa*MAIT cells. We determined the extent to which *aa*MAIT cells in the lung differed from steady state. Single-cell RNA sequencing (scRNA-seq) analysis indicated that, 40 days after infection, 92% of *aa*MAIT cells clustered in a separate uniform manifold approximation and projection (UMAP) space compared with steady-state MAIT cells (Fig. 1c). A large fraction of MAIT cells from vaccinated mice had a Th1 signature, others, including the steady-state MAIT cells, had a Th17-like transcriptome. Some, for example cluster 3, were intermediate (Fig. 1d). The Th1-like *aa*MAIT cells had increased *Klrg1* and *Gzmb* transcripts, although *Tbx21* and *Ifng* were more broadly expressed. Klrg1+ cells were absent at steady state, peaked at approximately day 28 and then declined but were maintained long term (Fig. 1e and Extended Data Fig. 2a). At the protein level, they expressed T-bet, but also some Rorγt (Fig. 1f,g), in line with an expanded Th1-like population previously described after infections[16,18,23]. The Klrg1+ *aa*MAIT cells had increased CD8α expression, but decreased CD103, Sdc1 and Icos, similar to steady-state MAIT1 cells (Extended Data Fig. 2b)[15]. However, compared with previously published steady-state MAIT1 cells[15,19], Klrg1+ *aa*MAIT cells had increased transcripts for NK receptors, *Ifng*, *Il12rb2* and molecules involved in cytotoxicity and its regulation, such as *Fasl*, *Gzma* and *Serpinb9* (Extended Data Fig. 2c,d). To explore the timing of *aa*MAIT cell development, we carried out scRNA-seq analysis of MAIT cells at day 6 after vaccination. Pathway analysis at day 6 did not indicate an activated MAIT1 cell cluster, consistent with the absence of substantial Klrg1, but showed cells with an effector phenotype (Extended Data Fig. 3a–c), and a capacity for cytokine production, especially IL-17A (Extended Data Fig. 3d).

**Fig. 1 | A *Salmonella* vaccine strain induces two *aa*MAIT cell subsets.** MAIT cells from lungs of C57BL/6J mice were analysed at the indicated times post BRD509 vaccination by flow cytometry. **a**, Representative plots (left) and quantification (right) of MR1:5-OP-RU Tetramer+ TCR-β+ MAIT cells. *n* = 36, 9, 9, 13, 7, 11, 9 and 16 mice per group, combined from 12 independent experiments. Statistical significance assessed via one-way analysis of variance (ANOVA), ****P < 0.0001. **b**, Representative histograms (top) of expression of MR1 by lung cDC2 after infection at the indicated times (left) or with the indicated *Salmonella* strains (right). Quantification (bottom) of MR1, CD86 and CD80 expression by cDC2. *n* = 13, 8, 4, 7, 7, 8, 5, 6, 9 and 8 mice per group, combined from two independent experiments, statistical significance assessed via one-way ANOVA with Tukey's multiple comparison test, ****P < 0.0001 for MR1 on DC or unpaired *t*-test, ***P = 0.0007 (CD86), ***P = 0.0008 (CD80). **c**, UMAP representations of scRNA-seq data from pulmonary MAIT cells at day 0 and day 40 post BRD509 vaccination. MAIT cells sorted from lungs pooled from ten mice per timepoint in one experiment. **d**, Th1 and Th17 gene signature plots (left) and feature plots for the indicated genes. **e**, Representative plots (left) and quantification (right) of expression of Klrg1 by lung MAIT cells at indicated times post BRD509 vaccination. *n* = 28, 6, 8, 4, 7, 11, 9 and 16 mice per group, respectively, combined from nine independent experiments, statistical significance via one-way ANOVA, ****P < 0.0001. **f,g**, Representative plots (**f**) and quantification (**g**) of RORγt and T-bet by lung MAIT cells at indicated times post BRD509 vaccination. RORγt: *n* = 11 (untreated), *n* = 6 (day 6), *n* = 13 (day 60), combined from three independent experiments; RORγt subsets: *n* = 20 (untreated), *n* = 25 (day 60), combined from five independent experiments. T-bet: *n* = 11 (untreated), *n* = 6 (day 6), *n* = 13 (day 60), combined from three independent experiments, statistical significance via one-way ANOVA with Tukey's multiple comparisons test; left (*P = 0.0267, **P = 0.0034), left middle (**P = 0.0045), middle (****P < 0.0001, **P = 0.0013), right (*P = 0.0449, ****P < 0.0001). **h**, Representative cytokine production by intracellular flow cytometry by MAIT cell populations at indicated timepoints in response to PMA and ionomycin. All data displayed as mean ± s.d. Source numerical data are available in source data. n.s., not significant.

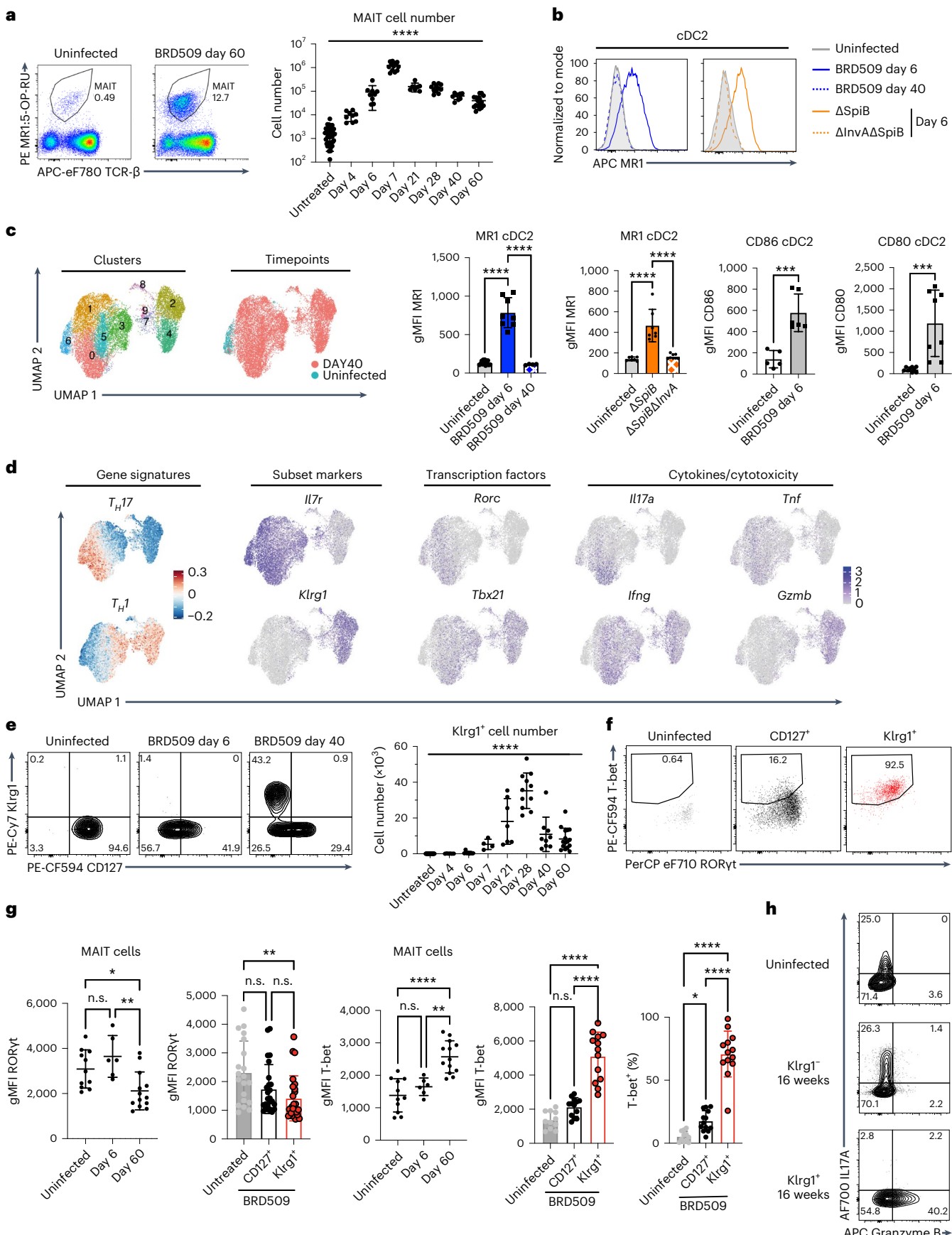

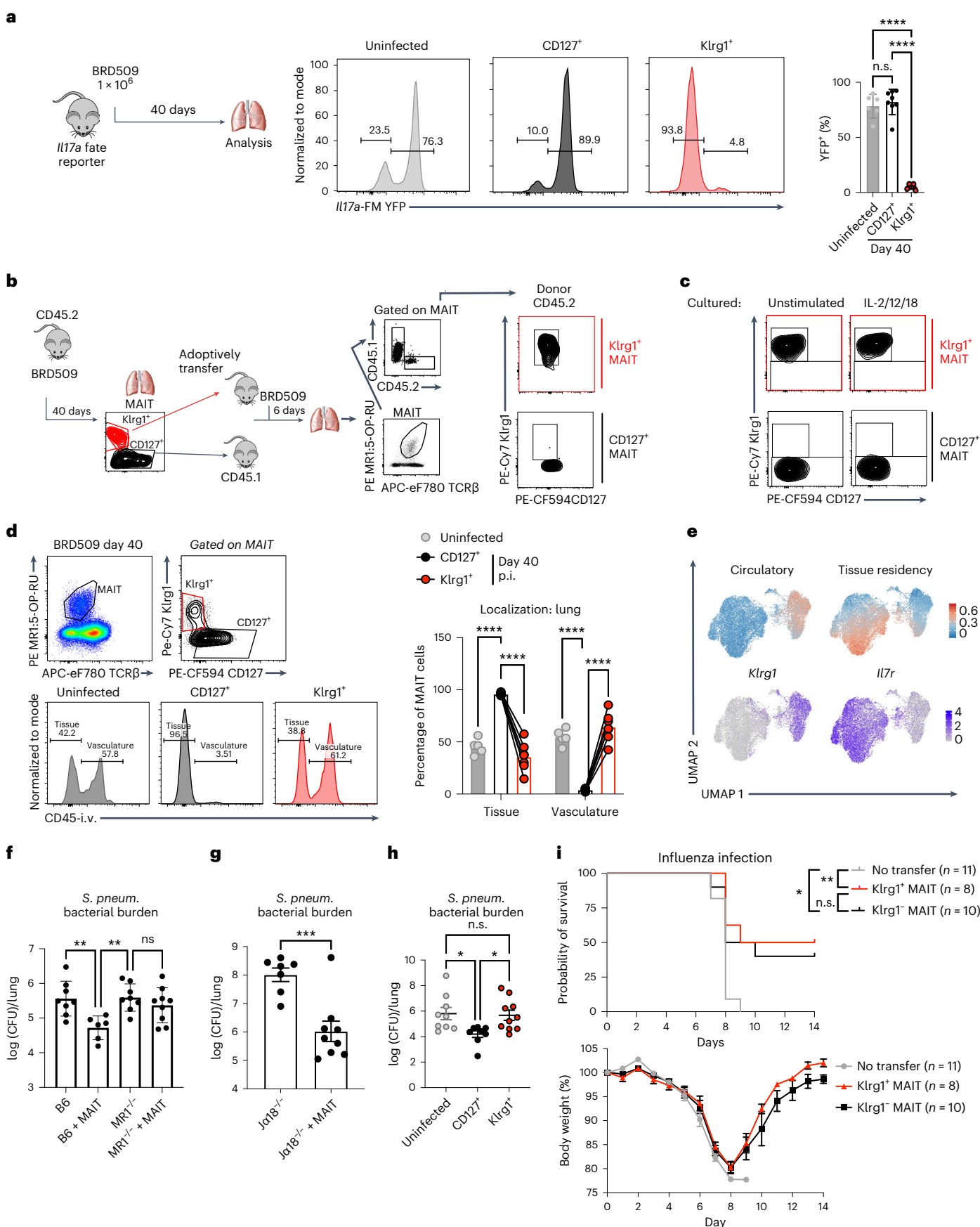

**Fig. 2 | Stable *aa*MAIT cell subsets differ in localization and protective capacity. a**, Il7a-FM YFP mice for FM *Il17a* expression were analysed at indicated times post BRD509 vaccination. Experimental setup (left). *aa*MAIT cell YFP expression is shown in representative histograms (middle) and quantified as percentage of YFP$^+$ cells (right). *n* = 6 (untreated), *n* = 8 (day 40) mice per group, combined from two independent experiments. Data displayed as mean ± s.d.; statistical significance assessed via ANOVA with Tukey's multiple comparisons test, ****$P$ < 0.0001. **b**, Experimental setup and gating (left). The indicated populations of CD45.2 donor lung *aa*MAIT cells at day 40 post vaccination were sorted and transferred into CD45.1 recipient mice. After gating on MR1:5-OP-RU tetramer$^+$ TCRβ$^+$ MAIT cells and CD45.2 (donor), Klrg1 and CD127 expression was plotted in recipient mice 6 days post vaccination with BRD509 (right). Representative data shown from one of two independent experiments. **c**, Populations of *aa*MAIT cells were gated as in **b**, sorted and cultured for 72 h under indicated stimulation conditions and analysed for expression of Klrg1 and CD127. Representative data shown from one of two independent experiments. **d**, Vascular localization of indicated *aa*MAIT cell subsets at indicated times post BRD509 vaccination represented as percentage of cells accessible to staining with intravenously injected CD45 antibody 3 min before tissue collection. Gating (left) and quantification (right). Paired *t*-test, *n* = 5 (untreated) *n* = 6 (day 40) mice

per group, ****$P$ < 0.0001. p.i., post infection. **e**, UMAP presentation of circulatory and tissue residency signatures from scRNA-seq data from pulmonary MAIT cells at day 0 and day 40 post BRD509 vaccination. **f–i**, Donor mice were vaccinated with BRD509. Total MAIT cells were sorted at day 7 (**f** and **g**) or subsets were sorted at day 40 (**h** and **i**). MAIT cells were adoptively transferred into recipient mice, and recipients were infected with *S. pneumoniae* (**f–h**) or influenza H1N1 (**i**). **f–h** show bacterial burden in lung 18 h post infection. In **f**, *n* = 8, 6, 8 and 9 mice per group, respectively, combined from two independent experiments, data displayed as mean ± s.d., one-way ANOVA with Tukey's multiple comparisons test, **$P$ = 0.0094, **$P$ = 0.0068, respectively. In **g**, *n* = 7 and 9 mice per group, respectively, unpaired *t*-test, ***$P$ = 0.0007, combined from two independent experiments. In **h**, *n* = 9, 8 and 10 mice per group, respectively, combined from two independent experiments, one-way ANOVA with Tukey's multiple comparisons test, *$P$ = 0.0274, *$P$ = 0.0397, respectively. In **i**, *n* = 11, 8 and 10 mice per group, combined from two independent experiments. Data displayed as mean ± s.e.m. Survival represented as Kaplan–Meier plot (top) and weight loss (bottom) following influenza infection of recipient mice. Log-rank (Mantel–Cox) test *$P$ = 0.0111, **$P$ = 0.0048. Source numerical data are available in source data. n.s., not significant.

*Il7r* expression, encoding CD127, marked *aa*MAIT Th17-like subsets, which also expressed *Tnf*, *Il17a* and *Rorc* (Fig. 1d). Confirming the transcriptional phenotypes, CD127$^+$ *aa*MAIT cells had a capacity to produce IL-17A, while Klrg1$^+$ *aa*MAIT cells had strongly elevated levels of Granzyme B, but little capacity to produce IL-17A (Fig. 1h). The expansion of Klrg1$^+$ *aa*MAIT cells could be due to increased proliferation as they expressed increased Ki67 after infection (Extended Data Fig. 3e). They are unlikely to be recruited from other tissues, as there were no pre-existing Klrg1$^+$ MAIT cells at steady state in any other organ (Extended Data Fig. 4a,b). When analysing T cell antigen receptor (TCR) β chain usage, there was no bias in the expression of predominant MAIT cell Vβ chains (Extended Data Fig. 4c). Taken together, vaccination with BRD509 induced a long-term MAIT cell increase, with two *aa*MAIT cell populations that were marked by CD127 and Klrg1 expression, respectively.

**Stable *aa*MAIT cell subsets differ in localization**

We employed cytokine gene expression fate mapping (FM) to explore if these subsets interconvert. The *Il17a*-FM YFP mice label cells and their precursors that at any point have expressed *Il17a*[33]. In naïve mice, the majority of pulmonary MAIT cells were YFP$^+$ (Fig. 2a)[28,34]. Forty days after infection, the great majority of CD127$^+$ *aa*MAIT cells also were YFP$^+$, while ~95% of Klrg1$^+$ *aa*MAIT cells were reporter negative. This indicated that Klrg1$^+$ *aa*MAIT cells never expressed *Il17a*, and probably represented a separate lineage. Analysis of mice that fate map T-bet expression confirmed that Klrg1$^+$ *aa*MAIT cells were *Tbx21*-reporter positive, and also expressed T-bet protein at 40 days post infection (Extended

Data Fig. 5a,b). Purified CD127$^+$ and Klrg1$^+$ *aa*MAIT subsets maintained their phenotypes after transfer and infection of recipients (Fig. 2b). Similarly, cytokine-mediated re-activation in vitro suggested that these subsets were stable (Fig. 2c). Of note, 10–20% of CD127$^+$ *aa*MAIT cells are *Il17a*-YFP$^-$ (Fig. 2a), and therefore, these possibly could give rise to either subset. Notably, these *Il17a*-YFP$^-$ cells expressed multiple transcripts typical for Klrg1$^+$ *aa*MAIT cells (Extended Data Fig. 5c–e), although for other genes they were more similar to CD127$^+$ *aa*MAIT cells that also have increased ΔΨm visualized with MitoTracker DR FM (MTR, Extended Data Fig. 5f). When adoptively transferred, MTR$^{low}$ *aa*MAIT cells had increased capacity to give rise to Klrg1$^+$ MAIT cells compared with CD127$^+$MTR$^{high}$ *aa*MAIT cells (Extended Data Fig. 5g). These data suggest that a minority subset of CD127$^+$ *aa*MAIT cells, enriched for CD8, MTR$^{low}$ and *Il17a*-YFP$^-$, gave rise to both *aa*MAIT cell populations.

We tested if the *aa*MAIT cell subsets localized differently in vasculature by intravenous injection of α-CD45 (ref. 35). Virtually all CD127$^+$ *aa*MAIT cells were inaccessible to the CD45 antibody, and therefore not in vasculature. Klrg1$^+$ *aa*MAIT cells, were distributed more equally (Fig. 2d). scRNA-seq data analysis confirmed a correlation of a previously published tissue residency gene expression signature[36] with CD127$^+$ *aa*MAIT cell clusters. A circulatory gene expression signature was increased in Klrg1$^+$ *aa*MAIT cells (Fig. 2e).

**aaMAIT cell subsets differ for protective capacity**

IL-17A production and tissue localization may be important for clearance of lung bacterial pathogens relevant to human health, including *S. pneumoniae*[24,37]. Therefore we tested if *aa*MAIT subsets conferred

**Fig. 3 | Effector response by re-activated *aa*MAIT cells. a–e**, Mice were uninfected (naïve), vaccinated with BRD509 and analysed 40 days later (BRD509), or analysed 10 h after *S. pneumoniae* infection; infected mice were naïve mice (*S. pneum.*) or vaccinated 40 days previously (BRD509 + *S. pneum.*). Pulmonary MAIT cells were pooled from ten mice per group and were sorted and analysed by scRNA-seq. UMAP representation of MAIT cell cluster composition separated by infection regimens as indicated (**a**) and combined UMAP representations (**b**). Data from uninfected and BRD509 infected mice are the same sequences as those in Fig. 1c. **c**, UMAP representation of tissue residency, circulatory, Th1 and Th17 gene signatures **d**, Feature plots of the indicated genes. **e**, Top five differentially expressed genes of each cluster represented as dot plots, where circle size represents percentage of cells expressing each gene and colour scale depicts relative expression value. **f**, Cytokine production by MAIT cell subsets in untreated and BRD509-vaccinated mice 16 h following infection with *S. pneumoniae* or without infection. Number of pulmonary MAIT cells positive for IL-17A, IFN-γ and Granzyme B are shown for each group. Data

displayed as mean ± s.d. *n* = 5, 7, 5 and 7 mice per group, respectively, combined from two independent experiments (left, middle). *n* = 5, 10, 5 and 11, mice per group, combined from three independent experiments (right). Statistical significance assessed via one-way ANOVA, with Tukey's multiple comparisons test; ****$P$ < 0.0001, *$P$ = 0.0487, **$P$ = 0.0029 (left); **$P$ < 0.0068 (middle); *$P$ = 0.0283 (right). **g–i**, Cytokine and Granzyme production by pulmonary MAIT cells isolated 40 days after vaccination with BRD509, cultured as pulmonary cell suspensions with or without re-activation as indicated. Protein production determined by intracellular flow cytometry. Representative histograms of total MAIT cell IL-17A and IFN-γ (**g**) or TNF and Granzyme B (**h**) and quantification (**i**) of each cytokine produced by either CD127$^+$ MAIT cells (black) or Klrg1$^+$ MAIT cells (red). *n* = 3 mice per group, representing one of two independent experiments. Two-way ANOVA corrected for multiple comparisons, left: ****$P$ < 0.0001; centre left: *$P$ = 0.0305, ****$P$ < 0.0001; centre right: *$P$ = 0.0283, **** < 0.0001; right: ****$P$ < 0.0001. Data displayed as mean ± s.e.m. Source numerical data are available in source data. n.s., not significant.

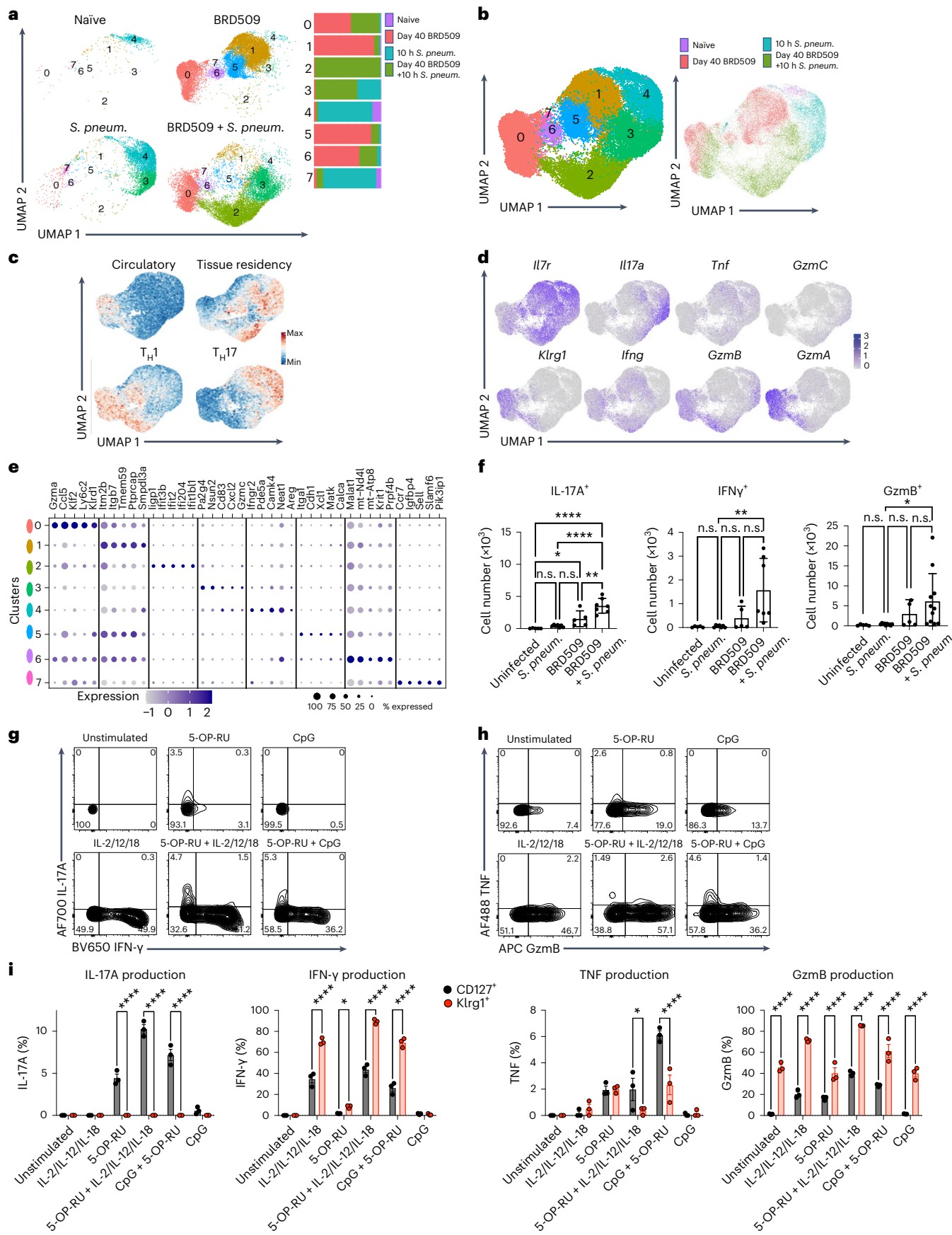

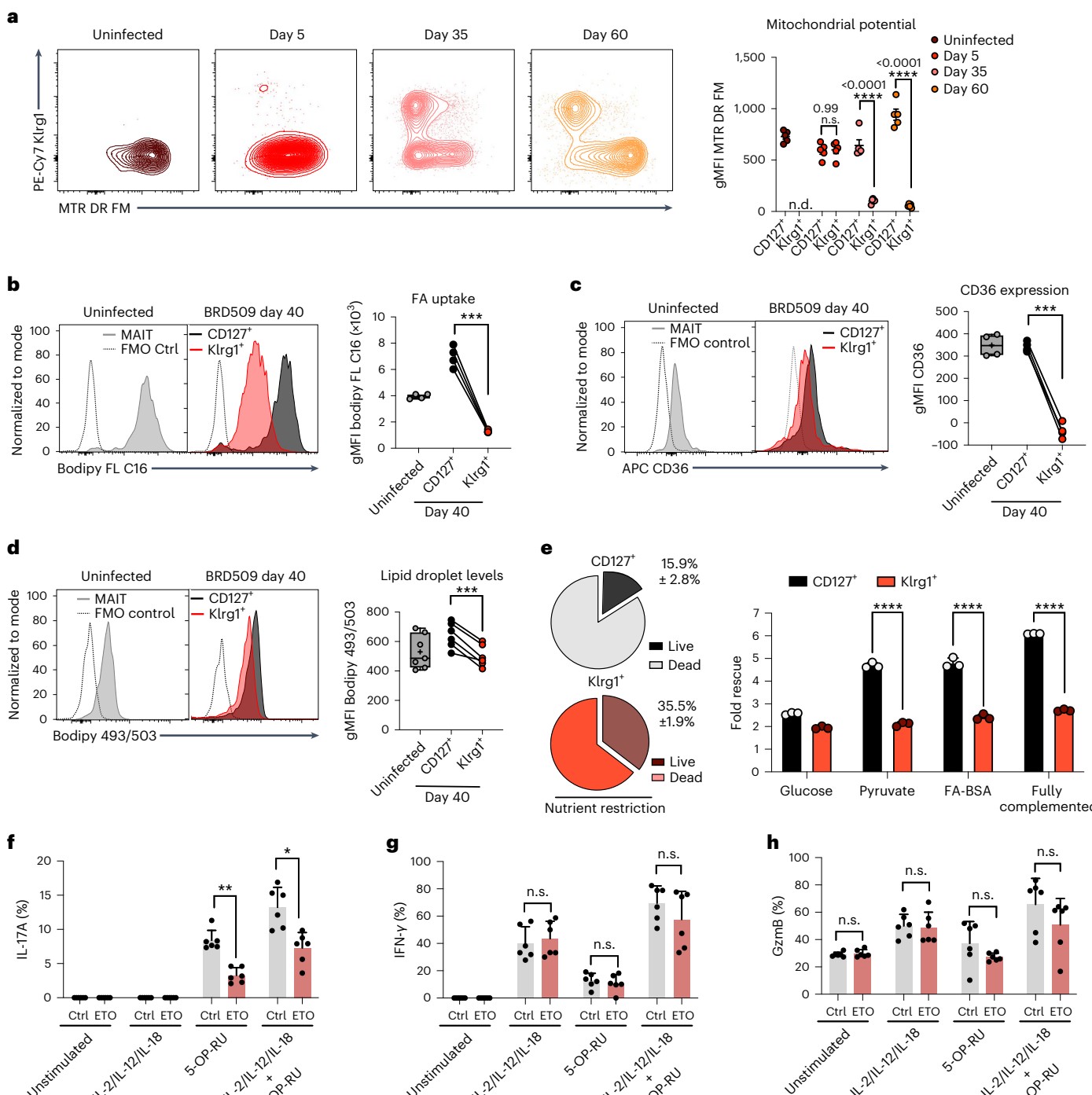

**Fig. 4 | CD127⁺ *aa*MAIT cells depend on FA uptake and oxidation. a–d,** Mouse lung *aa*MAIT cell subsets were analysed at steady state or at the indicated times post BRD509 vaccination. Representative histograms (left) and quantification (right) are plotted as geometric mean fluorescence intensity (gMFI). Mitochondrial membrane potential (**a**) quantified by MTR DR FM intensity at the indicated times post vaccination. n.d., Klrg1⁺ MAIT cells not detected. Two-way ANOVA, *n* = 5 mice per group from one of two independent experiments. Bodipy-FL C16 uptake of fluorescently labelled FA (**b**). Paired two-tailed *t*-test, ***P = 0.00014; *n* = 4 mice per group, *n* = 2 independent experiments. Surface expression of the scavenger receptor CD36 (**c**). Paired two-tailed *t*-test, ***P = 0.0006, *n* = 4 mice per group, *n* = 2 independent experiments. Neutral lipid droplet content quantified by Bodipy 493/503 fluorescence (**d**). Paired two-tailed *t*-test, ***P = 0.00075. *n* = 7 and 6 mice per group, *n* = 2 independent experiments. **e**, Left: lung MAIT cells at 40 days post BRD509 vaccination were subjected to nutrient restriction for 48 h, and percentage survival was measured via flow cytometry by viability dye positivity. Right: fold rescue of the percentage survival afforded by the indicated nutrient supplementations for CD127⁺ (black) or Klrg1⁺ (red) *aa*MAIT cells (right), ****P < 0.0001. *n* = 3 mice per group from one of two experiments. **f–h,** Pulmonary *aa*MAIT cells were isolated 40 days after vaccination with BRD509, cultured as cell suspensions for 18 h, with or without re-activation as indicated, and assessed for cytokine/granzyme production. FA oxidation/OXPHOS inhibition by Etomoxir (ETO, 90 μM). Two-tailed unpaired *t*-tests with false discovery rate testing: IL17A **P = 0.0050, *P = 0.02863; IFNγ: n.s.P = 0.659194, n.s.P = 0.581563, n.s.P = 0.258063; GzmB: n.s.P = 0.534601, n.s.P = 0.828580, n.s.P = 0.166802, n.s.P = 0.206856. *n* = 6 mice per group from one experiment. All data displayed as mean ± s.e.m. Source numerical data are available in source data. n.s., not significant.

differential protection against a clinical isolate of *S. pneumoniae*, strain URF918, that produces an antigen that activates MAIT cells[38]. Transfer of total lung *aa*MAIT cells into naïve wild-type mice reduced *S. pneumoniae* bacterial burden in an MR1-dependent manner (Fig. 2f), especially if the highly protective role of *i*NKT cells in mice was removed in *i*NKT cell-deficient *Jα18*[−/−] (*Traj18*[−/−]) mice (Fig. 2g). When transferred separately, protection from *S. pneumoniae* was mediated by CD127[+] *aa*MAIT, but not Klrg1[+] *aa*MAIT cells (Fig. 2h), consistent with data indicating that IL-17A-producing innate-like T cells are important for protection from this extracellular bacterium[37,39,40]. In contrast, either *aa*MAIT cell subset conferred a survival benefit when transferred to recipient mice infected with influenza virus A/PR/8/34 (H1N1) (Fig. 2i). Anti-viral protection was probably mediated via activation of *aa*MAIT cells by cytokines during viral infection[22]. These data suggest that the protection by different *aa*MAIT subsets depended in part on the nature of the infection.

## *aa*MAIT cell response to re-activation

We sought to characterize the mechanisms by which *aa*MAIT cell subsets enhanced protection. Re-infection with BRD509 at 9 weeks after vaccination led to a stronger MAIT cell response, and bacterial colony-forming units (CFUs) were more rapidly decreased (Extended Data Fig. 6a). To characterize the effector responses of *aa*MAIT cells to challenge with a heterologous pathogenic bacterium, we conducted scRNA-seq analysis comparing the response of steady-state lung MAIT cells or *aa*MAIT 10 h after *S. pneumoniae* infection. At 10 h after *S. pneumoniae*, multiple new MAIT cell clusters were evident (Fig. 3a,b). Cluster 0, composed equally of *aa*MAIT cells before and following *S. pneumoniae* infection, showed strong features of effector memory or terminal effector memory T cells[41], with increased expression of *Ifng*, *Gzmb* and *Gzma* transcripts. Steady-state MAIT cells responding to *S. pneumoniae*, especially cluster 4, were distinct from responding *aa*MAIT cells, especially cluster 2 (Fig. 3a,b). Both clusters expressed tissue residency genes (Fig. 3c), but cluster 4 cells had a predominant Th17 signature, while cluster 2 *aa*MAIT cells had a mixed Th1/h17 pattern (Fig. 3c–e). Also, expression of tissue repair genes[42] was detected in lung *aa*MAIT cells 40 days after BRD509 and in steady-state MAIT cells immediately after *S. pneumoniae* (Extended Data Fig. 6b). In contrast, an interferon response signature was increased in the immediately activated *aa*MAIT cell responders (Fig. 3e). A numerical increase in MAIT cells producing effector molecules was confirmed in *aa*MAIT cells responding to *S. pneumoniae*, as compared with responding steady-state MAIT cells (Fig. 3f and Extended Data Fig. 6c). Overall, these data indicate that re-activation of vaccine-mediated *aa*MAIT cells led to a quantitatively and qualitatively altered response.

As they also protected from viral challenge, we tested if the *aa*MAIT cell response in vitro was different in the presence or absence of 5-OP-RU antigen. In vitro re-activation experiments were carried out with IL-2 + IL-12 + IL-18, or with CpG, in the presence or absence of antigen. 5-OP-RU antigen was strictly required for IL-17A production, while

strong IFN-γ and Granzyme responses were induced through cytokine stimulation alone (Fig. 3g,h and Extended Data Fig. 6d). Importantly, IL-17A production was limited to CD127[+] cells, while a high frequency of Klrg1[+] *aa*MAIT cells produced IFN-γ and Granzyme B (Fig. 3i). These data provide an explanation for the different protective capacities of the *aa*MAIT cell subsets, whereby the presence of antigen, as during *S. pneumoniae* infection[38], elicited a qualitatively different *aa*MAIT cell response. Also, they provided an explanation for the differential protective capacities of *aa*MAIT lineages against an extracellular bacterium, which are sensitive to IL-17, and influenza viral challenge of the lung, which responds to several cytokines including IFNγ.

## CD127[+] *aa*MAIT cells depend on FA uptake and oxidation

The long-term alteration in effector functions in *aa*MAIT cells suggested they were more similar to a memory/effector memory-like state. Many MAIT cells are tissue resident[43,44], and a feature of tissue-resident memory T cells is their active consumption of fatty acids (FAs), coupled to FA oxidation within active mitochondria[45]. We therefore measured the mitochondrial membrane potential ($\Delta\Psi_m$), indicative of an active electron transport chain. At steady state, most lung MAIT cells expressed CD127 (Fig. 1e) and had high $\Delta\Psi_m$ (Fig. 4a). CD127[+] *aa*MAIT cells were even higher, while Klrg1[+] *aa*MAIT cells had very low $\Delta\Psi_m$ (Fig. 4a). FA uptake correlated with $\Delta\Psi_m$, with moderate levels of FA uptake at steady state, which was strongly enhanced in CD127[+] *aa*MAIT cells, and almost absent in Klrg1[+] counterparts (Fig. 4b). This also correlated with higher CD127[+] *aa*MAIT cell gene expression of lipid uptake receptors (Extended Data Fig. 7a), and high surface expression of CD36 (Fig. 4c). Lipid droplet content was also highest in CD127[+] *aa*MAIT cells (Fig. 4d). The greatly increased FA uptake and mitochondrial polarization of CD127[+] *aa*MAIT cells suggested that FAs may be continually metabolized in CD127[+] *aa*MAIT cells.

We tested if these metabolic differences impacted the capacity to survive in a nutrient-poor environment. When maintained in the absence of FA, amino acids or carbohydrate sources, Klrg1[+] *aa*MAIT cells had increased survival compared with their CD127[+] counterparts. Supplementation with pyruvate or bovine serum albumin (BSA)-conjugated FA, but not glucose, rescued survival of CD127[+] *aa*MAIT cells to a greater extent than Klrg1[+] *aa*MAIT cells (Fig. 4e). We tested if the requirement for an active, mitochondrial metabolic programme in CD127[+] *aa*MAIT cells extended also to their capacity for cytokine responses. Inhibition of fatty acid oxidation (FAO) using low-dose etomoxir[46] significantly reduced oxygen consumption rate (OCR) of T cells (Extended Data Fig. 7b) and reduced *aa*MAIT IL-17A production (Fig. 4f), but had no impact on IFN-γ production or Granzyme B (Fig. 4g,h). Therefore, FA uptake and active mitochondrial oxidation sustained survival and effector responses by CD127[+] *aa*MAIT cells.

## Klrg1[+] *aa*MAIT cells are more dependent on glucose

As Klrg1[+] *aa*MAIT cells depended less on mitochondrial activity and FAO, we hypothesized that they may be glycolysis dependent. By day

**Fig. 5 | Klrg1[+] *aa*MAIT cells depend on glucose consumption. a**, Pulmonary MAIT cells were assessed for uptake of fluorescent glucose 2-NBDG at the indicated timepoints post BRD509 vaccination. Representative histograms (left) and quantification (right). \*\**P* = 0.002736, \*\*\*\**P* < 0.0001. *n* = 5 mice per group from one of two independent experiments. **b**–**d**, Pulmonary MAIT cell subsets were isolated from mice 40 days post BRD509 vaccination and analysed for metabolic and functional parameters during culture for 18 h with or without re-activation with 5-OP-RU and/or IL-2, IL-12 and IL-18, as indicated. Kinetics of fluorescent glucose 2-NBDG uptake (added for last 30 min) at indicated durations of re-activation in minutes and quantified as percentage increase over baseline (**b**). *n* = 6 mice per group, *n* = 2 experiments. Representative histograms (left) and quantification (right) (**c**) of IFN-γ production following 18 h re-activation. Glycolysis was inhibited by 2-deoxy-glucose (2-DG, green colour). \*\*\*\**P* < 0.0001, n.s.*P* = 0.142985, \*\**P* = 0.001908; multiple two-tailed unpaired *t*-tests. *n* = 9, 9, 8 and

8 independent samples over *n* = 2 experiments. Dose and activation mode (TCR and/or cytokines) dependence of the concentration of 2-DG on IFN-γ production at 18 h re-activation (**d**). \*\**P* = 0.0085, \*\*\*\**P* < 0.0001; ANOVA with Dunnett's post-test for multiple comparisons. *n* = 3 independent samples per group from one experiment. **e**–**g**, Metabolic inhibition of effector functions was assessed in pulmonary *aa*MAIT cells isolated from mice 40 days post BRD509 vaccination. Dependence and capacity were calculated according to Extended Data Fig. 8d. Oligomycin A (O); 2-DG plus Oligomycin A (DGO). Ordinary one-way ANOVAs with Tukey's multiple comparisons testing. Left: \*\*\**P* = 0.0004, \*\*\*\**P* = 0.0002, \*\*\*\**P* < 0.0001; right: \*\*\**P* = 0.0003, \*\*\*\**P* < 0.0001, n.s.*P* = 0.1411 (**e**); \*P = 0.0247, n.s.*P* = 0.6949, \*P = 0.0210 (**f**); \*\**P* = 0.009319, \*\**P* = 0.003056, \*P = 0.029, \*P = 0.0106 (**g**). *n* = 6, 3, 3 and 3 from one of two experiments (**e** and **f**) or *n* = 3 from one of two experiments (**g**). All data displayed as mean ± s.e.m. Source numerical data are available in source data. n.s., not significant.

5 after infection there was a transient increase in the uptake of the labelled glucose molecule 2-(*N*-(7-nitrobenz-2-oxa-1,3-diazol-4-yl) amino)-2-deoxyglucose (2-NBDG) in both *aa*MAIT cell subsets, followed by a drop at later timepoints (Fig. 5a). In vitro, both subsets transiently increased labelled glucose uptake following activation. The kinetics and magnitude were similar, with Klrg1[+] *aa*MAIT cells trending towards higher uptake (Fig. 5b). Klrg1[+] *aa*MAIT cells also expressed higher Glut1

receptor compared with their CD127[+] counterparts at 40 days post infection (Extended Data Fig. 8a).

We tested the dependence on glycolysis for cytokine responses using the glycolysis inhibitor 2-deoxyglucose (2-DG). 2-DG inhibited glycolysis effectively and strongly reduced cytokine-induced IFN-γ in a dose-dependent manner, which was partially reversed in the added presence of 5-OP-RU (Fig. 5c,d and Extended Data Fig. 8b).

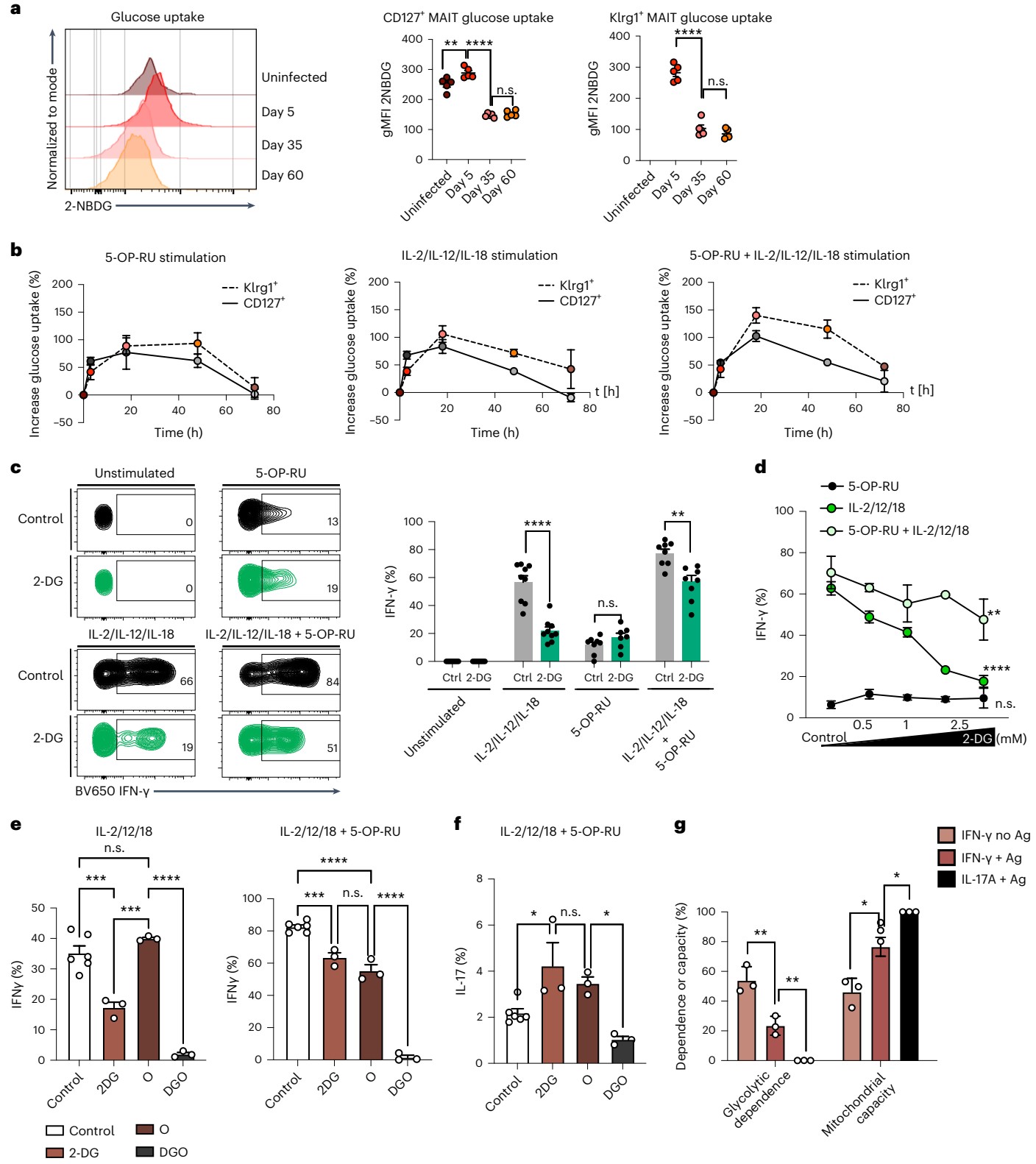

In contrast, IL-17A production was not impacted (Extended Data Fig. 8c). Taken together, IFN-γ secretion strongly required glucose consumption, while antigen-dependent IL-17A secretion was not affected. We confirmed and extended these findings in an assay that quantified metabolic dependencies (Extended Data Fig. 5d)[47]. This analysis revealed a high dependence on glycolysis by cytokine-induced IFN-γ, which was reduced by antigen (Fig. 5e,g). IL-17A production was not dependent on glycolysis and had a greater requirement for mitochondrial capacity (Fig. 5f,g). These data corroborated the finding that IFN-γ production by *aa*MAIT cells was most sensitive to glycolytic inhibition when elicited by cytokines (Fig. 5c) and suggested that the presence of antigen and signalling through the TCR reduced glycolytic dependence.

## Klrg1+ *aa*MAIT cells have responsive mitochondria

Although Klrg1+ *aa*MAIT cells had very low mitochondrial polarization and FA uptake at baseline, we determined if they induced mitochondrial pathways during antigen-mediated re-activation. Surprisingly, transmission electron microscopy (TEM) analysis revealed that Klrg1+ *aa*MAIT cells had mitochondria with regular cristae structure that were equally abundant compared with CD127+ *aa*MAIT cells (Fig. 6a). Morphologically, mildly increased circularity and reduced perimeter were evident (Fig. 6b), which correlated with reduced mitochondrial activity in Klrg1+ *aa*MAIT cells. Electron-dense cytotoxic granules were evident only in Klrg1+ *aa*MAIT cells, as expected (Figs. 6a,c).

Bodipy-mediated visualization of lipid stores confirmed flow cytometry data that indicated increased lipid droplet content in CD127+ *aa*MAIT cells (Fig. 6d,e). Co-staining of mitochondria with anti-Tom20 antibody, which does not depend on membrane potential, confirmed equal numbers of mitochondria in the *aa*MAIT cell subsets (Fig. 6d,f). However, visualization of $\Delta\Psi_m$ with MTR revealed a significantly smaller area and number of mitochondria with high intensity in Klrg1+ *aa*MAIT cells, and more low-intensity mitochondria were present instead (Fig. 6g–i). Mitochondrial OCR measurements confirmed that Klrg1+ *aa*MAIT cells had significantly reduced basal respiration, while the maximal respiratory capacity in response to FCCP-mediated uncoupling and the extracellular acidification rate (ECAR) were comparable between subsets (Fig. 6j). Higher baseline respiratory activity and energy-metabolic status of CD127+ *aa*MAIT was corroborated by significantly increased [13]C incorporation into ATP as measured by carbon flux mass spectrometry and higher ATP/AMP and ATP/ADP ratios (Extended Data Fig. 9a,b). Together, these data suggest that CD127+ *aa*MAIT cells were in an elevated, mitochondrially centred metabolic state, while Klrg1+ *aa*MAIT cells maintained relatively inactive mitochondria, with reduced $\Delta\Psi_m$ and less elongated morphology. To probe whether Klrg1+ *aa*MAIT cells mitochondria were metabolically responsive, re-activation of *aa*MAIT subsets was carried out.

Following stimulation, the high mitochondrial potential and FA uptake of CD127+ *aa*MAIT cells rapidly decreased, perhaps reflecting a shift towards anabolism-conducive metabolic programmes. In contrast, Klrg1+ *aa*MAIT cells increased their mitochondrial membrane potential and rapidly upregulated FA uptake (Fig. 6k). Together, these data show that Klrg1+ *aa*MAIT cells were in a metabolically poised state with low mitochondrial and FA metabolic activity, but were quickly able to engage these pathways in the context of antigen- or cytokine-induced activation.

## Mitochondrial respiration and autophagy modulate *aa*MAITs

The metabolic and functional characterization of lung *aa*MAIT cell subsets showed that disparate metabolic programmes correlate with divergent effector programmes in Klrg1+ and CD127+ memory-like *aa*MAIT cells. Furthermore, although both *aa*MAIT subsets were capable of utilizing glycolytic and mitochondrial pathways, they depended on different metabolic programmes for their survival and effector responses in vitro. We therefore tested if genetically impacting the metabolic balance could tune the balance of *aa*MAIT subsets. Deletion of the gene encoding the von Hippel Lindau (*Vhl*) E3-ligase prevents degradation of Hif1a and Hif2a transcription factors, resulting in a constitutive hypoxic signal, induction of glycolysis and limitation of oxygen-dependent pathways[48]. Accordingly, 40 days post infection of *Vhl*^f/f^ *dLck*-Cre mice, which delete the floxed *Vhl* gene in mature T cells, the uptake of fluorescent glucose in *aa*MAIT cells was increased while mitochondrial potential trended lower (Fig. 7a,b). This effect was normalized in *Hif1a*^f/f^ *Vhl*^f/f^ *dLck*-Cre mice, indicating that it was mediated by Hif1a. The proportion of Klrg1+ *aa*MAIT cells in the lung was increased, while the percentage of CD127+ *aa*MAIT cells was decreased, an effect that was eliminated by co-deletion of *Hif1a* (Fig. 7c,d and Extended Data Fig. 10a). Together, these data suggest that an increased glycolytic programme skewed the balance of *aa*MAIT cells in favour of the Klrg1+ subset.

Autophagy is a key pathway that enables robust mitochondrial metabolism[49–51]. Quantification of autophagic flux in LC3 reporter mice[52] with the autophagosome–lysosome fusion inhibitor Bafilomycin A1 (Baf) revealed a substantial accumulation of LC3+ autophagosomes indicating active autophagy in sorted CD127+ *aa*MAIT cells, but not their Klrg1+ counterparts (Fig. 7e–g). Deletion of the essential autophagy gene *Atg5* in mature MAIT cells led to increased glucose uptake and a trend towards lower mitochondrial potential in *aa*MAIT cells from *Atg5*^f/f^ *dLck*-Cre mice (Fig. 7h,i), indicative of a metabolic switch towards glycolysis. Accordingly, the balance of *aa*MAIT cell subsets was skewed towards Klrg1+ *aa*MAIT cells, which were increased at day 40 post BRD509 (Fig. 7j and Extended Data Fig. 10b). *Atg5*-deficient *aa*MAIT cells were less capable of producing IL-17A, while production of IFN-γ was not altered and Granzyme B was increased (Extended Data

**Fig. 6 | Klrg1+ *aa*MAIT cells have abundant but dormant mitochondria that are rapidly responsive to activation.** Sorted mouse pulmonary *aa*MAIT cell subsets were analysed for mitochondrial content and function at >40 days post BRD509 vaccination. **a**, Representative images. m, mitochondria; ld, lipid droplet; g, granule. Scale bar, 1 μm. **b,c**, Quantification of mitochondrial number and morphology (**b**) and cytotoxic granule count (**c**) from transmission electron micrograph sections of *aa*MAIT cell subsets. n.s.*P* = 0.504661. **P* = 0.003697, ***P* = 0.003851; unpaired two-tailed *t*-tests. In **b**, *n* = 22, 22, 21, 54, 22 and 54 cells from five pooled mice for each group. In **c**, *n* = 24 and 29 cells per group. **d–f**, Representative images (**d**) and quantification (**e** and **f**) of Bodipy493/503-positive lipid droplets (green) and mitochondrial Tom20 protein (purple) from z-stack composite airy-scan confocal micrographs of sorted *aa*MAIT cell subsets. Hoechst 33342 demarcates nucleus (blue). Scale bar, 5 μm; ***P* = 0.00130, n.s.*P* = 0.9; Mann–Whitney, two-tailed. *n* = 25, 20, 25 and 20 cells per group, pooled from five mice per group over two independent experiments. **g–i**, Representative images (**g**) and quantification (**h** and **i**) of mitochondrial $\Delta\Psi_m$ by MTR DR FM intensity. Hoechst 33342 demarcates nucleus (blue, in bottom row combined

channel images). Scale bar, 5 μm. Quantification represents the area (**h**) and count (**i**) of mitochondria that have high or low $\Delta\Psi_m$ signal, respectively. *n* = 27 and 20 cells per group, pooled from five mice per group over two independent experiments. ***P* = 0.0005 (**h**), ***P* = 0.0004, *P* = 0.047 (**i**); all Mann–Whitney, two-tailed. **j**, OCR kinetics (left) in response to injection of Oligomycin (Oligo), FCCP, Rotenone (Rot) and Antimycin A (AA) measured by Seahorse Bioanalyzer in sorted subsets of pulmonary *aa*MAIT cell subsets 40 days after vaccination with BRD509. OCR (middle) and basal ECAR (right) were measured without drug treatments or stimulation. ***P* = 0.0072, n.s.*P* = 0.84; unpaired, two-tailed *t*-tests. *n* = 8, 7, 6 and 4 independent samples per group, pooled from ten mice per group from one of two experiments. **k**, Quantification of mitochondrial $\Delta\Psi_m$ by MTR DR FM and uptake of Bodipy-FL C16 FA in MAIT cell subsets isolated from lungs 40 days post vaccination with BRD509 and cultured as pulmonary cell suspensions for 18 h with or without re-activation with antigen (5-OP-RU), cytokines (IL-2/-12/-18) or the combination. *n* = 6, 3 and 3 independent samples per group from one experiment. All data displayed as mean ± s.e.m. Source numerical data are available in source data. n.s., not significant.

Fig. 10c,d). Together, these data indicate that genetic alteration of the metabolic state of MAIT cells in vivo, through interference with oxygen sensing or impacting autophagy, was sufficient to tune the ratio and function of lung *aa*MAIT cell populations.

## Discussion

The concept of memory-like responses by innate immune cells, including cells in the lymphoid lineage, such as NK cells and innate lymphoid cells, has gained supportive evidence[5–8,53,54]. Similarly, long-term

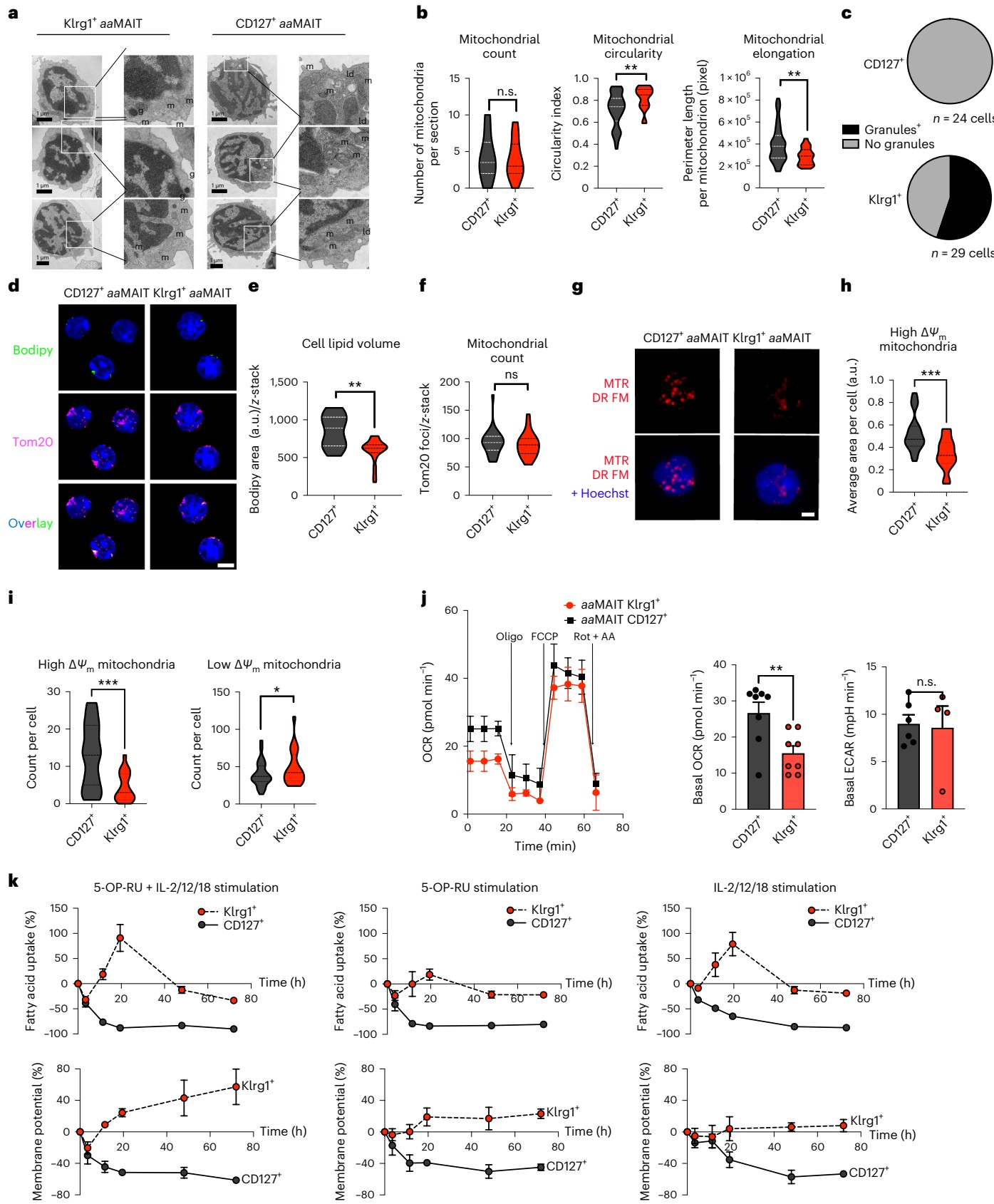

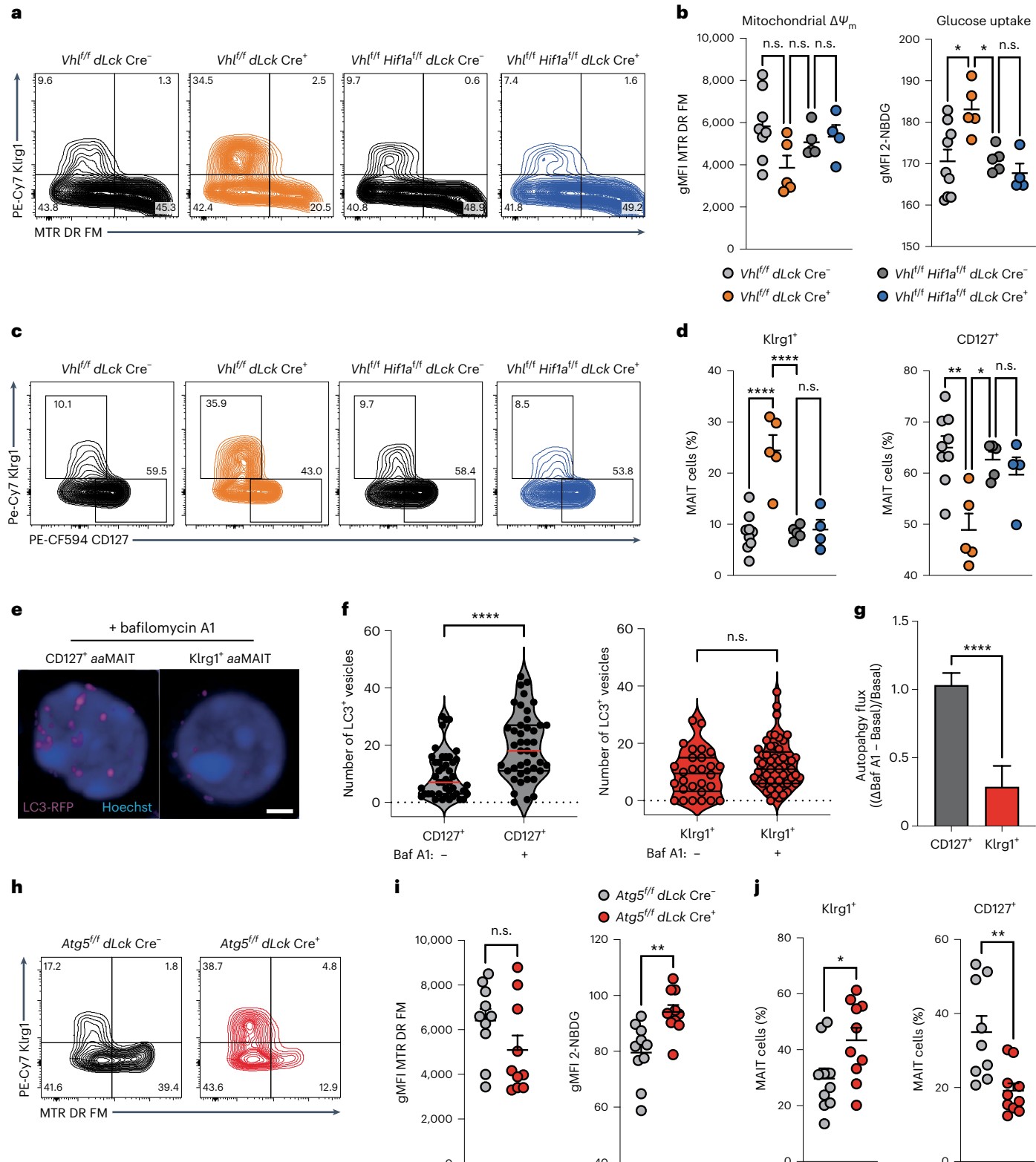

changes following antigenic stimulation in the population size and responses by MAIT cells have also been characterized[16–18,22,23]. We have confirmed these changes and used the term 'antigen-adapted' (*aa*) to describe their state, because MAIT cells acquire rapid effector functions in the thymus and may be considered natural memory. Here we describe the gene programmes that characterize the two states of *aa*MAIT cells in the lung after pulmonary vaccination that are distinct from steady-state MAIT cells. We show that there are two largely separate *aa*MAIT cell

subsets that differentially localize in lung tissue and have disparate effector functions and different metabolic programmes. Moreover, their respective metabolic programmes influence the balance of subsets and their ability to survive and carry out effector responses.

On the basis of cell transfers and fate-mapping experiments, expression of Klrg1 and CD127 defined two stable *aa*MAIT populations, despite infection and increases in population size. A minority population of Klrg1⁻CD127^low^MTR^low^ *aa*MAIT cells gave rise to both

**Fig. 7 | Oxygen sensing and autophagy facilitate CD127⁺ aaMAIT.** Pulmonary *aa*MAIT cells at >40 days post BRD509 vaccination from *Vhl*^f/f *dLck* Cre⁺ and *Vhl*^f/f *Hif1a*^f/f *dLck* Cre⁺ mice were analysed. **a**,**b**, Representative histograms (**a**) and quantification (**b**) of mitochondrial $\Delta\Psi_m$ (MTR DR FM) and labelled glucose uptake. In **b**, left: n.s.P = 0.072, n.s.P = 0.74, n.s.P = 0.92; right: *P = 0.0391, *P = 0.0118, n.s.P = 0.8829, ordinary ANOVAs with Tukey's multiple comparisons testing. n = 8, 5, 5 and 4 mice per group from n = 2 experiments. **c**,**d**, Representative histograms (**c**) and quantification (**d**) of pulmonary CD127⁺ and Klrg1⁺ *aa*MAIT cell ratios from mice with the indicated genotypes. n.s.P = 0.9962, ****P < 0.0001, **P = 0.0012, *P = 0.0140, n.s.P = 0.0864; ordinary one-way ANOVAs with Tukey's multiple comparisons testing. n = 8, 5, 5 and 4 mice per group from n = 2 experiments. **e**, Representative confocal z-stack projections of LC3-RFP in sorted *aa*MAIT cells

cultured in the presence of the autophagy flux inhibitor Bafilomycin A1 (Baf A1). **f**,**g**, Quantification of autophagosome buildup in presence of Baf A1 in sorted *aa*MAIT cell subsets (**f**) and calculation of autophagy flux as (ΔBaf A1 − Basal)/ Basal) (**g**). ****P < 0.0001, Mann–Whitney, two-tailed. In **f**, n = 70, 62, 60 and 29; in **g**, n = 45 and 44. **h**,**i**, Representative histograms (**h**) and quantification (**i**) of mitochondrial $\Delta\Psi_m$ (MTR DR FM) and labelled glucose uptake from *Atg5*^f/f *dLck* Cre⁺ mice and Cre-negative littermates. n.s.P = 0.129, **P = 0.0027, unpaired, two-tailed t-test. n = 10 mice from n = 2 experiments. **j**, Quantification of pulmonary CD127⁺ and Klrg1⁺ *aa*MAIT cell ratio from the indicated genotypes. *P = 0.029, **P = 0.0033; unpaired, two-tailed t-test. n = 10 mice from n = 2 experiments. All data displayed as mean ± s.e.m. Source numerical data are available in source data. n.s., not significant.

subsets, but it is uncertain if this represents a precursor that has the potential to give rise to both populations, or if within this group, there are already cells committed to one or the other lineage.

CD127 has been reported to identify CD8 memory precursor cells that display an increased ability to form long-lived CD8⁺ central memory cells, while Klrg1 defines short-lived effector cells[55,56]. CD8⁺, CD127⁻, Klrg1⁺ memory T lymphocytes, however, also can be found several months after infection[57–59], and a separate population of Klrg1⁺ terminal T effector memory cells has been described previously[41]. Our data show that Klrg1⁺ *aa*MAIT cells displayed enhanced effector functions that were maintained long term. Therefore, cells in both the CD127⁺ and Klrg1⁺ *aa*MAIT populations survived contraction after stimulation and displayed key features of memory. Klrg1⁺ *aa*MAIT cells expressed a cytotoxic effector programme and protected from lethal influenza challenge. Our data agree with a study in which adoptively transferred total pulmonary MAIT cells protected from viral infection in immune deficient *Rag2*^−/− *gc*⁻ mice through IFNγ[22]. The Klrg1⁺ *aa*MAIT cells in our study were not protective against *S. pneumoniae* infection, however, in which cytokine production by innate-like Th17 cells is important[37,39,40]. CD127⁺ *aa*MAIT cells required an antigenic signal to produce IL-17A in vitro, which can be provided by *S. pneumoniae*[38], providing a rationale for their protective role in that context.

In conventional CD4 and CD8 T cells, naïve, memory and effector states are controlled by distinct metabolic programmes, with an active FAO-fuelled mitochondrial programme crucial for memory T cell responses[25–27]. This raised the issue as to how this model applies to MAIT cells, which are not naïve. CD127⁺ *aa*MAIT cells had highly polarized and active mitochondria, in contrast to the low $\Delta\Psi_m$ of Klrg1⁺ *aa*MAIT cells. A similar 'ready to respond' or poised activation state, characterized by abundant but depolarized mitochondria, has recently been described for intraepithelial T lymphocytes[60]. Activation of intraepithelial T lymphocytes, similar to Klrg1⁺ *aa*MAIT cells, induces a rapid induction of mitochondrial membrane potential coinciding with a triggering of effector functions[60]. Furthermore, a previous analysis found a 'ready to respond' phenotype of human MAIT cells derived from peripheral blood mononuclear cells, with similarities to mouse Klrg1⁺ *aa*MAIT cells, including a Th1-cytokine bias, low mitochondrial activity, the ability to rapidly activate glucose uptake and inhibition of Th1-effector function by 2-DG[61]. An important difference is that mouse Klrg1⁺ *aa*MAIT cells already contained Granzyme B and cytotoxic granules before re-activation. Furthermore, IFN-γ production, but not Granzyme B, was susceptible to glycolytic interference in our study. Regardless, the similarity between Klrg1⁺ *aa*MAIT cells in mice and human PMBC-derived MAIT cells raises the question as to whether natural bacterial infections in humans influence the state of human MAIT cells[19].

In contrast to Klrg1⁺ *aa*MAIT cells, CD127⁺ *aa*MAIT cells continuously take up FAs, and have highly polarized mitochondria and active ATP turnover. In this regard, their metabolic programme is similar to CD8⁺ tissue-resident memory T cells ($T_{RM}$)[45,62]. CD8⁺ $T_{RM}$ cells require FABP4/5 driven mitochondrial β-oxidation of exogenous FAs for maintenance and function[45]. Likewise, CD127⁺ *aa*MAIT cell IL-17A production was sensitive to FAO perturbation, but unaffected by interference with

glycolysis, indicating that a similar metabolic programme may be operating, although *Fabp4/5* and *Ppar* genes are not highly expressed by *aa*MAIT cells.

Our studies linked the difference in metabolism in *aa*MAIT cell subsets to both survival and responsiveness. IFN-γ production by Klrg1⁺ *aa*MAIT cells was more dependent on glycolysis, and constitutive Hif1a activation by *Vhl* deletion enhanced glucose uptake and shifted the balance of *aa*MAIT cells towards the Klrg1⁺ subset. *Vhl* deletion in CD8⁺ conventional T cells also increased glucose uptake, and supported acquisition of an effector-memory phenotype in the context of viral infection[63]. CD127⁺ *aa*MAIT cells depended to a greater extent on autophagy, which is a key pathway for sustaining mitochondrial activity[49], for their accumulation and for IL-17A synthesis. In conventional memory T cells, autophagy disruption impaired mitochondrial health, inducing a shift towards glycolysis, increased IFN-γ and an effector phenotype[64,65]. While both autophagy and lipid oxidation are similarly essential for CD127⁺ *aa*MAIT cells and CD8⁺ memory T cells[64,66], it remains uncertain to what extent autophagy is facilitating FAO in these cases, and whether intrinsic lipolysis or IL-7-mediated synthesis of FA is critical[26,67,68]. Regardless, the CD127⁺ *aa*MAIT cells take up, store and break down FAs, and at least the latter is essential for their re-activation. The shift towards Klrg1⁺ *aa*MAIT cells in mice with increased Hif1a or decreased autophagy could reflect alteration in the survival of cells in these subsets, consistent with in vitro data, but we cannot rule out a role for differential impact on proliferation.

Interestingly, γδ T cells are also highly dependent on lipid metabolism for IL-17 effector function, whereas IFN-γ⁺ γδ T cells almost exclusively depended on glycolysis[9]. These subsets also were stable, and this had important functional implications for anti-tumour activity. The striking similarities between γδ T cells and *aa*MAIT cell subsets suggest that the metabolic signatures and dependencies described here may operate more broadly in innate-like T lymphocytes.

There is evidence that alterations in FA metabolism impact MAIT cell function in the context of obesity. In mice, obesity promoted MAIT17 cell inflammatory responses[69], and increased MAIT cell production of IL-17 also was observed in obese individuals[70,71]. IL-17-producing human MAIT cells from obese individuals also displayed lipid metabolic alterations[71]. This suggests a link between the cellular lipid metabolic programme key to IL-17A-producing *aa*MAIT cells discovered here, and human organ/organism-level diseases related to lipid dysbiosis, including metabolic syndrome and obesity.

Taken together, we discovered two fundamentally opposing metabolic dependencies that influence two stable, memory-like or *aa*MAIT cell subsets following infection, with implications for protective host responses and vaccinations that depend on MAIT cells[72].

## Online content

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

## Methods

### Animals

All procedures were approved by the La Jolla Institute for Immunology Animal Care and Use Committee and are compliant with the ARRIVE standards. Mice were bred and housed under specific pathogen-free conditions in the vivarium of the La Jolla Institute for Immunology (La Jolla, CA). C57BL/6J mice, *Atg5*[f/f], *Vhl*[f/f], *Hif1a*[f/f] and *dLck*-Cre mice were purchased from Jackson Laboratory and crossed to generate *Atg5*[f/f] *dLck*-Cre, *Vhl*[f/f] *dLck*-Cre and *Vhl*[f/f] *Hif1a*[f/f] *dLck*-Cre mice, to achieve gene deletions in mature T cells. Heterozygous Cre mice were used, together with littermate controls, in all experiments. All mice were on the C57BL/6J background. B6;CBA-Tg(*Tbx21-cre*)1Dlc/J (Tbet-cre) were bred with B6.Cg-*Gt(ROSA)26Sor*[tm14(CAG-tdTomato)Hze]/J mice (from Jackson Laboratories) to generate the *Tbx21* FM line that report with tdTomato fluorescence. B6.129(SJL)-*Il17a*[tm1.1(icre)Stck]/J with Cre recombinase knocked into the *Il17a* gene[33] were purchased from the Jackson laboratory and crossed to *Gt(ROSA)26Sor*[tm1(EYFP)Cos] mice to generate the *Il17a* FM strain, *Il17a*[Cre]*R26R*[eYFP] mice or *Il*17a-FM YFP mice. Mice deficient for iNKT cells, *Ja18*[−/−] (*Traj18*[−/−]) mice, were generated as described previously[73]. *Mr1*[−/−] mice were kindly provided by Dr Gilfillan (Washington University, St. Louis, MO)[74]. C57BL/6-[TgCAG-RFP/EGFP/Map1lc3b] transgenic mice for reporting on LC3 were from Jackson Laboratory. Male and female mice were used at 6–12 weeks of age. Mice were group-housed under a standard rodent chow (Pico Lab Diet 20, #5053) at ambient temperature (68 °F), 50% humidity in average and a 12 h light–dark cycle in individually ventilated cages.

### Antibodies and other materials

MR1 tetramers loaded with either 5-OP-RU or 6-FP were obtained from the NIH Tetramer Facility. Antibodies with clone, dilution, catalogue number and supplier indicated in parentheses: anti-mouse CD16/32 (2.4G2, cat. no. 553142, 1:500, BD Bioscience) anti-mouse Sdc1 (281-2, 1:200, cat. no. 564068 BD Bioscience), anti-mouse CD36 (HM36, 1:200, cat. no. 102608, BioLegend), anti-mouse CD45 (30-F11, 1:800, cat. no. 564225 BD Bioscience); anti-mouse IgD (clone 11-26c.2a, 1:200, cat. no. 564273 BD Bioscience); anti-mouse Klrg1 (clone 2F1/KLRG1, 1:300, 25-5893-82 BD Bioscience); anti-mouse γδTCR (clone GL3, 1:300, cat. no. 553178 BD Bioscience); anti-mouse TER-119 (TER-119, 1:200, cat. no. 116204 BioLegend) anti-mouse CD127 (clone SB/199, 1:200, cat. no. 562419 BD Bioscience); anti-mouse CD8α (53-6.7, 1:400, cat. no. 47-0081-82 eBioscience); anti-mouse CD8β (H35-17.2, 1:400, cat. no. 741992 BDBioscience), anti-mouse EpCAM (clone G8.8, 1:200, cat. no. 13-5791 eBioscience); anti-mouse IFN-γ (clone XMG1.2, 1:400, cat. no. 505831 BioLegend); anti-mouse TNF (clone MP6-XT22, 1:400); anti-mouse IL-17A (clone TC11-18H10, 1:400, cat. no. 506914 BioLegend); anti-mouse CD69 (clone H1.2F3, 1:200, cat. no. 104530 BioLegend); anti-mouse CD44 (IM7, 1:200, cat. no. 586116 BD Bioscience), anti-mouse Ly6A/E (D7, 1:200, cat. no. 108110 BioLegend), anti-mouse Icos (C398.4a, 1:200, cat. no. 313520 BioLegend), anti-human/mouse T-bet (clone O4-46, 1:16 cat. no. 562467 BD Bioscience); anti-mouse RORγT (clone B2D, 1:100, cat. no. 46-6981-82 eBioscience); anti-mouse Ly6G (clone 1A8, 1:400, cat. no. 127639 BioLegend); anti-mouse CD11b (M1/70, 1:400, cat. no. 561114 BD Bioscience); anti-mouse CD45R/B220 (RA3-6B2, 1:200 cat. no. 552771 BD Bioscience); anti-mouse CD11c (N418, 1:200, cat. no. 48-0114-80 eBioscience); anti-mouse TCRβ (H57-597, 1:300, cat. no. 47-5961-82 eBioscience), anti-mouse Ki67 (B56, 1:16, cat. no. 561281 BD Bioscience); anti-mouse CD80 (16-10A1, 1:200, cat. no. 553768 BD Bioscience); anti-mouse CD86 (GL1, 1:200); anti-mouse MR1 (26.5, 1:200, cat. no. 361107 BioLegend); anti-mouse Granzyme B (GB11, 1:100, cat. no. GRB05 Invitrogen); anti-mouse CD103 (2e7, 1:200, cat. no. 749393, BD Bioscience), anti-mouse siglec F (E50-2440 1:200, cat. no. 565527 BD Bioscience), anti-mouse Tom20 antibody (D8T4N, 1:400, cat. no. 13929, Cell Signaling).

### 5-OP-RU synthesis

5-A-RU HCl synthesis was based on Li et al.[75] and used to generate 5-OP-RU by reaction with methylglyoxal. For synthesis details, please refer to Supplementary Note 1.

### Bacterial strains and infections

The attenuated strain of *Salmonella enterica* serotype Typhimurium BRD509 was a generous gift from Stephen P. Schoenberger (La Jolla Institute for Immunology, La Jolla, CA). The synchronized lysis circuit in the ELH1301 strain was described previously[32]. The circuit includes the luxl promoter that uses quorum firing to induce cell death due to expression of the φX174 E bacteriophage lysis gene. The Δ*SpiB* and Δ*InvA*Δ*SpiB* strains of *S. enterica* serotype Typhimurium were a generous gift from the Manuela Rafatellu Lab (University of California, San Diego, CA). The Δ*RibDH* mutant of BRD509 was a generous gift by Richard Strugnell (University of Melbourne, Australia). All strains were grown overnight in Luria broth with the appropriate antibiotics and growth additives, then subcultured the next day for 3 h to mid-log phase, except for the synchronized lysis mutant strain that was grown to an optical density of 0.1. Bacteria were then washed twice in phosphate-buffered saline (PBS) and resuspended in PBS for inoculation. *S. enterica* strains were administered at a dose of $1 \times 10^6$ CFU per mouse, except for the Δ*RibDH* mutant, which was given at a dose of $1 \times 10^7$ CFU per mouse, as previously described[16], and each dose was verified by plating serial dilutions on LB agar.

*Streptococcus pneumoniae* serotype 3 strain URF918 is a clinical isolate[76]. URF918 was cultured in Todd–Hewitt broth (BD Biosciences) at 37 °C in an incubator at 5% $CO_2$, collected at a mid-log phase and washed twice in PBS and resuspended in PBS for infection ($1-3 \times 10^6$ CFU per mouse). For calculation of total lung bacterial burden, tissues were collected 16 h after infection and were homogenized in PBS to assess bacterial burden. Homogenates were inoculated at different dilutions in a volume of 50 ml on 5% sheep blood agar plates (Hardy Diagnostics) and cultured for 18 h, followed by counting of colonies. For retropharyngeal inoculations, mice were anaesthetized with isoflurane and elevated on a board. Mice were inoculated with bacteria at a volume of 50 μl by insertion of a pipet tip into the trachea.

### Isolation of cells

Lung tissue was digested with spleen dissociation medium (Stemcell), and mechanically dissociated using a GentleMacs Dissociator (Miltenyi). Cells were then strained through a 70 μm filter and washed with Hanks' Balanced Salt Solution supplemented with 10% foetal bovine serum (FBS) followed by red blood cell lysis. For adoptive transfer experiments, penicillin–streptomycin (Gibco) was added to media throughout the experiment.

### Flow cytometry

For staining of cell surface molecules, cells were suspended in staining buffer (PBS, 1% BSA and 0.01% $NaN_3$) and stained using phycoerythrin (PE)- or allophycocyanin (APC)-conjugated MR1 tetramers at a dilution of 1:300 in staining buffer for 30 min at room temperature followed by staining with fluorochrome-conjugated antibodies at 0.1–1 μg per $10^7$ cells. Cells were stained with Live/Dead Yellow (Thermo Fisher) at 1:500 and blocked with anti CD16/32 (2.4G2) antibody at 1:500 and free streptavidin at 1:1,000 for 15 min at 4 °C, continued with surface antibody staining for 30 min at 4 °C. For cytokine staining, cells were previously stimulated with 50 ng ml$^{-1}$ of phorbol myristate acetate (PMA) and 1 μg ml$^{-1}$ of ionomycin for 2 h at 37 °C and then incubated in GolgiStop and GolgiPlug (both from BD PharMingen) for 2 h at 37 °C. For in vitro re-activation experiments, cultures were carried out for 18 h without stimulation or with 5-OP-RU (1 μM) and/or IL-2 + IL-12 + IL-18 (10 ng ml$^{-1}$ each). GolgiStop and GolgiPlug (both from BD PharMingen) were added for the last 2 h. Following *S. pneumoniae* infection, cells

were incubated in GolgiStop and GolgiPlug for 2 h at 37 °C with no restimulation. For intracellular staining, cells were fixed with CytoFix (BD) for 20 min, and permeabilized with Perm 1X solution (Thermo Fisher) overnight with antibodies for intracellular marker detection. For high-parameter flow cytometry experiments, data were acquired on Fortessa and LSR-II cytometers with FACS DIVA 8.0 (BD Biosciences) and analysed with FlowJo v10.8.1 (TreeStar).

## Discrimination of tissue and circulating MAIT cells
Mice were anaesthetized with isofluorane and injected retro-orbitally with 3 μg of AlexaFluor-700-labelled anti-CD45 antibody (30-F11), as described previously[35]. After 3 min, the lungs were removed for processing.

## Cell enrichment and cell sorting
MAIT cell enrichment before sorting was achieved by negative selection of cells using biotinylated antibodies against CD11b, CD11c, F4/80, B220, Gr1, Ly6G, IgD, Epcam and γδTCR together with Rapidspheres (StemCell Technologies #19860) and either the Big Easy (StemCell Technologies #18001) or Easy Eight magnets (StemCell Technologies #18103) according to respective protocols. MAIT cells were sorted using FACS Aria III and FACS Fusion instruments (BD Biosciences).

## Adoptive transfer experiments
For adoptive transfer experiments, BRD509-expanded total lung MAIT cell populations or Klrg1+ and Klrg1− MAIT cell subsets were sorted as described above. Cells were transferred into gender-matched recipient mice by retro-orbital injection. Five to 7 days post-transfer, mice were infected via retropharyngeal inoculation with *S. pneumoniae* ($3–5 \times 10^6$ CFU per mouse) or influenza (100 p.f.u. per mouse, A/PR/8/34(H1N1)). For *S. pneumoniae* experiments, bacterial burden was assessed 18 h post infection. For influenza infection, weight loss and survival were monitored for 14 days following infection.

## Metabolic assays
Metabolic cytometry-based assays have been described previously[77]. Briefly, cells were stained with MitoTracker Deep-Red (Life Technologies) at 100 nM concentration, 37 °C, 5% $CO_2$ for 30–45 min in RPMI 1640 (Gibco) containing 5% FBS. For glucose uptake measurements, cells were incubated in glucose-free medium containing 5 μg ml$^{-1}$ 2-NBDG (Thermo Fisher) and 2.5% FBS at 37 °C, 5% $CO_2$ for 30 min or the time indicated. For lipid droplet quantification, cells were incubated in medium containing 1 μg ml$^{-1}$ Bodipy 493/503 (Thermo Fisher) for 30 min. Uptake of FAs was quantified after incubation with 1 μM 4,4-difluoro-5,7-dimethyl-4-bora-3a,4a-diaza-s-indacene-3-hexadecanoic acid (Bodipy-FL C16, Thermo Fisher) at 37 °C, 5% $CO_2$ for 30 min. Optimal incubation periods for metabolic dye and metabolite uptake depended on tissue and required fluorescence intensity, but did not exceed 45 min, except where indicated. For nutrient supplementation experiments, minimum essential medium was supplemented with glucose (5 mM); pyruvate (10 mM); FA mix (10 μg ml$^{-1}$, BSA-conjugated) or fully complemented +10% FBS. Data were acquired using Fortessa or LSR II flow cytometers (BD Biosciences) and analysed with FlowJo v10.8 software (BD Life Sciences). Metabolic marker fluorescence intensity depends on the instrument type and laser intensity, and therefore quantitative comparisons between different experiments cannot be directly compared.

## Metabolic flux analysis
The real-time ECAR and OCR were measured using a XF-96 extracellular flux analyzer (Seahorse Bioscience). A total of $1 \times 10^5$ MAIT cells were sorted from lung and washed twice in RPMI 1640 without sodium bicarbonate, 20 mM glucose, 5% FBS and 2 mM pyruvate and seeded in corresponding assay medium in a XF plate coated with poly-L-lysine (Sigma). Cells were rested for 1 h at 37 °C before analysis according to the mitochondrial stress test protocol (Seahorse Bioscience).

## Mass spectrometry
A total of $1 \times 10^5$ MAIT cells were sorted by flow cytometry. Isolated cells were incubated in RPMI (glucose-free formulation) containing 10 mM [U-$^{13}$C]glucose (Cambridge Isotope Laboratories), 2 mM glutamine and 10% dialysed FBS (Thermo Fisher) at 37 °C for 1 h. Cells were washed in 150 mM of ice-cold ammonium acetate, pH 7.3, and metabolites were extracted in 80% methanol on dry ice before evaporation under vacuum. Dried metabolites were resuspended in 50 μl of 50% acrylonitrile, and 5 μl was injected for chromatographic separation using the Ion Chromatography System 5000 (Thermo Fisher) coupled to a Q Exactive run in negative polarity mode (Thermo Fisher). The gradient ran from 5 mM to 95 mM KOH, and HESI-II source settings were S-lens, 50; sheath gas, 18; aux gas, 4; spray heater, 320 °C; and spray voltage, −3.2 kV. Metabolites were identified on the basis of accurate mass (±3 p.p.m.) and retention times of pure standards. Relative amounts, mass isotopologue distributions and fractional contributions of metabolites were quantified using TraceFinder 3.3.

## Microscopy
For TEM, $1 \times 10^5$ MAIT cells were sorted by flow cytometry and fixed in 2% glutaraldehyde in 0.1 M sodium cacodylate buffer and processed according to the University of California San Diego EM core protocol. Images were acquired on a JEOL 1400 plus microscope equipped with a bottom-mount Gatan OneView (4k × 4k) camera. For confocal microscopy, $1 \times 10^5$ MAIT cells were sorted and then stained with antibodies or metabolic dyes as described for flow cytometry, and fixed in 2% para-formaldehyde for 30 min. Tom20 (D8T4N, Cell Signaling Technologies) antibody was used for detection of mitochondria. Cells were cytospun on glass coverslips and mounted in the presence of DAPI. Images were acquired on a ZEISS LSM 880 inverted confocal microscope with a 63×/1.46 numerical aperture plan-apochromat objective running Zeiss Zen Blue v3. The Airyscan module was used to improve resolution and signal-to-noise ratio. Automated image quantification was performed in Imaris 10 (Bitplane) using the spot detection algorithm.

## scRNA-seq assay (10x Genomics platform)
Cells were sorted by flow cytometry into a low retention 1.5 ml collection tubes, containing 500 μl of a solution of PBS:FBS (1:1) supplemented with RNase inhibitor (1:100). After sorting, ice-cold PBS was added to make up to a volume of 1,400 μl, then spun down, and single-cell libraries were prepared as per the manufacturer's instructions (10x Genomics). Samples were processed using 10x v2 chemistry as per the manufacturer's recommendations; 11 and 12 cycles were used for cDNA amplification and library preparation, respectively.

## Single-cell transcriptome analysis
Libraries were mapped with Cell Ranger v6.1.2 count pipeline. Then multiple libraries were aggregated with the aggr pipeline. Aggregated data were then imported into the R environment where Seurat (4.1.1) was used to filter and find clusters. Cells with fewer than 200 genes and more than 2,500 genes were discarded. Furthermore, cells with more than 5% unique molecular identifiers coming from mitochondrial genes were filtered out. Genes expressed in fewer than three cells were ignored. The gene expression matrix was then normalized and scaled. Principal component analysis was performed on the scaled data, and, on the basis of the elbow plot, 12 principal components were selected for clustering, with a default resolution (0.6), and a perplexity of 100 was chosen for the dimensionality reduction in Fig. 3. Principal component analysis was carried out separately for representations of the data subset that excludes *S. pneumoniae* infection data in Fig. 1. To determine each cluster's enriched genes (markers), Seurat's v4.1.1 FindAllMarkers function was used with test.use = MAST (adjusted *P* value <0.05 and |log fold change| >0.25). Signature module scores were calculated with Seurat's AddModuleScore function using default

parameters. This function calculates the average expression levels of gene set of interest, subtracted by the aggregated expression of control gene sets randomly selected from genes binned by average expression.

## Statistics and reproducibility

All data are shown as mean ± standard error of the mean (s.e.m.) or mean ± standard deviation (s.d.), as stated in the figure legends. We did not use any criteria to determine the sample size. Depending on the nature of the experiments and availability of mice we collected the maximum possible dataset. Where single-cell analysis was performed, we scored at least 20 cells per condition in each biological replicate. Data were excluded only after performing outlier tests, and biological replicates were always kept separate except when cell number restriction required pooling. These instances are detailed in the figure legends. The number of animals, replicates and experiments is indicated in each figure legend. No blinding method was applied. No statistical method was used to pre-determine sample sizes. All graphs and statistical analysis were generated using Excel v16 or Prism 9 software (GraphPad Software).

## Reporting summary

Further information on research design is available in the Nature Portfolio Reporting Summary linked to this article.

## Data availability

Bulk and sc-RNA-seq data that support the findings of this study have been deposited in the Gene Expression Omnibus (GEO) under accession code GSE226524. Publicly available RNA-seq data were used from the C7 immunological database: accession codes GSE1000002_1582_200_UP and GSE1000002_1582_200_DN. Source data are provided with this paper. All other data supporting the findings of this study are available from the corresponding authors on reasonable request.

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

## Acknowledgements

We thank the La Jolla Institute (LJI) Flow Cytometry core for assisting with cell sorting. We are grateful to H. Simon, C. Ramirez Suastegui, A. Sethi and the Sequencing and Bioinformatics cores for performing scRNA-seq and analysis. We thank S. P. Schoenberger (La Jolla Institute for Immunology, La Jolla, CA), M. Rafatellu (University of California, San Diego, La Jolla, CA) and R. Strugnell (University of Melbourne, Melbourne, Australia) for generously providing bacterial strains used in this study. We thank Y. Jones and the UCSD electron microscopy facility for help with TEM sample processing and access to microscopes. J. ten Hoeve-Scott and the UCLA metabolomics core kindly helped with metabolomic mass spectrometry experiments. Funding was provided by National Institutes of Health grant AI71922 (M.K.), National Institutes of Health grant AI137230 (M.K.) and National Institutes of Health grant T32 AI125179 (M.P.M.). This research was funded in part by the Wellcome Trust, grant 210842_Z_18_Z (T.R.). For the purpose of open access, the authors have applied a CC-BY public copyright licence to any Author Accepted Manuscript version arising from this submission. M.K. received funding by the Bill and Melinda Gates Foundation, and C.A.B. by National Institutes of Health grant AI101423. S.M. was funded by an Imaging Scientist grant (2019-198153) from the Chan Zuckerberg Initiative. Utilized equipment was supported by National Institutes of Health grant S10RR027366 (BD FACSAria), National Institutes of Health grant S10OD025052 (NovaSeq 6000), National Institutes of Health grant S10-OD016262 (Illumina HiSeq 2500) and National Institutes of Health grant S10-OD021831 (Zeiss LSM 880).

## Author contributions

Conceptualization: T.R., M.K. and M.P.M. Investigation and analysis: T.R., C.W., M.P.M., T.-F.C., S.C., C.D. and S.B. Bioinformatics analysis and scRNA-seq pipeline: T.R., S.C., V.C.-C., S.B., S.M., G.S. and P.V. Funding acquisition: T.R. and M.K. Supervision: T.R. and M.K. Tools: J.D. and A.N. Writing—original draft: T.R. and M.K. Writing—review and editing: T.R., M.K., M.P.M., C.W., S.C., V.C.-C., T.-F.C., S.M., C.D., S.B., D.M.L., J.H., G.S., C.A.B. and P.V.

## Competing interests

The authors declare that they have no competing interests.

## Additional information

**Extended data** is available for this paper at https://doi.org/10.1038/s41556-023-01152-6.

**Correspondence and requests for materials** should be addressed to Thomas Riffelmacher or Mitchell Kronenberg.

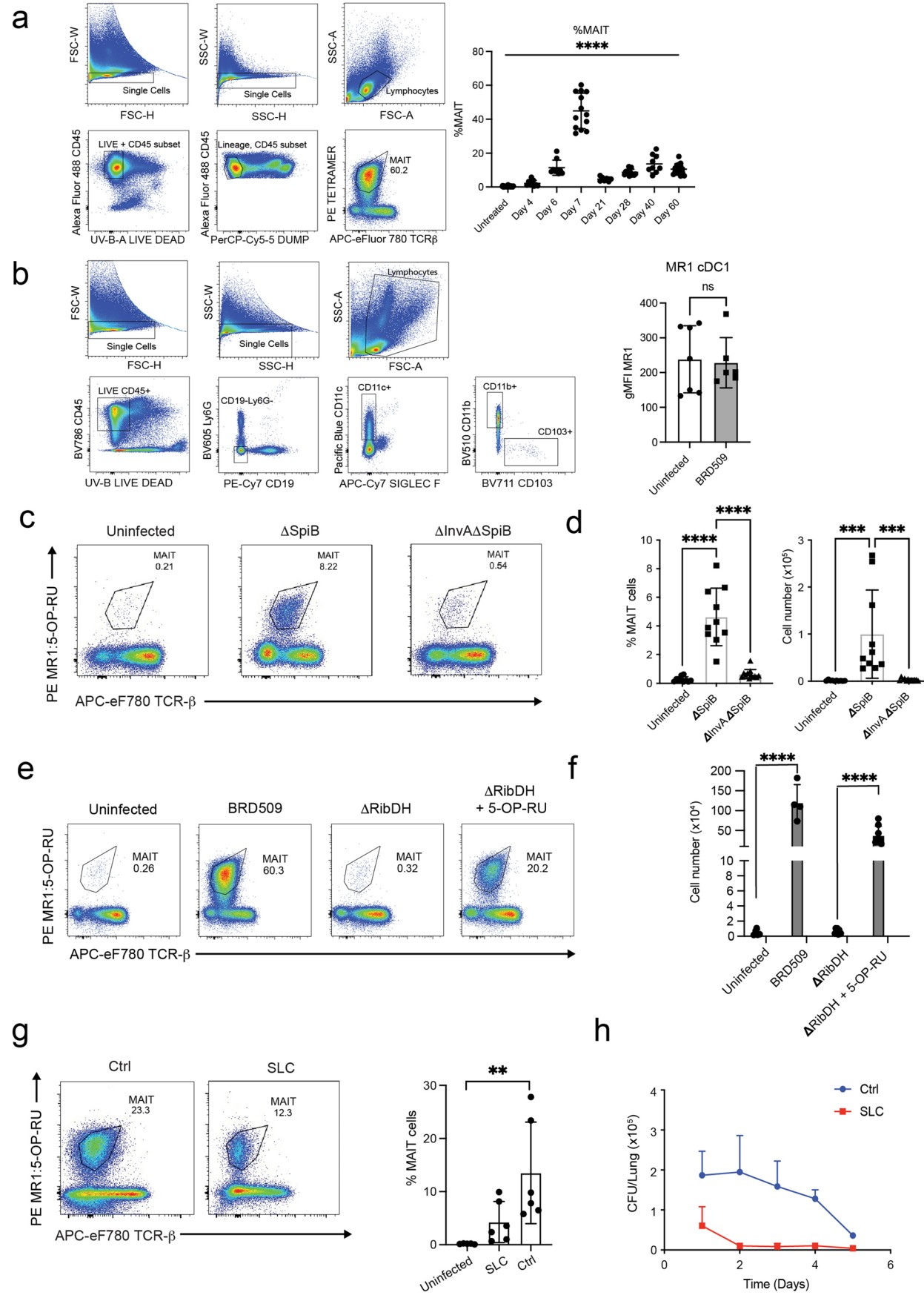

Extended Data Fig. 1 | See next page for caption.

**Extended Data Fig. 1 | Invasive Salmonella strains induce an effector response. a**) Gating strategy for MAIT cells in lung (left) and quantification of MAIT cells expression at different times post BRD509 (right). n = 36, 9, 9, 13, 7, 13, 9, 16, mice per group, respectively, combined from 12 independent experiments. One way ANOVA, ****$P$ < 0.0001. **b**) Gating strategy for CD11b⁻CD103⁺ conventional dendritic cells (cDC1) and CD11b⁺ CD103- (cDC2) (left), and MR1 quantification as geometric MFI in cDC1 (right). n = 7 (untreated), n = 6 (BRD509), unpaired two-tailed t test. **c**) Representative flow cytometric analysis and (**d**) Quantification of MAIT cells 6 days after infection with invasion mutant (Δ*InvA*Δ*SpiB*) or control strain (Δ*SpiB*). Combined from 3 independent experiments, n = 10, 10, 12 mice per group, respectively. One-way ANOVA, with Tukey's multiple comparisons test, ****$P$ < 0.0001

(left), ***$P$ = 0.0006, ***$P$ = 0.0005, (right). **e**) Flow cytometric analysis and (**f**) quantification of MAIT cells in lung 7 days after infection with indicated bacterial strains, or Δ*RibDH* in combination with 5 nmol 5-OP-RU. n = 5, 4, 8, 8 mice per group, respectively, combined from 2 independent experiments. One-way ANOVA, with Tukey's multiple comparisons test, ****$P$ < 0.0001. (**g**) Representative flow cytometric analysis and quantification of lung MAIT cells 4 weeks post infection with Salmonella strain ELH1301 expressing a synchronized lysis circuit (SLC) and control strain (Ctrl). N = 5, 6, 6 mice per group, respectively, combined from two independent experiments, One-way ANOVA, **$P$ = 0.0082. **h**) Growth kinetics of SLC and control strains in mouse lung, 3 mice per group, one experiment. All error bars represent mean ± S.D. Source numerical data are available in source data.

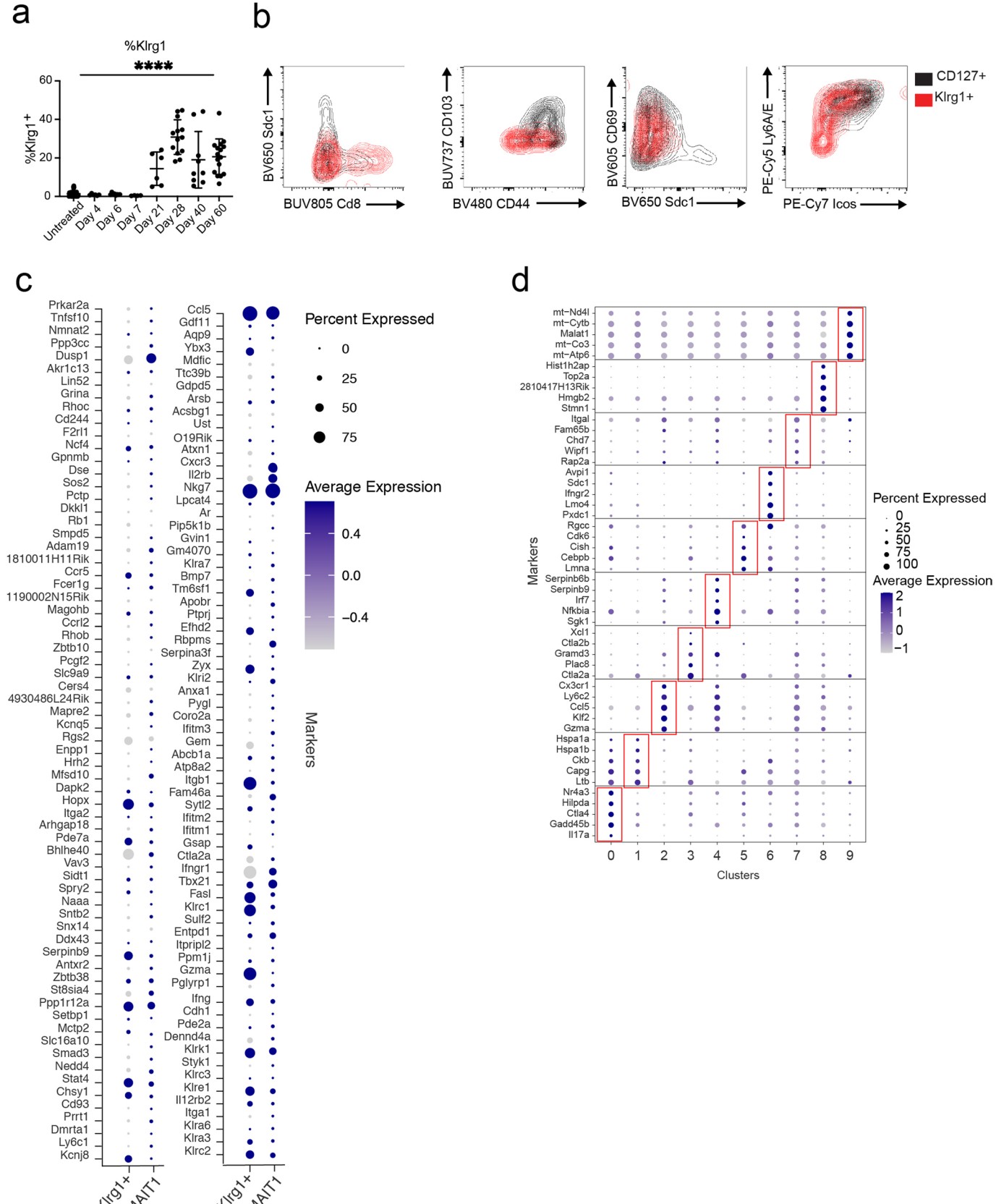

**Extended Data Fig. 2 | mRNA and protein marker phenotypes of aaMAIT cell subsets. a)** % of Klrg1⁺ MAIT cells at indicated timepoints. n = 28, 6, 8, 8, 4, 6, 13, 9, 16, mice per group, respectively, combined from 9 independent experiments. One-way ANOVA, ****$P$ < 0.0001. **b)** Protein expression of the indicated proteins measured by flow cytometry in subsets of day 40 *aa*MAIT cells. **c)** *aa*MAIT cell clusters 2 and 4 with a Th1 gene expression were compared with steady state

MAIT1 cells from publicly available data for expression of MAIT1 signature genes[15,19]. **d)** Top 5 differentially expressed genes of each cluster represented as dot plots, where circle size represents % of cells expressing each gene and color scale depicts relative expression value. Source numerical data are available in source data.

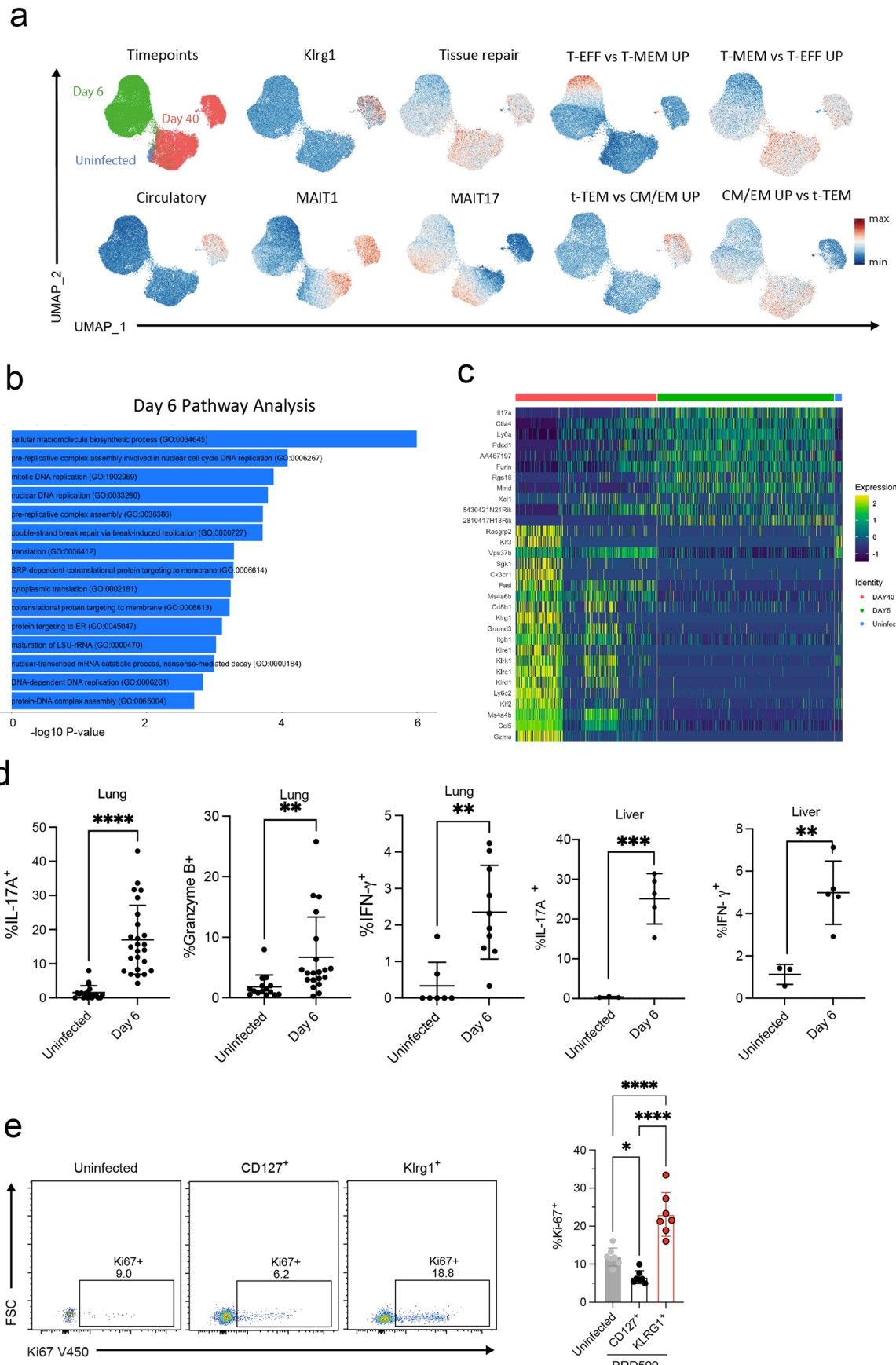

**Extended Data Fig. 3 | See next page for caption.**

**Extended Data Fig. 3 | Day40 *aa*MAIT cell transcriptional and functional phenotype. a**) UMAP representation of day 0 and day 40 scRNA-seq data from Fig. 1c but including day 6 after infection. Expression of key genes and gene signatures as in Fig. 1d. GO terms and C7 datasets: GSE1000002_1582_200_UP, GSE1000002_1582_200_DN and Milner et al.[41] **b**) Ingenuity pathway analysis of day 6 DE genes, ordered by -log10 *P* value. **c**) Heatmap representation of top DE genes across indicated timepoints. **d**) Cytokine production by intracellular flow cytometry at day 6 post BRD509 vaccination by lung and liver MAIT cells, cultured with Brefeldin A for 2 hrs but without restimulation in vitro. Data points indicate individual mice, statistics assessed via unpaired two-tailed t test. IL-17A: n = 19 (uninfected), n = 25 (day 6) mice per group, combined from 5 independent experiments, ****$P$ < 0.0001. Granzyme B: n = 15 (uninfected), n = 20 (day 6), combined from 4 independent experiments, **$P$ = 0.0099. IFN-γ: n = 7 (untreated), n = 10 (day 6), combined from 2 independent experiments, **$P$ = 0.0017. Liver cytokines: n = 3 (untreated), n = 5 (day 6), ***$P$ = 0.0006, **$P$ = 0.0055. **e**) Ki-67 expression in untreated pulmonary MAIT cells and *aa*MAIT cells 40 days post vaccination with BRD509. n = 7 mice per group, combined from 2 independent experiments. Statistical significance via one-way ANOVA with Tukey's multiple comparisons test. *$P$ = 0.0388, ****$P$ < 0.0001. All error bars represent mean ± S.E.M. Source numerical data are available in source data.

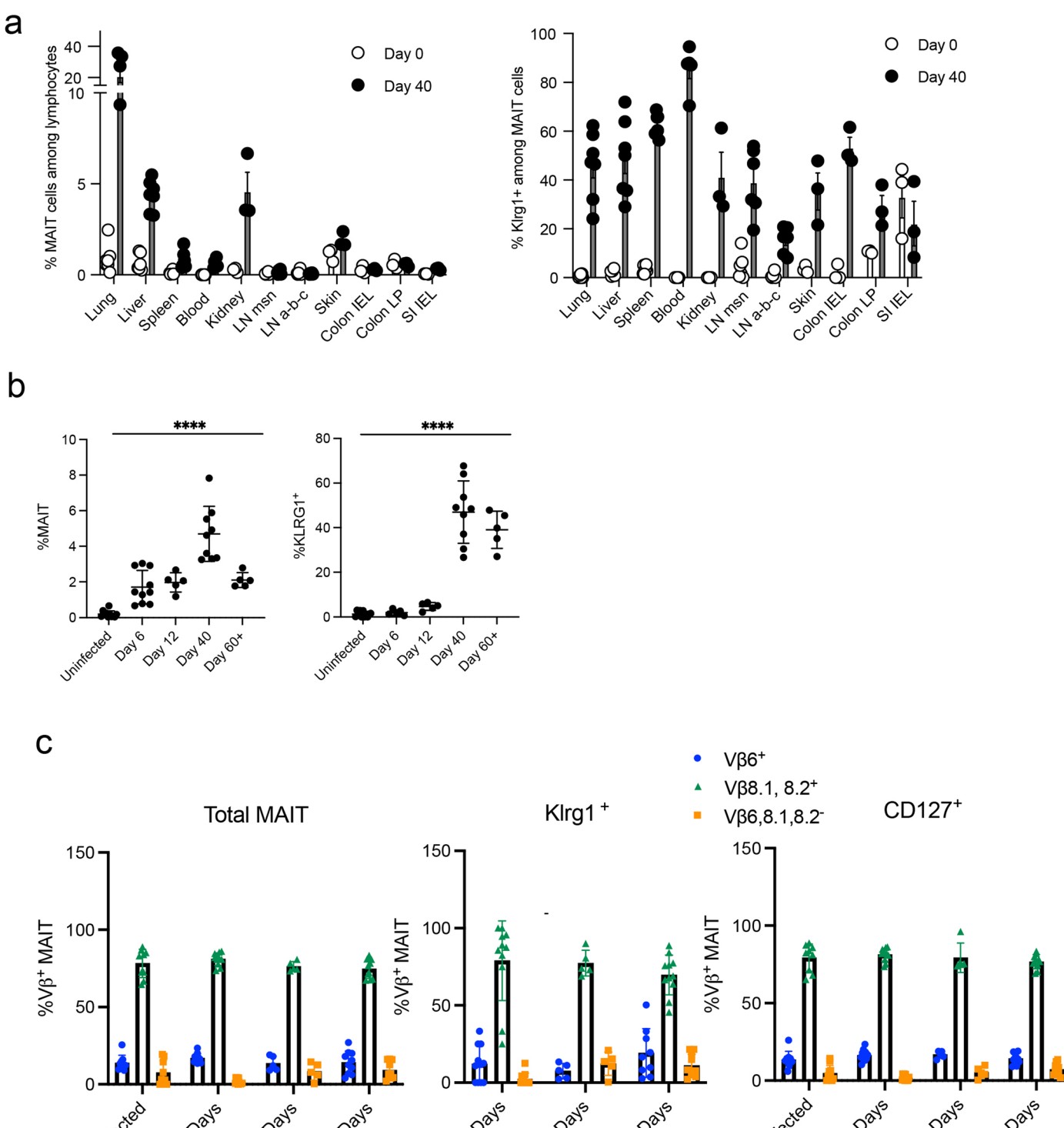

**Extended Data Fig. 4 | aaMAIT cell location, clonotype and expansion kinetics. a**) % of MAIT cells (left) and Klrg1⁺ MAIT cells (right) at day 0 and 40 post BRD509 vaccination in the indicated tissues. **b**) Kinetics of MAIT cell expansion (left) and Klrg1⁺ MAIT cells (right) at indicated timepoints post vaccination in the liver. n = 11, 10, 5, 9, 5 mice per group, respectively, combined from 3 independent experiments; one-way ANOVA ****P < 0.0001 (left). n = 11, 5, 5, 9, 5 mice per group, respectively, combined from 3 independent experiments, one-way ANOVA ****P < 0.0001(right). **c**) Prevalence of Vβ6, Vβ8.1, 8.2, or Vβ6,8.1,8.2⁻ MAIT cells in the lung at steady state (n = 9) and at days 6 (n = 11), 35 (n = 5) and 60 (n = 10) post vaccination in total MAIT cells (left), Klrg1⁺ (center) or CD127⁺ *aa*MAIT cells (right). All data displayed as mean ± S.D. Source numerical data are available in source data.

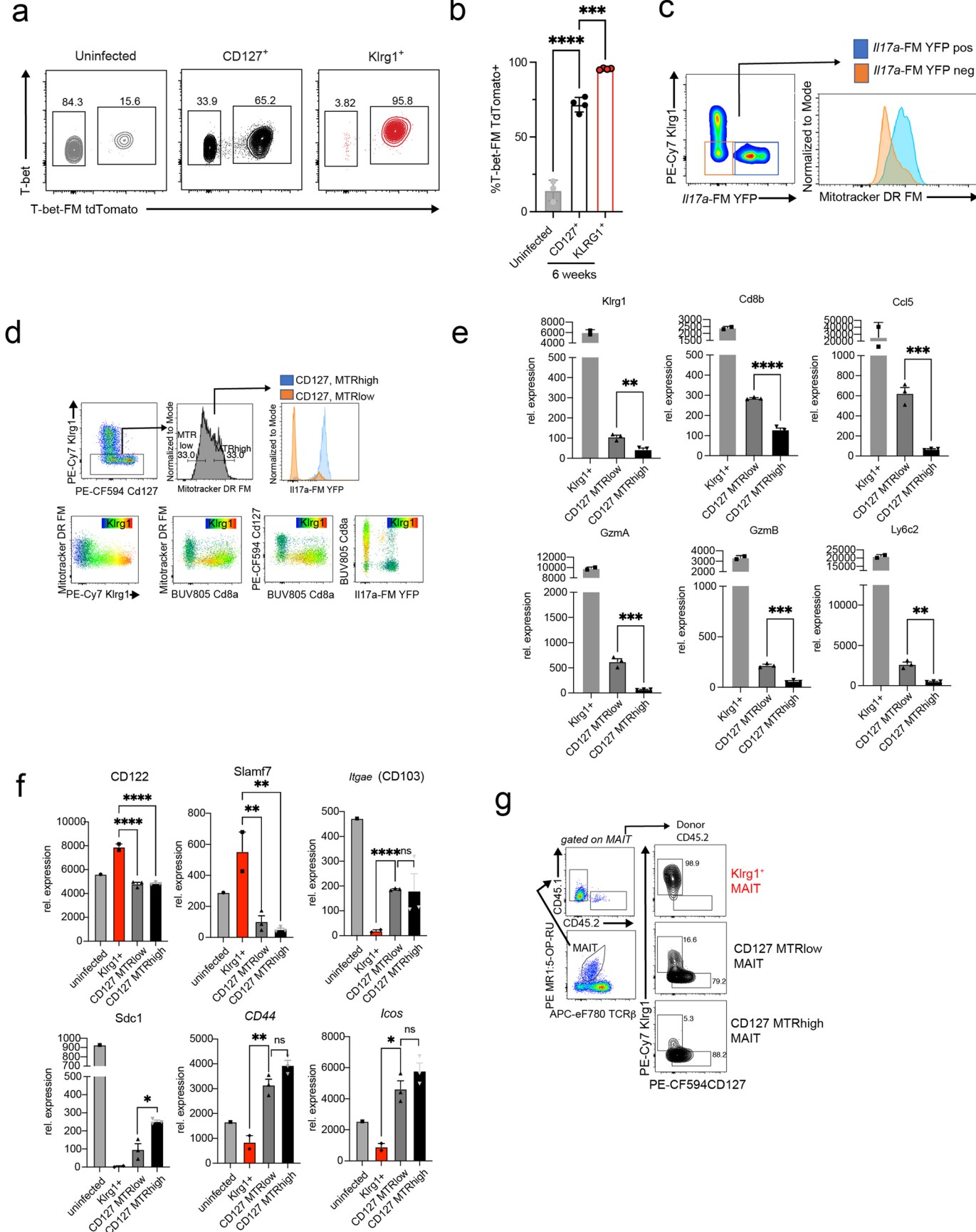

**Extended Data Fig. 5 | See next page for caption.**

**Extended Data Fig. 5 | MTRlow KIrg1⁻ *aa*MAIT cells may be precursors to stable aaMAIT cell subsets. a**,**b**) Representative histograms (**a**) and quantification (**b**) of T-bet-TdTomato fate mapper and current T-bet protein expression in indicated pulmonary MAIT cells subsets at baseline or at >40 days post BRD509 vaccination. Data displayed as mean ± S.D., n = 3, 4, 4 mice per group, respectively, representative of one experiment. Statistical significance via one-way ANOVA with Tukey's multiple comparisons test, ****$P$ < 0.0001, ***$P$ < 0.0002. **c**) *aa*MAIT cells that are Klrg1⁻ from *IL17a*-FM YFP mice were gated as shown at day 40 post vaccination and tested for mitochondrial ΔΨm (MitoTracker DR FM). Data representative of n = 4 mice. **d**) Klrg1⁻ day 40 *aa*MAIT cells were separated into top/bottom 33% based on mitochondrial ΔΨm (MitoTracker DR FM) and tested for *IL17a*-FM YFP expression (top row). Day40 *aa*MAIT cell expression of indicated markers, with expression of Klrg1 overlayed as a color scale. Data representative of n = 6 mice and two independent experiments. **e**) Relative expression measured by bulk RNA-seq of the indicated genes in Day 40 Klrg1⁺ *aa*MAIT cells and CD127⁺, MTRlow and MTRhigh *aa*MAIT cells.

Each datapoint reflects an individual mouse from one experiment. Unpaired two-tailed t tests; Klrg1 **$P$ = 0.006353, Cd8b ****$P$ = 0.000078, Ccl5 ***$P$ = 0.000731, GzmA ***$P$ = 0.000742, GzmB ***$P$ = 0.000332, Ly6c2 **$P$ = 0.001577. Data displayed as mean ± s.e.m. **f**) Relative expression measured by bulk RNA-seq of the indicated genes in naïve MAIT cells, day 40 Klrg1⁺ *aa*MAIT cells and CD127⁺, MTRlow and MTRhigh *aa*MAIT cells. Each datapoint reflects an individual mouse from one experiment. One-way ANOVA CD122 ****$P$ < 0.0001, Slamf7 **$P$ = 0.0047, **$P$ = 0.0021, Itgae ****$P$ < 0.0001, Scd1 *$P$ = 0.009193, Cd44 **$P$ = 0.0079, Icos *$P$ = 0.0127. Data displayed as mean ± s.e.m. **g**) The indicated populations of CD45.2 donor lung *aa*MAIT cells at day 40 post vaccination were sorted and transferred into CD45.1 recipient mice. After gating on MR1:5-OP-RU tetramer⁺ TCRβ⁺ MAIT cells and CD45.2 (=donor) markers (left), Klrg1 and CD127 expression was plotted in recipient mice 14 days post vaccination with BRD509 (right). Representative data shown from one of two independent experiments. Source numerical data are available in source data.

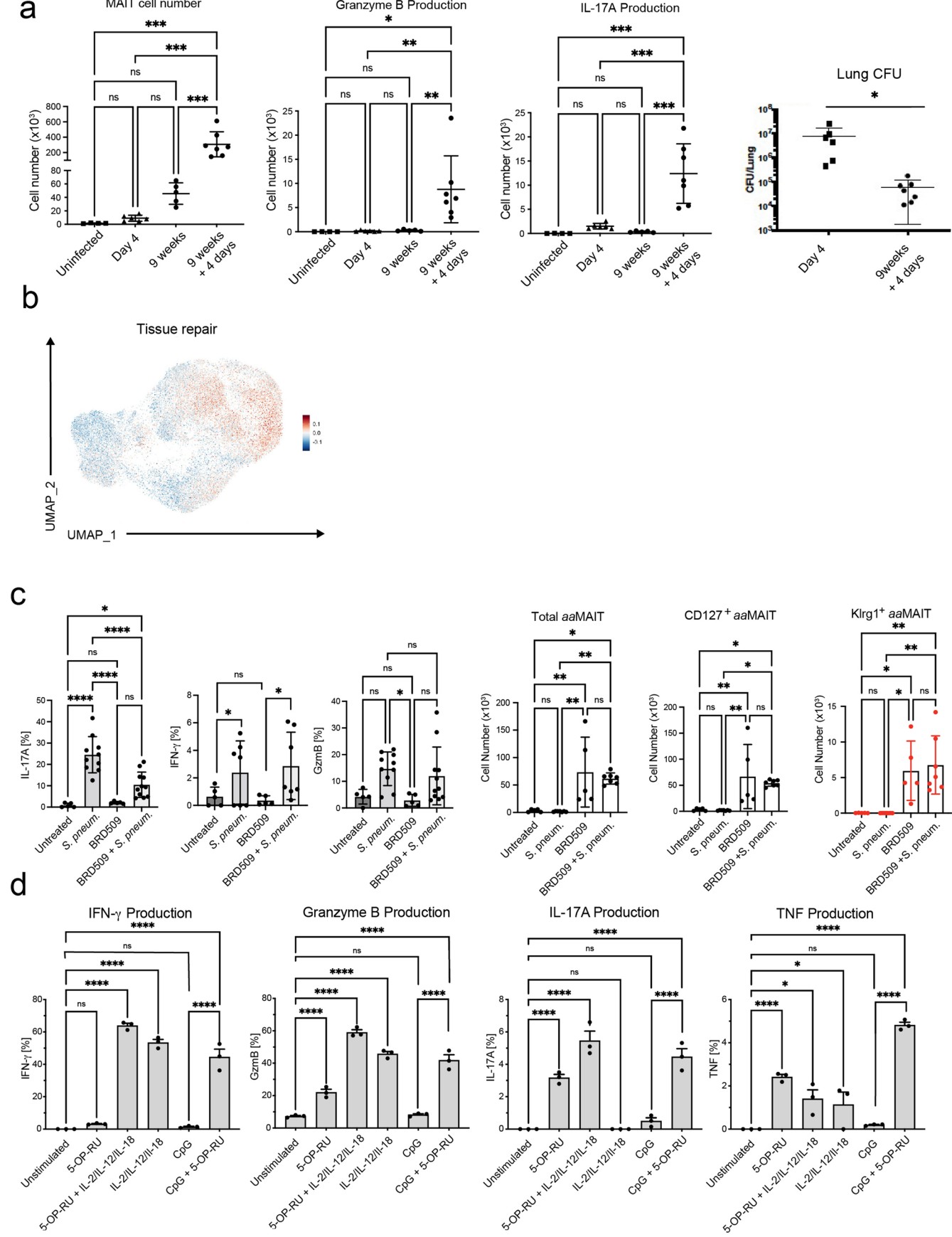

**Extended Data Fig. 6 | See next page for caption.**

**Extended Data Fig. 6 | Re-activated *aa*MAIT cells provide protective responses. a**) Number of total MAIT cells, granzyme B[+] and IL-17A[+] MAIT cells at baseline and at indicated times after BRD509 vaccination or 4 days after re-infection. CFU per lung is shown on the right. Data displayed as mean ± S.D., n = 4, 6, 5, 7 mice per group, one experiment. Statistical significance via one-way ANOVA with Tukey's multiple comparisons test. ***$P = 0.0003$ (uninfected vs 9w + 4d), ***$P = 0.0001$ (4d vs 9w + 4d), ***$P = 0.0009$ (9w vs 9w + 4d) (Left). *$P = 0.0124$, **$P = 0.0055$ (4d vs 9w + 4d), **$P = 0.0092$ (9w vs 9w + 4d) (GrB). ***$P = 0.0002$ (uninfected vs 9w + 4d), ***$P = 0.0002$ (4d vs 9w + 4d), ***$P = 0.0001$ (9w vs 9w + 4d) (IL-17A). *$P = 0.0456$ (right) **b**) Feature plot UMAP representation of a module score for "Tissue repair" derived from the TiRe database **c**) Cytokine production by MAIT cell subsets in untreated and BRD509 vaccinated mice following infection with *S. pneumoniae* URF918, 16 hrs post infection. % of pulmonary MAIT cells positive for IL-17A, IFN-γ and Granzyme B are shown for each group (left). Abundance of indicated populations of MAIT cells quantified as absolute count (right). Data displayed as mean ± S.D. n = 5, 10, 5, 11 mice per group, respectively combined from 3 independent experiments (IL-17A, GzmB). n = 5, 7, 5, 7 mice per group, respectively, combined from 2 independent experiments (IFN-γ). Statistical significance assessed via one-way ANOVA with Tukey's multiple comparisons test, ****$P < 0.0001$,

*$P = 0.0328$ (IL-17A); *$P = 0.0414$ (GrB). Cell counts: n = 5, 7, 5, 7 mice per group, respectively, combined from 2 independent experiments. Statistical significance assessed via one-way ANOVA with Tukey's multiple comparisons test. Total *aa*MAIT: **$P = 0.0054$(Untreated vs BRD509), *$P = 0.0126$(Untreated vs BRD509 + *S.pneum.*), **$P = 0.0022$(*S.pneum.* vs BRD509), **$P = 0.0047$ (BRD509 vs BRD509 + *S.pneum.*). CD127[+] *aa*MAIT: **$P = 0.0087$(Untreated vs BRD509), *$P = 0.0249$ (Untreated vs BRD509 + *S.pneum.*), **$P = 0.0036$(*S.pneum.* vs BRD509), **$P = 0.0101$(*S.pneum.* vs BRD509 + S.pneum.). Klrg1[+] *aa*MAIT: *$P = 0.0207$(Untreated vs BRD509), **$P = 0.0040$ (Untreated vs BRD509 + *S. pneum.*), *$P = 0.0115$(*S.pneum.* vs BRD509), **$P = 0.0017$(*S.pneum.* vs BRD509 + *S. pneum.*). **d**) Cytokine production by pulmonary *aa*MAIT cells isolated 40 days after vaccination with BRD509, cultured as cell suspensions for 18 hrs with or without re-activation as indicated. Cytokine production determined by intracellular staining and plotted as % of MAIT cells positive for IFN-γ, Granzyme B, IL-17A and TNF. n = 3 mice per group, representing 1 of 2 independent experiments. One-way ANOVA with Tukey's test for multiple comparisons; IFNγ: ****$P < 0.0001$, [n.s]$P = 0.8945$. GzmB: [n.s]$P = 0.9994$, ****$P < 0.0001$ IL17: [n.s]$P > 0.9$, ****$P < 0.0001$. TNF: ****$P < 0.0001$. *$P = 0.044$, *$P = 0.034$, [n.s]$P > 0.5$. Data displayed as mean ± s.e.m. Source numerical data are available in source data.

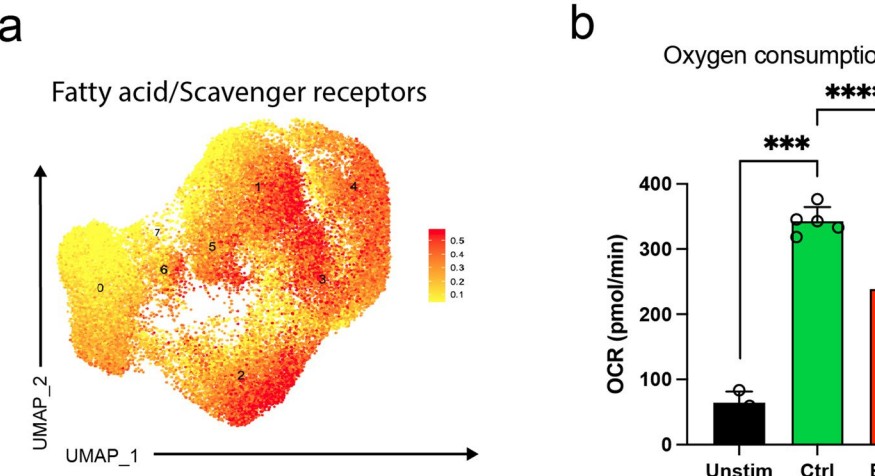

**Extended Data Fig. 7 | CD127⁺ *aa*MAIT cells express transcripts for scavenger receptors. a**) UMAP representation of gene signature analysis for fatty acid/scavenger receptor genes from GO_term 20220131-155727⁻ scavenger_receptors. Analysis of the combined sequence data from unstimulated lung MAIT cells, *aa*MAIT cells, and these two populations 10 hrs after stimulation with *S. pneumoniae*, as shown in Fig. 3. **b**) Effect of etomoxir on the baseline oxygen consumption rate (OCR) as detected by Seahorse Bioanalyzer. Spleen CD8⁺

T cells from mice at day 40 post BRD509 vaccination were analyzed at baseline (Unstim) and when activated (CD3/28) in the presence (ETO) or absence (Ctrl) of etomoxir. Same concentration of ETO as in Fig. 4f–h, showing an on-target effect of the drug. n = 3,5,4 mice per group from one experiment. Two-tailed, unpaired t tests, ***P = 0.00071, ****P < 0.0001. Data displayed as mean ± s.e.m. Source numerical data are available in source data.

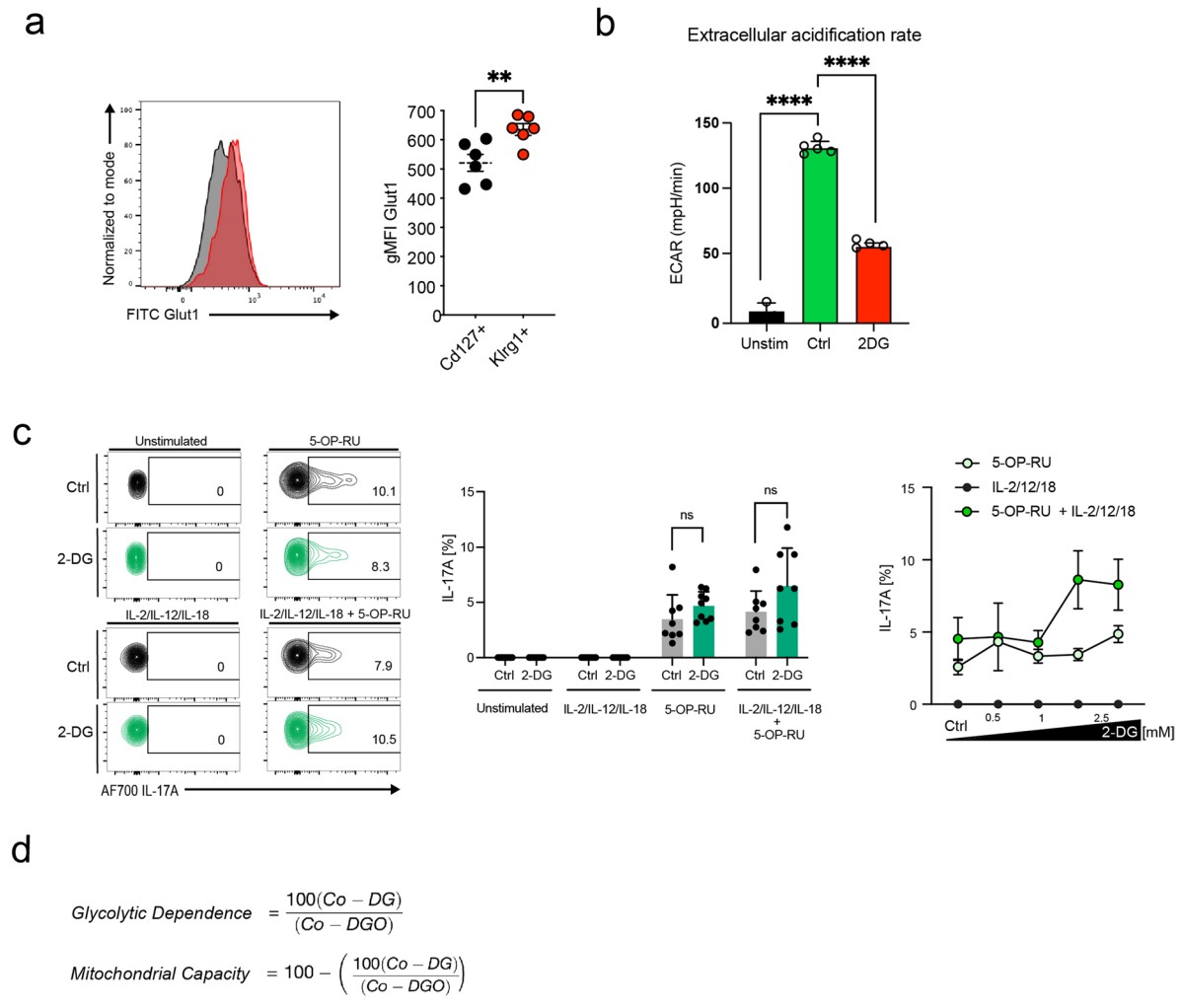

**Extended Data Fig. 8 | MAIT cell IL-17A is not highly dependent on glucose.**
**a)** Pulmonary *aa*MAIT cells were isolated from mice 40 days post BRD509 vaccination and analyzed for Glut1 expression by flow cytometry. Representative histogram (left) and quantification as gMFI (right) in indicated *aa*MAIT cell subsets. n = 6 mice/group combined from two experiments. Unpaired, two tailed t test, **$P$ = 0.00873. **b)** Extracellular acidification rate (ECAR) as detected by the Seahorse Bioanalyzer in spleen CD8[+] T cells from mice at day 40 post BRD509 vaccination at baseline (Unstim) and in activated cells (CD3/28) in presence or absence of 2-Deoxyglucose (2-DG) with the concentration identical to Fig. 5c, showing an on-target effect. n = 3,5,4 mice per group from one experiment. Two-tailed, unpaired t tests, ****$P$ < 0.0001. **c)** Pulmonary *aa*MAIT cells were isolated from mice >40 days post BRD509 vaccination and analyzed for metabolic

and functional parameters during culture as pulmonary cell suspensions for 18 hrs with or without re-activation with 5-OP-RU and/or IL-2, IL-12, IL-18, as indicated. Representative histograms (left) and quantification (middle) of IL-17A production following 18 hrs re-activation. Glycolysis was inhibited by 2-deoxy-glucose (2-DG, green color). Each datapoint reflects an individual mouse, combined from two experiments. One way ANOVA, n.s$P$ = 0.1821, n.s$P$ = 0.1143. Right: Dose response and tests of activation mode dependence, TCR and/or cytokines, of 2-DG effect on IL-17A production with the indicated concentrations of 2-DG during 18 hrs of re-activation. **d)** Formulas applied for calculation of metabolic capacities and dependencies. Co = control, O = with oligomycin, DG = 2-DG, DGO = 2-DG plus oligomycin. All data displayed as mean ± s.e.m. Source numerical data are available in source data.

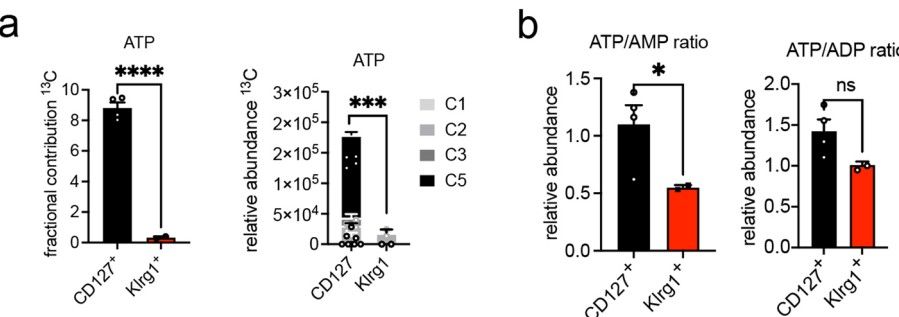

**Extended Data Fig. 9 | Lower ATP in Klrg1$^+$ *aa*MAIT cells. a**) Fractional contribution of $^{13}$C to ATP after labeling for 18 hrs in absence of re-activation was quantified by mass spectrometry in CD127$^+$ and Klrg1$^+$ *aa*MAIT cells (>40 days after vaccination with BRD509). n = 4,2 from one experiment. Left: Unpaired two-tailed t test, ****$P$ = 0.000098. Right: Mixed effect model, ***$P$ = 0.0002.

**b**) Relative abundances of ATP/ADP/AMP quantified by mass spectrometry in CD127$^+$ or Klrg1$^+$ MAIT cells. n = 4,2 from one experiment. Unpaired, two-tailed t-test, *$P$ = 0.0212, $^{n.s}P$ = 0.150. All data displayed as mean ± s.e.m. Source numerical data are available in source data.

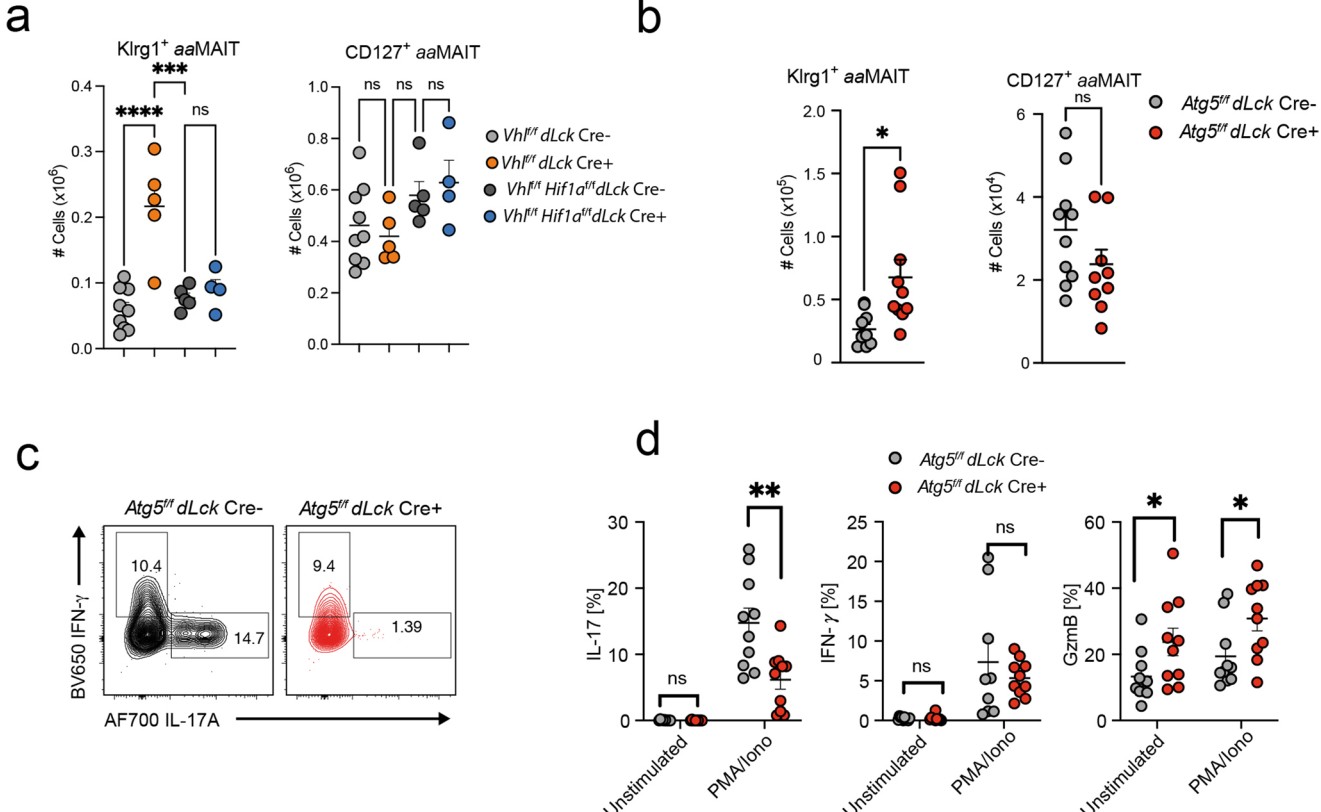

**Extended Data Fig. 10 | Oxygen sensing and autophagy facilitate CD127+**
***aa*MAIT and IL-17 production. a-b**) Absolute numbers of Klrg1+ and CD127+
*aa*MAIT cells in lungs of mice with indicated genotypes at day40 post BRD509
vaccination. Each datapoint reflects an individual mouse from one of two
experiments. **a**) Left: ****$P < 0.0001$, n.s$P = 0.9668$. Right: n.s$P = 0.948$, n.s$P = 0.308$,
n.s$P = 0.953$. One-way ANOVA with Tukey´s multiple comparison testing. **b**) Left:
*$P = 0.0115$, Right: n.s$P = 0.145$. Unpaired two-tailed t tests. **c-d**) Pulmonary *aa*MAIT

cell subsets were isolated from mice and analyzed for cytokine production
by intracellular flow cytometry following PMA/Ionomycin. Representative
histograms and quantification of IL-17A and IFN-γ production. Each datapoint
reflects an individual mouse from one of two experiments. Unpaired two-tailed
t tests. **$P = 0.00435$, *$P = 0.0294$, *$P = 0.0211$. All data displayed as mean ± s.e.m.
Source numerical data are available in source data.

# Reporting Summary

## Statistics

For all statistical analyses, confirm that the following items are present in the figure legend, table legend, main text, or Methods section.

| n/a | Confirmed | |
|---|---|---|
| ☐ | ☒ | The exact sample size (*n*) for each experimental group/condition, given as a discrete number and unit of measurement |
| ☐ | ☒ | A statement on whether measurements were taken from distinct samples or whether the same sample was measured repeatedly |
| ☐ | ☒ | The statistical test(s) used AND whether they are one- or two-sided<br>*Only common tests should be described solely by name; describe more complex techniques in the Methods section.* |
| ☒ | ☐ | A description of all covariates tested |
| ☐ | ☒ | A description of any assumptions or corrections, such as tests of normality and adjustment for multiple comparisons |
| ☐ | ☒ | A full description of the statistical parameters including central tendency (e.g. means) or other basic estimates (e.g. regression coefficient) AND variation (e.g. standard deviation) or associated estimates of uncertainty (e.g. confidence intervals) |
| ☐ | ☒ | For null hypothesis testing, the test statistic (e.g. *F*, *t*, *r*) with confidence intervals, effect sizes, degrees of freedom and *P* value noted<br>*Give P values as exact values whenever suitable.* |
| ☒ | ☐ | For Bayesian analysis, information on the choice of priors and Markov chain Monte Carlo settings |
| ☒ | ☐ | For hierarchical and complex designs, identification of the appropriate level for tests and full reporting of outcomes |
| ☒ | ☐ | Estimates of effect sizes (e.g. Cohen's *d*, Pearson's *r*), indicating how they were calculated |

*Our web collection on statistics for biologists contains articles on many of the points above.*

## Software and code

Policy information about availability of computer code

| Data collection | FACS DIVA 8.0 (BD Biosciences) was used for all Flow Cytometry data acquisition. Zeiss Zen Blue v3 was used for confocal image data aquisition. BCL (BinaryBaseCall) were used as input to create FASTQ files with cellranger mkfastq v 6.1.2 |
|---|---|
| Data analysis | FlowJo v 10.8.1 (TreeStar Inc) was used for flow cytometry data analysis. Excel v16 and Graphpad Prism v9.0 were used for statistical tests, data analysis and graph plotting. Automated image quantification: Imaris 10 (Bitplane), using the spot detection algorithm. scRNA-seq data: FastQ file alignment: refdata-gex-mm10-2020-A with cellranger count v 6.1.2. Analysis via Seurat v 4.1.1 and Seurat's FindAllMarkers function and Seurat's AddModuleScore function. For Mass spectometry analysis, mass isotopologue distributions and fractional contributions of metabolites were quantified using TraceFinder 3.3. |

For manuscripts utilizing custom algorithms or software that are central to the research but not yet described in published literature, software must be made available to editors and reviewers. We strongly encourage code deposition in a community repository (e.g. GitHub). See the Nature Portfolio guidelines for submitting code & software for further information.

## Data

Policy information about availability of data

All manuscripts must include a data availability statement. This statement should provide the following information, where applicable:
- Accession codes, unique identifiers, or web links for publicly available datasets
- A description of any restrictions on data availability
- For clinical datasets or third party data, please ensure that the statement adheres to our policy

Bulk and sc-RNA-seq data that support the findings of this study have been deposited in the Gene Expression Omnibus (GEO) under accession code GSE226524. Publicly available RNA-seq data was used from the C7 immunological database: Accession codes GSE1000002_1582_200_UP, GSE1000002_1582_200_DN, 20220131-155727-scavenger_receptors

## Human research participants

Policy information about studies involving human research participants and Sex and Gender in Research.

| | |
|---|---|
| Reporting on sex and gender | N/A |
| Population characteristics | N/A |
| Recruitment | N/A |
| Ethics oversight | N/A |

Note that full information on the approval of the study protocol must also be provided in the manuscript.

# Field-specific reporting

Please select the one below that is the best fit for your research. If you are not sure, read the appropriate sections before making your selection.

☒ Life sciences         ☐ Behavioural & social sciences         ☐ Ecological, evolutionary & environmental sciences

For a reference copy of the document with all sections, see nature.com/documents/nr-reporting-summary-flat.pdf

# Life sciences study design

All studies must disclose on these points even when the disclosure is negative.

| | |
|---|---|
| Sample size | No sample size calculations were performed. Sample size was determined based on the magnitude and variation of measurable differences between groups, as well as the feasibility of performing highly technical experiments with rare cell populations. Sample sizes are indicated in figure legends. All data points represent individual mice and mice from genetically distinct comparison groups (e.g. cre+ vs cre-) are littermates from the same breeders unless indicated otherwise. This limits sample size, in addition to ethical considerations when determining necessary power of animal experiments. |
| Data exclusions | Post-sequencing, stringent, pre-established quality controls were applied: For mouse single cell RNA sequencing analysis, cells with less than 200 genes and more than 2,500 genes were discarded. Cells with more than 5% UMIs coming from mitochondrial genes were filtered out. Genes expressed in less than 3 cells were ignored. No other data was excluded in any studies. |
| Replication | All experiments reported here with the exception of scRNAseq experiment (n=1) have been repeated independently 2 or more times, and exact number of individual experiments are indicated in legends. In many instances data from individual experiments have been pooled in the representations, and this is indicated in legends. All experiments were reproducible. |
| Randomization | Randomization was not applied. Mice were gender and aged matched and breedings were setup so that every experimental group is represented in each litter. |
| Blinding | The investigators were not blinded during data collection. In most cases, data was quantified in an unsupervised manner/by data analysis software. In many cases (scRNA-seq, imaging) samples were processed and analyzed by separate scientists via automated algorythms and bioinformatic analysis was done by separate bioinformaticians. |

# Reporting for specific materials, systems and methods

We require information from authors about some types of materials, experimental systems and methods used in many studies. Here, indicate whether each material, system or method listed is relevant to your study. If you are not sure if a list item applies to your research, read the appropriate section before selecting a response.

## Materials & experimental systems

| n/a | Involved in the study |
|-----|----------------------|
| ☐ | ☒ Antibodies |
| ☒ | ☐ Eukaryotic cell lines |
| ☒ | ☐ Palaeontology and archaeology |
| ☐ | ☒ Animals and other organisms |
| ☒ | ☐ Clinical data |
| ☒ | ☐ Dual use research of concern |

## Methods

| n/a | Involved in the study |
|-----|----------------------|
| ☒ | ☐ ChIP-seq |
| ☐ | ☒ Flow cytometry |
| ☒ | ☐ MRI-based neuroimaging |

# Antibodies

**Antibodies used**

MR1 tetramers loaded with either 5-OP-RU or 6-FP were obtained from the NIH Tetramer Facility. Antibodies with clone, dilution, catalogue number and supplier indicated in parentheses: anti-mouse CD16/32 (2.4G2, Cat#553142, 1:500, BD Bioscience), anti-mouse Sdc1 (281-2, 1:200, Cat# 564068 BDBioscience), anti-mouse CD36 (HM36, 1:200, Cat#102608, Biolegend), anti-mouse CD45 (30-F11, 1:800, Cat#564225 BDBioscience); anti-mouse IgD (clone 11-26c.2a, 1:200, Cat#564273 BDBioscience); anti-mouse Klrg1 (clone 2F1/KLRG1, 1:200, 25-5893-82 Invitrogen); anti-mouse gdTCR (clone GL3, 1:200, Cat#553178 BDBioscience); anti-mouse TER-119 (TER-119, 1:200, Cat#116204 Biolegend) anti-mouse CD127 (clone SB/199, 1:200, Cat#562419 BD Bioscience); anti-mouse CD8a (clone 53-6.7, 1:400, Cat#47-0081-82 eBioscience); anti-mouse CD8b (H35-17.2, 1:400, Cat#741992 BDBioscience), anti-mouse EpCAM (clone G8.8, 1:200, Cat#13-5791 eBioscience); anti-mouse IFN-g(clone XMG1.2, 1:400, Cat# 505831 Biolegend); anti-mouse TNF (clone MP6-XT22, 1:400); anti-mouse IL-17A (clone TC11-18H10, 1:400, Cat# 506914 Biolegend); anti-mouse CD69 (clone H1.2F3, 1:200, Cat# 104530 Biolegend); anti-mouse CD44 (IM7, 1:200, Cat# 586116 BDBioscience), anti-mouse Ly6A/E (D7, 1:200, Cat# 108110 Biolegend), anti-mouse Icos (C398.4a, 1:200, Cat# 313520 Biolegend), anti-mouse T-bet (clone O4-46, 1:100 Cat# 562467 BDBioscience); anti-mouse RORgT (clone B2D, 1:100, Cat# 46-6981-82 eBioscience); anti-mouse Ly6G (clone 1A8, 1:400, Cat# 127639 Biolegend); anti-mouse CD11b (M1/70, 1:400, Cat# 561114 BDBioscience); anti-mouse CD45R/B220 (RA3-6B2, 1:200 Cat# 552771 BDBioscience); anti-mouse CD11c (N418, 1:200, Cat# 48-0114-80 eBioscience); anti-mouse TCRb(H57-597, 1:200, Cat# 47-5961-82 eBioscience), anti-mouse Ki67 (B56, 1:200, Cat# 561281 BDBioscience); anti-mouse CD80 (16-10A1, 1:200, Cat# 553768 BDBioscience); anti-mouse CD86 (GL1, 1:200); anti-mouse MR1 (26.5, 1:200, Cat# 361107 Biolegend); anti-mouse Granzyme B (GB11, 1:400, Cat# GRB05 Invitrogen); anti-mouse CD103 (2e7, 1:200, Cat# 749393, BDBioscience); anti-mouse siglec F (E50-2440 1:200, Cat# 565527 BDBioscience), anti-mouse Tom20 antibody (D8T4N, 1:400, Cat#13929, Cell Signaling).

**Validation**

1. anti-mouse CD16/32 antibody: The manufacturer states that specificity has been validated for Flow Cytometry: https://www.bdbiosciences.com/en-us/products/reagents/flow-cytometry-reagents/research-reagents/single-color-antibodies-ruo/purified-rat-anti-mouse-cd16-cd32-mouse-bd-fc-block.553142
2. anti-mouse Sdc1 antibody: The manufacturer states that specificity has been validated for Flow Cytometry:https://www.bdbiosciences.com/en-us/products/reagents/flow-cytometry-reagents/research-reagents/single-color-antibodies-ruo/bv650-rat-anti-mouse-cd138.564068
3. anti-mouse CD36 antibody: https://www.biolegend.com/en-us/products/alexa-fluor-488-anti-mouse-cd36-antibody-3311?GroupID=BLG10704
4. anti-mouse CD45 antibody: https://www.bdbiosciences.com/en-us/products/reagents/flow-cytometry-reagents/research-reagents/single-color-antibodies-ruo/bv786-rat-anti-mouse-cd45.564225
5. anti-mouse IgD antibody: https://www.bdbiosciences.com/en-sg/products/reagents/flow-cytometry-reagents/research-reagents/single-color-antibodies-ruo/percp-cy-5-5-rat-anti-mouse-igd.564273
6. anti-mouse Klrg1 antibody: https://www.thermofisher.com/antibody/product/KLRG1-Antibody-clone-2F1-Monoclonal/25-5893-82
7. anti-mouse gdTCR antibody: https://www.bdbiosciences.com/en-us/products/reagents/flow-cytometry-reagents/research-reagents/single-color-antibodies-ruo/pe-hamster-anti-mouse-t-cell-receptor.561997
8. anti-mouse TER-119 antibody: https://www.biolegend.com/en-us/products/biotin-anti-mouse-ter-119-erythroid-cells-antibody-1864
9. anti-mouse CD127 antibody: https://www.bdbiosciences.com/en-ca/products/reagents/flow-cytometry-reagents/research-reagents/single-color-antibodies-ruo/pe-cf594-rat-anti-mouse-cd127.562419
10. anti-mouse CD8a antibody: https://www.thermofisher.com/antibody/product/CD8a-Antibody-clone-53-6-7-Monoclonal/47-0081-82
11. anti-mouse CD8b antibody: https://www.bdbiosciences.com/en-at/products/reagents/flow-cytometry-reagents/research-reagents/single-color-antibodies-ruo/BUV805-Rat-Anti-Mouse-CD8b.741992
12. anti-mouse EpCAM antibody: https://www.thermofisher.com/antibody/product/CD326-EpCAM-Antibody-clone-G8-8-Monoclonal/13-5791-82
13. anti-mouse Ifng antibody: The manufacturer states that specificity has been validated for Flow Cytometry:https://www.biolegend.com/fr-fr/products/brilliant-violet-650-anti-mouse-ifn-gamma-antibody-7681
14. anti-mouse Tnf antibody
15. anti-mouse IL-17A antibody: The manufacturer states that specificity has been validated for Flow Cytometry:https://www.biolegend.com/en-us/products/alexa-fluor-700-anti-mouse-il-17a-antibody-3539?GroupID=GROUP24
16. anti-mouse Cd69 antibody: The manufacturer states that specificity has been validated for Flow Cytometry:https://www.biolegend.com/en-us/products/brilliant-violet-605-anti-mouse-cd69-antibody-7864?GroupID=BLG10515
17. anti-mouse Cd44 antibody: The manufacturer states that specificity has been validated for Flow Cytometry:https://www.bdbiosciences.com/en-us/products/reagents/flow-cytometry-reagents/research-reagents/single-color-antibodies-ruo/pe-rat-anti-mouse-cd44.561860
18. anti-mouse Ly6A/E antibody: The manufacturer states that specificity has been validated for Flow Cytometry:https://www.biolegend.com/en-us/products/pe-cyanine5-anti-mouse-ly-6a-e-sca-1-antibody-229?GroupID=BLG2524

19. anti-mouse Icos antibody: The manufacturer states that specificity has been validated for Flow Cytometry:https://www.biolegend.com/fr-fr/products/pe-cyanine7-anti-human-mouse-rat-cd278-icos-antibody-6908

20. anti-mouse T-bet antibody: The manufacturer states that specificity has been validated for Flow Cytometry:https://www.bdbiosciences.com/en-us/products/reagents/flow-cytometry-reagents/research-reagents/single-color-antibodies-ruo/pe-cf594-mouse-anti-t-bet.562467

21. anti-mouse Rorgt antibody: The manufacturer states that specificity has been validated for Flow Cytometry:https://www.thermofisher.com/antibody/product/ROR-gamma-t-Antibody-clone-B2D-Monoclonal/46-6981-82

22. anti-mouse Ly6G antibody: The manufacturer states that specificity has been validated for Flow Cytometry: https://www.biolegend.com/en-us/products/brilliant-violet-605-anti-mouse-ly-6g-antibody-12244?GroupID=BLG7234

23. anti-mouse Cd11b antibody: The manufacturer states that specificity has been validated for Flow Cytometry:https://www.bdbiosciences.com/en-us/products/reagents/flow-cytometry-reagents/research-reagents/single-color-antibodies-ruo/percp-cy-5-5-rat-anti-cd11b.561114

24. anti-mouse Cd45/B220 antibody: The manufacturer states that specificity has been validated for Flow Cytometry:https://www.bdbiosciences.com/en-us/products/reagents/flow-cytometry-reagents/research-reagents/single-color-antibodies-ruo/percp-cy-5-5-rat-anti-mouse-cd45r-b220.552771

25. anti-mouse Cd11c antibody: The manufacturer states that specificity has been validated for Flow Cytometry:https://www.thermofisher.com/antibody/product/42-0114-82.html?ef_id=CjwKCAjw5pShBhB_EiwAvmnNV0miRHViUP_dgGczGVPovK4rtZDNtuJVskFrnG1WY9useSwt-INpYRoCs80QAvD_BwE:G:s&s_kwcid=AL!3652!3!459737518508!!!g!!!10950825775!106531320406&cid=bid_pca_aup_r01_co_cp1359_pjt0000_bid00000_0se_gaw_dy_pur_con&gclid=CjwKCAjw5pShBhB_EiwAvmnNV0miRHViUP_dgGczGVPovK4rtZDNtuJVskFrnG1WY9useSwt-INpYRoCs80QAvD_BwE

26. anti-mouse TCRb antibody: The manufacturer states that specificity has been validated for Flow Cytometry:https://www.thermofisher.com/antibody/product/TCR-beta-Antibody-clone-H57-597-Monoclonal/47-5961-82

27. anti-mouse Ki67 antibody: The manufacturer states that specificity has been validated for Flow Cytometry:https://www.bdbiosciences.com/en-us/products/reagents/flow-cytometry-reagents/research-reagents/single-color-antibodies-ruo/v450-mouse-anti-ki-67.561281

28. anti-mouse Cd80 antibody: The manufacturer states that specificity has been validated for Flow Cytometry:https://www.bdbiosciences.com/en-us/products/reagents/flow-cytometry-reagents/research-reagents/single-color-antibodies-ruo/fitc-hamster-anti-mouse-cd80.553768

29. anti-mouse Cd86 antibody: The manufacturer states that specificity has been validated for Flow Cytometry:https://www.biolegend.com/en-us/products/pe-anti-mouse-cd86-antibody-256?GroupID=BLG10719

30. anti-mouse Mr1 antibody: The manufacturer states that specificity has been validated for Flow Cytometry:https://www.biolegend.com/fr-ch/products/apc-anti-human-mouse-rat-mr1-antibody-9373

31. anti-mouse GzmB antibody: The manufacturer states that specificity has been validated for Flow Cytometry:https://www.thermofisher.com/antibody/product/Granzyme-B-Antibody-clone-GB11-Monoclonal/GRB05

32. anti-mouse Cd103 antibody: The manufacturer states that specificity has been validated for Flow Cytometry:https://www.bdbiosciences.com/en-nz/products/reagents/flow-cytometry-reagents/research-reagents/single-color-antibodies-ruo/buv737-hamster-anti-mouse-cd103.749393

32. anti-mouse SiglecF antibody: The manufacturer states that specificity has been validated for Flow Cytometry:https://www.bdbiosciences.com/en-us/products/reagents/flow-cytometry-reagents/research-reagents/single-color-antibodies-ruo/apc-cy-7-rat-anti-mouse-siglec-f.565527

33. anti-mouse Tom20 antibody: The manufacturer states that specificity has been validated for Immunofluorescence:https://www.cellsignal.com/products/primary-antibodies/tom20-antibody/13929?Ntk=Products&Ntt=13929&gclid=CjwKCAjw5pShBhB_EiwAvmnNV9SFjaRKitrlHkhZLY0MERMoDN-6nc_MfziMG9MGlYpQ5CqqQK8oAhoCuz0QAvD_BwE&gclsrc=aw.ds

# Animals and other research organisms

Policy information about studies involving animals; ARRIVE guidelines recommended for reporting animal research, and Sex and Gender in Research

| | |
|---|---|
| Laboratory animals | C57BL/6J mice, Atg5f/f, Vhlf/f, Hif1af/f and dLck-Cre mice were purchased from Jackson Laboratory and crossed to generate Atg5f/f dLck-Cre, Vhlf/f dLck-Cre and Vhlf/f Hif1af/f dLck-Cre mice. Heterozygous Cre mice were used, together with littermate controls, in all experiments. All mice were on the C57BL/6J background and used at 6-12 weeks of age. B6;CBA-Tg(Tbx21-cre)1Dlc/J (Tbet-cre) were bred with B6.Cg-Gt(ROSA)26Sortm14(CAG-tdTomato)Hze/J (Td-tomato) mice (from Jackson Laboratories) to generate the T-bet fate mapping line. B6.129(SJL)-Il17atm1.1(icre)Stck/J (IL-17A-YFP fate-map) mice were purchased from Jackson Laboratory. ¬ Ja18-/-(Traj18-/-) mice were generated as described previously (see methods section). Mr1-/- mice were kindly provided by Dr. Gilfillan (Washington University, St. Louis)(see methods for detail). C57BL/6-TgCAG-RFP/EGFP/Map1lc3b transgenic mice were from Jackson Laboratory. Mice were group-housed under a standard rodent chow (Pico Lab Diet 20, #5053) at ambient temperature (68°F), 50% humidity in average and a 12 hour light-dark cycle in individually ventilated cages. Food and water were provided ad libitum. |
| Wild animals | This study did not include wild animals |
| Reporting on sex | Both male and female mice were used in these studies. No obvious sex based difference was evident in these experiments, but the data have not been specifically analyzed after disaggregation for sex. |
| Field-collected samples | No field collected samples were used in this study |
| Ethics oversight | All procedures were approved by the La Jolla Institute for Immunology or University of California San Diego Animal Care and Use Committee and are compliant with the ARRIVE standards. |

Note that full information on the approval of the study protocol must also be provided in the manuscript.

# Flow Cytometry

## Plots

Confirm that:

☒ The axis labels state the marker and fluorochrome used (e.g. CD4-FITC).

☒ The axis scales are clearly visible. Include numbers along axes only for bottom left plot of group (a 'group' is an analysis of identical markers).

☒ All plots are contour plots with outliers or pseudocolor plots.

☒ A numerical value for number of cells or percentage (with statistics) is provided.

## Methodology

| | |
|---|---|
| Sample preparation | Lung tissue was digested with spleen dissociation medium (Stemcell), and mechanically dissociated using a GentleMacs Dissociator (Miltenyi). Cells were then strained though a 70 μm filter and washed with HBSS supplemented with 10% fetal bovine serum (FBS) followed by RBC lysis. For adoptive transfer experiments, Penicillin-Streptomycin (Gibco) was added to media throughout the experiment. For staining of cell surface molecules, cells were suspended in staining buffer (PBS, 1% bovine serum albumin (BSA), and 0.01% NaN3) and stained using PE- or APC-conjugated MR1 tetramers at a dilution of 1:300 in staining buffer for 30 minutes at room temperature followed by staining with fluorochrome-conjugated antibody at 0.1–1 μg/107 cells. Cells were stained with Live/Dead Yellow (ThermoFisher) at 1:500 and blocked with 2.4G2 antibody at 1:500 and Free Streptavidin at 1:1000 for 15 min at 4℃, continued with surface antibody staining for 30 min at 4℃. For cytokine staining, cells were previously stimulated with 50 ng/ml of PMA and 1Ωg/ml of Ionomycin for 2h at 37℃ and then incubated in GolgiStop and GolgiPlug (both from BD Pharmingen) for 2 h at 37℃. For in vitro re-activation experiments, cultures were carried out for 18 hours without stimulation or with 5-OP-RU (1μM) and/or IL-2+IL-12+IL-18 (10ng/ml each). GolgiStop and GolgiPlug (both from BD PharMingen) were added for the last 2 hours. Following S. pneumoniae infection, cells were incubated in GolgiStop and GolgiPlug for 2 h at 37℃ with no restimulation. For intracellular staining, cells were fixed with CytoFix (BD) for 20 min, and permeabilized with Perm 1X solution (Thermo Fisher) overnight with antibodies for intracellular marker detection. |
| Instrument | Data were acquired on Fortessa and LSR-II cytometers and analyzed with FlowJo v10.8 (BD). Sorting was carried out on FACSAria(TM) Fusion and FACS Aria-3(TM) Cytometers (BD Biosciences) |
| Software | FACS DIVA 8.0 (BD Biosciences) was used for all Flow Cytometry data aquisition. Flowjo v10.8.1 was used for data analysis |
| Cell population abundance | Sorting efficiency was routinely controlled during sorting and didn't drop below 90%. |
| Gating strategy | Gating strategies are displayed in Extended Data Figure 1 +2 |

☒ Tick this box to confirm that a figure exemplifying the gating strategy is provided in the Supplementary Information.

