## [Peer Review File · Nature Cell Biology]

Peer Review Information

Journal: Nature Cell Biology

Manuscript Title: Divergent metabolic programs control two populations of MAIT cells that protect the lung

Corresponding author name(s): Dr Mitchell Kronenberg

Editorial Notes:

Reviewer Comments & Decisions:

Decision Letter, initial version:

*Please delete the link to your author homepage if you wish to forward this email to co-authors.

Dear Dr Kronenberg,

Thank you for submitting your manuscript, "Divergent metabolic programs control two populations of MAIT cells that protect the lung", to Nature Cell Biology, and thank you very much for your patience with the peer review process. The manuscript has now been seen by 3 referees, who are experts in T cell immunometabolism (referee 1); MAIT cells (referee 2); and immunometabolism (referee 3). As you will see from their comments (attached below), they found this work of potential interest but have raised substantial concerns, which in our view would need to be addressed with considerable revisions

before we can consider publication in Nature Cell Biology.

As per our standard editorial process, we have now discussed the referee reports in detail within the editorial team, including the chief editor, to identify key referee points that should be addressed with priority, as opposed to requests that are overruled as being beyond the scope of the current study. To guide the scope of the revisions, I have listed these points below. Our typical revision period is six months; we are committed to providing a fair and constructive peer-review process, so please feel free to contact me if you would like to discuss any of the referee comments further or if you anticipate any issues/delays addressing the reviews.

I should stress that the referees' concerns are significant and would need to be addressed with experiments and data, and that reconsideration of the study for this journal and re-engagement of the referees would depend on the strength of these revisions. In particular, it would be essential to:

1- Strengthen the conclusions about the metabolic dependencies of the cell subsets:
Rev#3 pt #1; Rev#1 pt #5

2- the differential protections conferred by the two populations were also questioned by the reviewers, which should be addressed:
Rev#1 point #3
Rev#3 point #2

3- strengthen the analyses of the two lineages, addressing whether they are two distinct lineages or if there is plasticity, and deepen the characterizations of the populations, including in comparison with past work:

Rev#1 pt #1

Rev#2 "Fate mapping experiments here indicated that CD127+ aaMAIT cells are a separate lineage that does not give rise to KLRG1+ aaMAIT – which are absent in the lung at steady state, instead appears to be a subset with a more circulatory signature. Does this indicate that KLRG1+ aaMAIT cells are recruited from other sites and subsequently expand to BRD509 challenge? This possible recruitment route should be explored

Several studies have found that MAIT1/17 cell lineages arise early in development and have provided markers to define these mutually exclusive subset, that are similarly defined by RORgt and Tbet. These protein signatures are not described here, did the authors clarify that aaMAIT cells that arise also express these subset-specific markers, for example, expression of CD122, CD319, ICOS? CD127+ aaMAIT cells appear to expand via TCR/antigenic cues and protect via an antigen/MR1-dependent manner while KLRG1+ aaMAIT cells respond to cytokine cues and protect via non TCR-mechanisms such as cytokines, did the authors explore the relative clonality of aaMAIT to further support this concept?

As this builds upon the BRD509 vaccination model for MAIT cell expansion described in ref 16 (Chen et al, 2016 Mucosal Imm), there appears to be a difference in the transcriptional signatures of the expanded cells. This paper shows that expanded lung MAIT cells acquire a RORgt+ Tbet+ double positive signature, and this phenotype persists even >100 days. In contrast, the authors here see mutually exclusive KLRG1+ Tbet+ or CD127+ RORgt+ cells; is there an explanation for this discrepancy?"

Rev#3 point #3

4- the reviewers asked about the potential for these MAIT subsets to be found outside of the lung and whether similar subsets exist in humans. We agree with the referees that addressing these points would enhance the impact of the study. While analyses along these lines would not be strictly required for reconsideration at the journal, we would welcome such data:

Rev#1 point #4

Rev#2 "The lung was the only tissue examined in this study. Though it makes sense to focus on the lung with the bacterial infection models used, it would have been very interesting to look in other tissues in case the MAIT cells that differentiate in infected lung can migrate elsewhere. This is particularly important given the associations with resident memory cells that are drawn in this manuscript. If aaMAIT cells can leave the lung they may be heavily influenced by other environments. Eg those that home to liver would encounter very different lipid and glucose environment that may influence their ongoing differentiation. Similarly, it would have been interesting to examine whether the aaMAIT cells studied here are present in lymph nodes and spleen. Also, do similar populations develop in gut mucosa or skin?

A very important question is how do the key findings in this study relate to human MAIT cell biology. Do humans have similar populations or aaMAIT cells? Human MAIT cells are known to be phenotypically and functionally heterogeneous – and are more likely to include antigen-experienced populations. Can populations of human MAIT cells that resemble KLRG1+ or CD127+ aaMAIT cells be detected in human blood or other tissue? This would significantly enhance the impact of this study for the target journal."

5- All other referee concerns pertaining to strengthening existing data, providing controls, methodological details, clarifications and textual changes, should also be addressed.

6- Finally, please pay close attention to our guidelines on statistical and methodological reporting (listed below) as failure to do so may delay the reconsideration of the revised manuscript. In particular, at resubmission, please provide:

We would be happy to consider a revised manuscript that would satisfactorily address these points, unless a similar paper is published elsewhere, or is accepted for publication in Nature Cell Biology in the meantime.

- ensure that it conforms to our format instructions and publication policies (see below and

www.nature.com/nature/authors/).

- provide a point-by-point rebuttal to the full referee reports verbatim, as provided at the end of this letter.

- provide the completed Editorial Policy Checklist (found here <https://www.nature.com/authors/policies/Policy.pdf>), and Reporting Summary (found here <https://www.nature.com/authors/policies/ReportingSummary.pdf>). This is essential for reconsideration of the manuscript and these documents will be available to editors and referees in the event of peer review. For more information see <http://www.nature.com/authors/policies/availability.html> or contact me.

Nature Cell Biology is committed to improving transparency in authorship. As part of our efforts in this direction, we are now requesting that all authors identified as 'corresponding author' on published papers create and link their Open Researcher and Contributor Identifier (ORCID) with their account on the Manuscript Tracking System (MTS), prior to acceptance. ORCID helps the scientific community achieve unambiguous attribution of all scholarly contributions. You can create and link your ORCID from the home page of the MTS by clicking on 'Modify my Springer Nature account'. For more information please visit www.springernature.com/orcid.

[Redacted]

We hope that you will find our referees' comments and editorial guidance helpful. Please do not hesitate to contact me if there is anything you would like to discuss. Thank you again for considering the journal for your work.

Best wishes,

Melina

Melina Casadio, PhD
Senior Editor, Nature Cell Biology
ORCID ID: <https://orcid.org/0000-0003-2389-2243>

Reviewers' Comments:

Reviewer #1:

Remarks to the Author:

The manuscript by Riffelmacher T et al. makes an interesting observation regarding the appearance of two distinct MAIT cell populations following immunization with a Salmonella vaccine strain. The two populations, marked by the expression of Klrp1 and CD127, differ in their gene expression signature, in their function and in their location in the lung. Both populations are metabolically different and engage in different metabolic pathways to carry out their functions. In general the manuscript is well

written and the observations are interesting and novel. The methods utilized are appropriate and the results convincingly demonstrate the existence and differential characteristics of these two MAIT cell populations. Please see below some comments.

1. My main comment for this manuscript is the origin of both MAIT populations and whether they are transient states or represent distinct lineages. This is an important question, but as the authors acknowledge, the experiments carried out to determine whether one originates from the other are not conclusive and the potential of cell plasticity cannot be ruled out. Moreover, the time points used to measure MAIT cell phenotype and function seems to be rather late during the immune response. How do these populations behave and function in the acute phase of the immune response?
2. Figure 1A – why was day 60 selected? A kinetics of the expansion of MAIT cells would be more informative than an end time point. Also, Why was day 40 chosen for scRNA-seq and not 60 days, as in Figure 1A?
3. How do the authors explain the differential role of both MAIT populations in protecting from bacterial vs. viral infections?
4. It would have been interesting to include in the discussion the potential presence/absence of these MAIT cell populations and their potential similarities/differences in other anatomical locations besides the lungs.
5. The metabolic differences are interesting and in line with what observed in T cell populations other than MAIT cells, but the use of cytokine production as a readout for quantifying energetic metabolism and making a parallel to SCENITH is not correct. SCENITH uses a global biological process (translation of all proteins) that has been shown to correlate with ATP production to examine the energetic status of single cells. Cytokine production is intrinsically not a global biological process, and is specific for the cytokine being examined. Moreover, the authors should provide evidence that the measurement of cytokine expression as a readout for energetic status correlates with the amount of ATP being produced.

Reviewer #2:

Remarks to the Author:

This is an important study that provides clear evidence for the existence of developmentally and functionally distinct lineages of MAIT cells that arise following stimulation in mice. That these lineages appear to be stable, and very distinct in terms of their metabolic requirements, cytokine production and response to antigenic versus cytokine stimulation provides an important advance for the field of MAIT cell biology. The relative role of cytokines versus antigen in driving different responses from these cells has been a long-standing question that this manuscript helps to resolve. In addition, analysis of the metabolic programs engaged by distinct MAIT1 and MAIT17 cell lineages show that they align to metabolic requirements characteristic of type-1 and type-17 T cell responses.

The lung was the only tissue examined in this study. Though it makes sense to focus on the lung with the bacterial infection models used, it would have been very interesting to look in other tissues in case the MAIT cells that differentiate in infected lung can migrate elsewhere. This is particularly important given the associations with resident memory cells that are drawn in this manuscript. If aaMAIT cells can leave the lung they may be heavily influenced by other environments. Eg those that home to liver would encounter very different lipid and glucose environment that may influence their ongoing differentiation. Similarly, it would have been interesting to examine whether the aaMAIT cells studied here are present in lymph nodes and spleen. Also, do similar populations develop in gut mucosa or skin?

Fate mapping experiments here indicated that CD127+ aaMAIT cells are a separate lineage that does not give rise to KLRG1+ aaMAIT – which are absent in the lung at steady state, instead appears to be

a subset with a more circulatory signature. Does this indicate that KLRG1+ aaMAIT cells are recruited from other sites and subsequently expand to BRD509 challenge? This possible recruitment route should be explored

Several studies have found that MAIT1/17 cell lineages arise early in development and have provided markers to define these mutually exclusive subset, that are similarly defined by RORgt and Tbet. These protein signatures are not described here, did the authors clarify that aaMAIT cells that arise also express these subset-specific markers, for example, expression of CD122, CD319, ICOS?

CD127+ aaMAIT cells appear to expand via TCR/antigenic cues and protect via an antigen/MR1-dependent manner while KLRG1+ aaMAIT cells respond to cytokine cues and protect via non TCR-mechanisms such as cytokines, did the authors explore the relative clonality of aaMAIT to further support this concept?

As this builds upon the BRD509 vaccination model for MAIT cell expansion described in ref 16 (Chen et al, 2016 Mucosal Imm), there appears to be a difference in the transcriptional signatures of the expanded cells. This paper shows that expanded lung MAIT cells acquire a RORgt+ Tbet+ double positive signature, and this phenotype persists even >100 days. In contrast, the authors here see mutually exclusive KLRG1+ Tbet+ or CD127+ RORgt+ cells; is there an explanation for this discrepancy?

A very important question is how do the key findings in this study relate to human MAIT cell biology. Do humans have similar populations or aaMAIT cells? Human MAIT cells are known to be phenotypically and functionally heterogeneous – and are more likely to include antigen-experienced populations. Can populations of human MAIT cells that resemble KLRG1+ or CD127+ aaMAIT cells be detected in human blood or other tissue? This would significantly enhance the impact of this study for the target journal.

Some more minor points.

Line 54 – The wording is questionable. MAIT cells will recognise host MR1, but they will be tolerant to it.

Lines 70-74. It is fair to cite the first paper that defined the existence of two separate lineages (IFN-g+ Tbet+ MAIT 1 and IL-17+ RORgt+ MAIT 17 cells) (Rahimpour et al 2015 JEM.)

Line 105. Does the invasion mutant strain lead to the same bacterial load as the wt strain? If not, that may explain why there is less DC activation.

Line 116. The SLC mutant has not been adequately explained to make it clear what this result means.

Line 125. Fig 1D. It looks like many of the IFNg+ MAIT cells fall within the same location as the IL17A+ MAIT cells and that most IL17A+ events are also IFNg+. Does this match with other data? What timepoint is this data from?

Line 140. "... which is attributable only partially to the expansion of Klrg1+ aaMAIT cells (Fig 1G)". I do not see how this statement is supported by the data.

Line 144. "and produced more IFN-g than either CD127+ aaMAIT or steady-state MAIT cells". The production of IFNg is not at all clear and is not convincingly different from Klrg1- MAIT cells. Is this reproducible and was there an isotype control to ensure the few positive cells are really positive?

Line 200. The slightly more pronounced protection from Klr1+ MAIT cells is because 3/5 mice survived versus 2/5 mice that received Klr1- MAIT cells. I do not think this is sufficiently powered to compare and contrast these two subsets in this assay, other than to say that both offered partial protection. A repeat experiment would be helpful here.

Figure 5E. The IFN γ response in the 5OPRU stimulated control sample is surprisingly high at 80%. This is not in line with what is typically seen in this or other studies (eg. Figure 5C).

Figure 6. The figure legend title stating 'inactivated but highly responsive mitochondria' is contradictory and confusing, please clarify.

Line 369. Based on the microscopy - It is not clear that there is any staining showing low intensity mitochondria in Klr1+ aaMAIT cells. This image at least needs labelling to show where these are as the image looks negative to me.

Reviewer #3:

Remarks to the Author:

In the manuscript "Divergent metabolic programs control two populations of MAIT cells that protect the lung", the authors identified two states of antigen-adapted MAIT cells and showed their distinct function, gene expression programs and metabolic profiles. The studies were of strong physiological significance and provided in-depth mechanistic insights. There are a few important questions for the authors to address, and in particular, question 1 below on the metabolic dependencies.

1. The authors mainly used pharmacological approaches to target fatty acid and glucose metabolism (e.g. Etomoxir and 2-DG), but these inhibitors are well appreciated to have off-target effects in various cells including immune cells. The authors should use alternative approaches to target these pathways in order to rigorously establish their findings. Also, more mechanistic studies will provide additional evidence for the metabolic dependencies as the authors proposed. For example, do the two populations of MAIT cells express distinct transporters (e.g. CD36 and Glut1) for fatty acids and glucose? Do Etomoxir and 2-DG indeed cause the expected metabolic changes?

2. In Fig. 2, the authors showed that the two populations of MAIT cells conferred differential protection. However, it is unclear if such functional differences are due to differential expansion in vivo, or due to cell-autonomous functional differences on a per-cell basis. It will be important for the authors to explore such mechanistic differences.

3. Given the expression of similar markers of antigen-adapted MAIT cells as effector and memory CD8 cells (also distinguished by KLRG1 and CD127 expression), the authors should compare their results with other T cell subsets, e.g. by GSEA using C7 immunological datasets.

READABILITY OF MANUSCRIPTS – Nature Cell Biology is read by cell biologists from diverse backgrounds, many of whom are not native English speakers. Authors should aim to communicate their findings clearly, explaining technical jargon that might be unfamiliar to non-specialists, and

avoiding non-standard abbreviations. Titles and abstracts should concisely communicate the main findings of the study, and the background, rationale, results and conclusions should be clearly explained in the manuscript in a manner accessible to a broad cell biology audience. Nature Cell Biology uses British spelling.

METHODS – Nature Cell Biology publishes methods online. The methods section should be provided as a separate Word document, which will be copyedited and appended to the manuscript PDF, and

incorporated within the HTML format of the paper.

Methods should be written concisely, but should contain all elements necessary to allow interpretation and replication of the results. As a guideline, Methods sections typically do not exceed 3,000 words. The Methods should be divided into subsections listing reagents and techniques. When citing previous methods, accurate references should be provided and any alterations should be noted. Information must be provided about: antibody dilutions, company names, catalogue numbers and clone numbers for monoclonal antibodies; sequences of RNAi and cDNA probes/primers or company names and catalogue numbers if reagents are commercial; cell line names, sources and information on cell line identity and authentication. Animal studies and experiments involving human subjects must be reported in detail, identifying the committees approving the protocols. For studies involving human subjects/samples, a statement must be included confirming that informed consent was obtained. Statistical analyses and information on the reproducibility of experimental results should be provided in a section titled "Statistics and Reproducibility".

All Nature Cell Biology manuscripts submitted on or after March 21 2016 must include a Data availability statement at the end of the Methods section. For Springer Nature policies on data availability see <http://www.nature.com/authors/policies/availability.html>; for more information on this particular policy see <http://www.nature.com/authors/policies/data/data-availability-statements-data-citations.pdf>. The Data availability statement should include:

- Accession codes for primary datasets (generated during the study under consideration and designated as "primary accessions") and secondary datasets (published datasets reanalysed during the study under consideration, designated as "referenced accessions"). For primary accessions data should be made public to coincide with publication of the manuscript. A list of data types for which submission to community-endorsed public repositories is mandated (including sequence, structure, microarray, deep sequencing data) can be found here <http://www.nature.com/authors/policies/availability.html#data>.
- Unique identifiers (accession codes, DOIs or other unique persistent identifier) and hyperlinks for datasets deposited in an approved repository, but for which data deposition is not mandated (see here for details <http://www.nature.com/sdata/data-policies/repositories>).
- At a minimum, please include a statement confirming that all relevant data are available from the authors, and/or are included with the manuscript (e.g. as source data or supplementary information), listing which data are included (e.g. by figure panels and data types) and mentioning any restrictions on availability.
- If a dataset has a Digital Object Identifier (DOI) as its unique identifier, we strongly encourage including this in the Reference list and citing the dataset in the Methods.

We recommend that you upload the step-by-step protocols used in this manuscript to the Protocol Exchange. More details can be found at www.nature.com/protocolexchange/about.

FIGURES – Colour figure publication costs \$600 for the first, and \$300 for each subsequent colour figure. All panels of a multi-panel figure must be logically connected and arranged as they would appear in the final version. Unnecessary figures and figure panels should be avoided (e.g. data

presented in small tables could be stated briefly in the text instead).

All imaging data should be accompanied by scale bars, which should be defined in the legend. Cropped images of gels/blots are acceptable, but need to be accompanied by size markers, and to retain visible background signal within the linear range (i.e. should not be saturated). The boundaries of panels with low background have to be demarked with black lines. Splicing of panels should only be considered if unavoidable, and must be clearly marked on the figure, and noted in the legend with a statement on whether the samples were obtained and processed simultaneously. Quantitative comparisons between samples on different gels/blots are discouraged; if this is unavoidable, it should only be performed for samples derived from the same experiment with gels/blots were processed in parallel, which needs to be stated in the legend.

All placed images (i.e. a photo incorporated into a figure) should be on a separate layer and independent from any superimposed scale bars or text. Individual photographic images must be a minimum of 300+ DPI (at actual size) or kept constant from the original picture acquisition and not

decreased in resolution post image acquisition. All colour artwork should be RGB format.

The total number of Supplementary Figures (not including the “unprocessed scans” Supplementary Figure) should not exceed the number of main display items (figures and/or tables (see our Guide to Authors and March 2012 editorial <http://www.nature.com/ncb/authors/submit/index.html#suppinfo>; <http://www.nature.com/ncb/journal/v14/n3/index.html#ed>). No restrictions apply to Supplementary Tables or Videos, but we advise authors to be selective in including supplemental data.

GUIDELINES FOR EXPERIMENTAL AND STATISTICAL REPORTING

REPORTING REQUIREMENTS – To improve the quality of methods and statistics reporting in our papers we have recently revised the reporting checklist we introduced in 2013. We are now asking all life sciences authors to complete two items: an Editorial Policy Checklist (found here <https://www.nature.com/authors/policies/Policy.pdf>) that verifies compliance with all required

editorial policies and a reporting summary (found here <https://www.nature.com/authors/policies/ReportingSummary.pdf>) that collects information on experimental design and reagents. These documents are available to referees to aid the evaluation of the manuscript. Please note that these forms are dynamic 'smart pdfs' and must therefore be downloaded and completed in Adobe Reader. We will then flatten them for ease of use by the reviewers. If you would like to reference the guidance text as you complete the template, please access these flattened versions at <http://www.nature.com/authors/policies/availability.html>.

Author Rebuttal to Initial comments

Reviewer #1:

Remarks to the Author:

The manuscript by Riffelmacher T et al. makes an interesting observation regarding the appearance of two distinct MAIT cell populations following immunization with a Salmonella vaccine strain. The two populations, marked by the expression of Klr1g1 and CD127, differ in their gene expression signature, in their function and in their location in the lung. Both populations are metabolically different and engage in different metabolic pathways to carry out their functions. In general the manuscript is well written and the observations are interesting and novel. The methods utilized are appropriate and the results convincingly demonstrate the existence and differential characteristics of these two MAIT cell populations. Please see below some comments.

**1. My main comment for this manuscript is the origin of both MAIT populations and whether they are transient states or represent distinct lineages. This is an important question, but as the authors acknowledge, the experiments carried out to determine whether one originates from the other are not conclusive and the potential of cell plasticity cannot be ruled out.*

The origin of the antigen-adapted (aa) MAIT subsets is a crucial question. We have addressed the origin of the Klr1g1⁺ aaMAIT cells previously, by experiments using fate-mapping (FM) for *Tbx21* (T-bet) (**Fig. S2A+B**) and *Il17* expression (**Fig. 2A**). We backed this up with adoptive transfer experiments of Klr1g1⁺ and Klr1g1⁻ populations followed by re-infections (**Fig. 2B**), as well as culture and re-activation assays of sorted subsets in vitro (**Fig. 2C**). These data indicated that the Klr1g1⁺ aaMAIT cells do not convert to IL-7R⁺ cells. We acknowledged that our data did not exclude the possibility that the minor fraction of CD127⁺ aaMAIT cells, that are IL17-FM negative (Fig. 2A), could give rise to the Klr1g1⁺ aaMAIT. We now have looked deeper into this issue with additional adoptive transfer experiments. We identified and purified a subset of Klr1g1 negative aaMAIT cells that have low mitochondrial potential as evidenced by mitoTracker (MTR) staining. MTR^{low} staining in IL-7R⁺ aaMAIT cells correlated with an IL-17 FM negative phenotype (**new Figs. S2D-E**). This sub-population that likely had never expressed *Il17a* was enriched for the ability to give rise to Klr1g1⁺ aaMAIT cells at later timepoints after transfer and BRD509 infection (**new Fig. S2C**). Furthermore, we sorted Klr1g1 negative aaMAIT cells that have high or low MTR and compared them separately to their Klr1g1⁺ counterparts in a bulk RNA-seq experiment. This analysis confirmed that many of the genes that are highly expressed in Klr1g1⁺ aaMAIT cells were also significantly increased in the IL-7R⁺ MTR^{low} aaMAIT cell subset (*Klr1g1, Cd8b, Ccl5, Gzmb, Gzma, Ly6c2*; **new Fig. S2F**). Together, these data indicated that the small fraction of CD127⁺, Klr1g1⁻ aaMAIT cells, which are low for MTR and IL17-FM negative, were enriched for aaMAIT cells that have increased capacity to form Klr1g1⁺ aaMAIT cells at later timepoints. This is a crucial new finding that identifies a precursor of the Klr1g1⁺ cells. This subset of Klr1g1⁻ aaMAIT cells also gave rise to IL-7R⁺ aaMAIT cells after transfer (**Fig. S2C**), however, and only a minority of the cells became Klr1g1⁺. These data suggest this population is heterogeneous, possibly with precursors

having different potential, or alternatively, there could be plasticity at this stage. Regardless, the new data confirm that the Klr1g1⁺ and IL-7R⁺ MTR^{high} cells did not readily inter-convert, despite infection and vast population increases, suggesting they are not cell states.

Moreover, the time points used to measure MAIT cell phenotype and function seems to be rather late during the immune response. How do these populations behave and function in the acute phase of the immune response?

We have tracked total MAIT cells (**new Fig. 1A**) and Klr1g1⁺ MAIT cells (**new Fig. 1E**) longitudinally during the acute phase and intermediate timepoints. As shown previously in the literature for total MAIT cells, and confirmed in our new time course analysis for Klr1g1⁺ aaMAIT cells (**new Fig. 1A+E**), Klr1g1⁺ cells were absent in the lung at steady state and during the acute phase, and started appearing in the lung 1-2 weeks after vaccination. In order to characterize the critical acute phase of the effector response in more detail, we now provide new sc-RNA-seq data of MAIT cells at day 6 post vaccination (**new Fig. S1M-O**). These data show that day 6 MAIT cells clustered separately from the later timepoints, did not express Klr1g1, did not exhibit a circulatory gene signature, were more MAIT17 biased than MAIT1, had expression of a conventional CD8⁺ T cell effector gene signature, and had reduced expression of genes associated with CD8 T cell memory than their day 40 counterparts (**new Fig. S1M**). Pathway analysis further revealed that the most differentially expressed pathways at day 6 were related to cell replication, protein translation, and macromolecular biosynthesis (**new Fig. S1N**), in keeping with the striking proliferative burst that these cells underwent during this time (**new Fig. 1A**). A heatmap view is also provided that shows the top differentially expressed genes during this time in relation to day 0 and day 40 (**new Fig. S1O**). At the protein level, we show that a strong IL-17A MAIT cell response occurred during this time in the lung and to a lesser extent, there was IFN γ and Granzyme B production (**new Fig. S1P**), similar to previously reported data.

2. Figure 1A – why was day 60 selected? A kinetics of the expansion of MAIT cells would be more informative than an end time point. Also, Why was day 40 chosen for scRNA-seq and not 60 days, as in Figure 1A?

We provide new data and corroborated the previous literature on the kinetics of MAIT cell population increase during vaccination (**new Fig. 1A**). We chose day 40 after infection because at this time, bacteria were cleared and Klr1g1⁺ aaMAIT cells were present at near peak levels. Our data from later timepoints, day 60 up to >day 100, did not show large changes in MAIT cell phenotype (**Fig. 1H** and data not shown), in agreement with Chen et al., 2017.

3. How do the authors explain the differential role of both MAIT populations in protecting from bacterial vs. viral infections?

When subsets were transferred separately, protection from *S. pneumoniae* was mediated by CD127⁺ aaMAIT, but not by Klrp1⁺ aaMAIT cells (**Fig. 2H**). This agrees with literature indicating that IL-17A producing innate T cells were important for this bacterium (Murray et al., ref. 37, Hassane et al, refs. 39+40). In contrast, both aaMAIT cell subsets conferred a survival benefit during influenza challenge (**Fig. 2I**). Anti-viral protection was likely mediated via activation of aaMAIT cells by IL-12 or other cytokines, as indicated by van Wilgenburg et al. (ref. 22) with protective roles for various cytokines, including IFN γ . These data suggest that the relative importance of responses by different aaMAIT subsets depended on the nature of the infection, the mode of activation of MAIT cells and the cytokines elicited. Our in vitro re-activation data further provide a rationale for understanding the different protective capacities of the two aaMAIT cell subsets: the presence of microbial antigen, as during *S. pneumoniae* infection, elicited a qualitatively different response by aaMAIT cells compared to a pure cytokine-driven stimulation (**Fig. 3G-I, Fig. S3C, Fig. 4F-H, Fig. 5B-G, Fig. S5A, B, Fig. 6K, L**). This provides a mechanistic explanation for the differential protective capacities of aaMAIT lineages against an extracellular bacterium as opposed to an influenza viral challenge.

4. It would have been interesting to include in the discussion the potential presence/absence of these MAIT cell populations and their potential similarities/differences in other anatomical locations besides the lungs.

We provide new data that showed MAIT cell frequencies in different sites before infection and 40 days post vaccination. Strong MAIT cell population increases (<10-fold) were evident in lung, and to a lesser extent in liver, spleen, blood and kidney. No expansion was evident in mesenteric, axillary, brachial or caudal lymph nodes, skin, and intestinal tissues (**new Fig. S1Q**). A significant pool of Klrp1⁺ MAIT cells was not present in naïve mice in any of these anatomical sites, with the exception of colon and small intestine, where very small numbers were evident. 40 days post vaccination, a significant increase in Klrp1⁺ aaMAIT cells was evident in all organs. This is in keeping with our data indicating that Klrp1⁺, but not CD127⁺ aaMAIT cells were in blood and recirculated (**Fig. 2D,E**). We further analyzed the expansion and contraction kinetics of aaMAIT cells in the liver after vaccination, which followed the pattern we observed in the lung (**new Fig. S2R**). Together these data suggested that there was not a significant pool of Klrp1⁺ aaMAIT cells in naïve mice, and following vaccination, these cells were circulating, presumably from lung or or brachial lymph nodes, and found in a number of peripheral and lymphatic organs.

**5. The metabolic differences are interesting and in line with what observed in T cell populations other than MAIT cells, but the use of cytokine production as a readout for quantifying energetic metabolism and making a parallel to SCENITH is not correct. SCENITH uses a global biological process (translation of all proteins) that has been shown to correlate with ATP production to examine the energetic status of single cells.*

Cytokine production is intrinsically not a global biological process, and is specific for the cytokine being examined. Moreover, the authors should provide evidence that the measurement of cytokine expression as a readout for energetic status correlates with the amount of ATP being produced.

SCENITH is a relatively new assay that has been applied in studies since its description in 2020. We were inspired by this assay when we designed our assay that uses the same formulas and drug intervention strategies. The difference is that as a readout we did not use overall translation as a correlate of ATP production, but instead tested the capacity of immune cells to produce their key effector cytokines as a function of their mitochondrial and glycolytic dependencies and capacities. We now clearly state the goal and design of this assay to distinguish it from SCENITH, and we pointed out that cytokine production is not a global cellular assay. The assay was valuable, however, because it provided insight into the extent to which cytokine production by MAIT cell subsets depended on different metabolic pathways, and it corroborated other functional studies (**Fig. 3G-I, Fig. 4F-H, Fig. 5C-D**).

Reviewer #2: (MAIT cell referee)

Remarks to the Author:

This is an important study that provides clear evidence for the existence of developmentally and functionally distinct lineages of MAIT cells that arise following stimulation in mice. That these lineages appear to be stable, and very distinct in terms of their metabolic requirements, cytokine production and response to antigenic versus cytokine stimulation provides an important advance for the field of MAIT cell biology. The relative role of cytokines versus antigen in driving different responses from these cells has been a long-standing question that this manuscript helps to resolve. In addition, analysis of the metabolic programs engaged by distinct MAIT1 and MAIT17 cell lineages show that they align to metabolic requirements characteristic of type-1 and type-17 T cell responses.

The lung was the only tissue examined in this study. Though it makes sense to focus on the lung with the bacterial infection models used, it would have been very interesting to look in other tissues in case the MAIT cells that differentiate in infected lung can migrate elsewhere. This is particularly important given the associations with resident memory cells that are drawn in this manuscript. If aaMAIT cells can leave the lung they may be heavily influenced by other environments. Eg those that home to liver would encounter very different lipid and glucose environment that may influence their ongoing differentiation. Similarly, it would have been interesting to examine whether the aaMAIT cells studied here are present in lymph nodes and spleen. Also, do similar populations develop in gut mucosa or skin?

The question regarding the presence of *aa*MAIT cells in other anatomical sites is important. We now provide extensive data on naïve and memory (d40) timepoints, describing the total MAIT cell frequency and Klr $g1^+$ MAIT cells in lung, liver, spleen, blood, kidney, mesenteric, axillary, brachial or caudal lymph nodes, skin, and intestinal tissues (**New Fig. S1Q**). We further analyzed the liver, a secondary site of MAIT cell expansion following lung infections. The data showed that the frequency of liver MAIT cells with the capacity for cytokine production at day 6 was similar to MAIT cells in the lung (**new Fig. S1P**), with IL-17 prevalent compared to IFN γ . Furthermore, the time course showed that a MAIT cell population increase in the liver preceded the increase in the *aa*MAIT Klr $g1^+$ subset (**new Fig. S1R**), also similar to the lung. We cannot exclude that there are important tissue differences in the *aa*MAIT cells in particular tissues, especially considering different routes of infection, but such a detailed analysis would be beyond the scope of this paper.

Fate mapping experiments here indicated that CD127⁺ aaMAIT cells are a separate lineage that does not give rise to KLRG1⁺ aaMAIT – which are absent in the lung at steady state, instead appears to be a subset with a more circulatory signature. Does this indicate that KLRG1⁺ aaMAIT cells are recruited from other sites and subsequently expand to BRD509 challenge? This possible recruitment route should be explored.

The analysis of many different tissues before infection indicated there was not a significant reservoir of Klr $g1^+$ MAIT cells that could be recruited to the lung from other sites (**new Fig. S1Q**). Therefore, the population increase likely occurred to some extent locally in the first 1-2 weeks. However, our evidence suggests that Klr $g1^+$ *aa*MAIT cells recirculate. This conclusion is derived from the gene expression signature of this subset (**Fig 2E**), their increased labeling with anti-CD45 (**Fig. 2D**), and the increased percentage of Klr $g1^+$ *aa*MAIT cells in diverse sites, even in which a MAIT cell population increase did not occur (**New Fig. S1Q**). These sites presumably were removed from the infection, and the data suggest that MAIT cells activated in the lung and perhaps liver seeded other sites.

*Several studies have found that MAIT1/17 cell lineages arise early in development and have provided markers to define these mutually exclusive subset, that are similarly defined by ROR γ t and Tbet. These protein signatures are not described here, did the authors clarify that *aa*MAIT cells that arise also express these subset-specific markers, for example, expression of CD122, CD319, ICOS?*

We provide staining of Ror γ t and T-bet (**Fig. 1F+G**). We provide new data showing additional subset-specific marker expression: CD8 α , Ccl5, Ly6c2, Slamf7 (=CD319), CD122, GzmA and GzmB were all higher in Klr $g1^+$ *aa*MAIT cells and many of them were also higher in the subset of MTR^{low}, CD127⁺, *aa*MAIT cells as compared to MTR^{high}, CD127⁺, *aa*MAIT cells. In contrast, Sdc1 and Cxcr5 were higher in CD127⁺, MTR^{high}, *aa*MAIT compared to their MTR^{low} counterparts and were absent in Klr $g1^+$ *aa*MAIT cells.

CD127⁺ aaMAIT cells appear to expand via TCR/antigenic cues and protect via an antigen/MR1-dependent manner while KLRG1⁺ aaMAIT cells respond to cytokine cues and protect via non TCR-mechanisms such as cytokines, did the authors explore the relative clonality of aaMAIT to further support this concept?

Both Klrp1 and IL-7R aaMAIT populations increased after BRD509 vaccination in an antigen-dependent manner (**Fig. S1E**). However, in protection from an extracellular bacterium, *S. pneumoniae*, IL-17 producing cells were more important, consistent with the literature (Murray et al., ref. 37, and Hassane et al, refs. 39+40). In contrast, MAIT responses to viral infections are predominantly cytokine driven (ref 22) and both subsets were important. Regarding the clonality of the MAIT cell increase following Salmonella vaccination, we found no changes in the prevalence of Vb6⁺ Vβ8.1, 8.2⁺, and Vβ6,8.1,8.2⁻ MAIT cells at steady state and at days 6, 35 and 60 post vaccination in either total MAIT cells, Klrp1⁺ or CD127⁺ aaMAIT cell subsets, respectively (**new Fig. S1S**). Unfortunately we did not obtain sufficient quality of TCR sequences from the single cell RNA-seq data, but we can rule out the expansion of a few clones. We will investigate the clonal dynamics in an additional study.

As this builds upon the BRD509 vaccination model for MAIT cell expansion described in ref 16 (Chen et al, 2017 Mucosal Imm), there appears to be a difference in the transcriptional signatures of the expanded cells. This paper shows that expanded lung MAIT cells acquire a RORgt⁺ Tbet⁺ double positive signature, and this phenotype persists even >100 days. In contrast, the authors here see mutually exclusive KLRG1⁺ Tbet⁺ or CD127⁺ RORgt⁺ cells; is there an explanation for this discrepancy?

Our transcriptomic data indicated that a significant proportion of aaMAIT cell subsets have both *Tbx21* (Tbet) and *Rorgt* messages (**Fig. 1D**). We have added additional protein staining data to **Fig. 1G**. These data indicated that the gmfi of Tbet is highest in Klrp1⁺ aaMAIT cells, but also trending higher in CD127⁺ aaMAIT cells compared to their naïve counterparts (**New Fig. 1G, fourth panel from the left**). Roryt was not significantly different between the day 40 subsets (**New Fig. 1G, second from left**). We now also quantify the % of MAIT cells that were Tbet⁺, which showed an increase in Tbet expression in the CD127⁺ aaMAIT cells compared to naïve MAIT cells (**New Fig. 1G, right panel**), although the increase was less than in the Klrp1⁺ aaMAIT cells. Therefore, many day 40 aaMAIT cells indeed are Tbet⁺, Roryt⁺.

A very important question is how do the key findings in this study relate to human MAIT cell biology. Do humans have similar populations or aaMAIT cells? Human MAIT cells are known to be phenotypically and functionally heterogeneous – and are more likely to include antigen-experienced populations. Can populations of human MAIT cells that

resemble KLRG1+ or CD127+ aaMAIT cells be detected in human blood or other tissue? This would significantly enhance the impact of this study for the target journal.

An analysis of PBMC-derived MAIT cells at steady state from 18 healthy donors by scRNA-seq has been carried out by us and the data are displayed in a figure for the reviewers (Fig. R1). The data indicate that *IL7R* expression was fairly uniform. A sub population that had marginally lower *KLRG1* transcripts was evident. Therefore, at steady-state MAIT cells from human blood express both *KLRG1* and *IL7R*. While it is noteworthy that *Klrg1* is not expressed at steady state in mouse MAIT cells, an analysis considering recent infection history and MAIT cells in different sites might be required to address subsets of human MAIT cells, which is beyond the scope of this study.

Fig. R1 Human PBMCs from 18 donors were sorted by flow cytometry and sequenced by scRNA-seq. Expression of *IL7R* and *KLRG1* are plotted as feature plots.

Some more minor points.

Line 54 – The wording is questionable. MAIT cells will recognise host MR1, but they will be tolerant to it.

Now reworded to: “Furthermore, because MR1 is highly conserved between humans, in fact in many mammals, allogeneic MAIT cells are unlikely to cause significant graft-versus-host disease

Lines 70-74. It is fair to cite the first paper that defined the existence of two separate lineages (IFN-g+ Tbet+ MAIT 1 and IL-17+ RORgt+ MAIT 17 cells) (Rahimpour et al 2015 JEM.)

This additional citation is now included to reference the existence of discrete MAIT1 and MAIT17 subsets, respectively (New ref. 21).

Line 105. Does the invasion mutant strain lead to the same bacterial load as the wt strain? If not, that may explain why there is less DC activation.

Wang et al (2019) (ref. 23) showed that the WT strain and the invasion mutant resulted in the same bacterial load in vivo at day 6 following infection of the lung, and we also did not detect any differences in their growth kinetics in our experiments.

Line 116. The SLC mutant has not been adequately explained to make it clear what this result means.

We reworded this section and added more detailed description in methodology to indicate that quorum firing is used to induce cell lysis. (new lines 118-119, 672-674).

Line 125. Fig 1D. It looks like many of the IFN γ + MAIT cells fall within the same location as the IL17A+ MAIT cells and that most IL17A+ events are also IFN γ +. Does this match with other data? What timepoint is this data from?

The timepoints for Fig. 1D are the same as Fig. 1C, day 0 and day 40, and this is now indicated in the legend. A partial overlap of Th1 and Th17 markers was also reported previously in MAIT cells following BRD509 vaccination, and is in line with the new data showing an increase in T-bet that was also evident in the CD127⁺ aaMAIT cells, although less than in Klrp1⁺ aaMAIT cells (**new Fig. 1G**).

Line 140. "... which is attributable only partially to the expansion of Klrp1+ aaMAIT cells (Fig 1G)". I do not see how this statement is supported by the data.

On a total MAIT cell population level, T-bet expression increases at later timepoints (**Fig. 1G**), but it is not certain if this is attributable only to the numerous Klrp1⁺ aaMAIT cells with high T-bet, or if CD127⁺ aaMAIT cells with somewhat increased T-bet staining also contribute. Our new data more clearly show that T-bet expression also increased to some extent in Klrp1 negative aaMAIT cells. We have re-worded this section to make the meaning more apparent.

Line 144. "and produced more IFN-g than either CD127+ aaMAIT or steady-state MAIT cells". The production of IFN γ is not at all clear and is not convincingly different from Klrp1- MAIT cells. Is this reproducible and was there an isotype control to ensure the few positive cells are really positive?

The IL-17A and Granzyme B production by separate aaMAIT cell subsets is highly robust at around 30% and 40% of cells, respectively. However, we find that IFN γ production following PMA + ionomycin stimulation is indeed low, and we have removed this statement and the IFN γ data. In other contexts, IFN γ expression was evident, and we comprehensively characterized IFN γ production in aaMAIT subsets at various timepoints and different activation conditions in **Figs. 3F,G,I, S3B+C, Fig. 4G, and Fig. 5C,D,E**.

Line 200. The slightly more pronounced protection from Klrp1+ MAIT cells is because 3/5 mice survived verses 2/5 mice that received Klrp1- MAIT cells. I do not think this is sufficiently powered to compare and contrast these two subsets in this assay, other than to say that both offered partial protection. A repeat experiment would be helpful here.

We now provide combined data from 2 experiments with higher mouse numbers (29 influenza-infected recipient mice split into three groups). The conclusion remains that both subsets confer partial protection (**Adapted Fig. 2I**).

Figure 5E. The IFN γ response in the 5OPRU stimulated control sample is surprisingly high at 80%. This is not in line with what is typically seen in this or other studies (eg. Figure 5C).

Here we tested if the presence or absence of antigen in the context of IL-12+18 stimulation impacted IFN- γ production. We have relabeled the left (cytokine) and right (Ag + cytokine) graphs in **Fig. 5E** to more clearly indicate the conditions. The high production of cytokine is expected under these circumstances and consistent across experiments, for example in **Fig. 5C (bottom right panels)**.

Figure 6. The figure legend title stating ‘inactivated but highly responsive mitochondria’ is contradictory and confusing, please clarify.

We apologize for the poor wording. The data indicate lower mitochondrial potential and activity but equal numbers of mitochondria in Klrg1⁺ aaMAIT cells. However, they quickly respond to various stimuli. We therefore rephrased to: “Klrg1⁺ aaMAIT cells have equally abundant but dormant mitochondria, that are rapidly responsive to activation”,

Line 369. Based on the microscopy - It is not clear that there is any staining showing low intensity mitochondria in Klrg1⁺ aaMAIT cells. This image at least needs labelling to show where these are as the image looks negative to me.

We apologize that, due to down sampling and file size limitations, it was difficult to clearly appreciate the low intensity mitochondria. We have updated the panel with new, higher contrast split channel images for better visualization.

Reviewer #3: immunometabolism expert

Remarks to the Author:

In the manuscript “Divergent metabolic programs control two populations of MAIT cells that protect the lung”, the authors identified two states of antigen-adapted MAIT cells and showed their distinct function, gene expression programs and metabolic profiles. The studies were of strong physiological significance and provided in-depth mechanistic insights. There are a few important questions for the authors to address, and in particular, question 1 below on the metabolic dependencies.

**1. The authors mainly used pharmacological approaches to target fatty acid and glucose metabolism (e.g. Etomoxir and 2-DG), but these inhibitors are well appreciated to have off-target effects in various cells including immune cells. The authors should use alternative approaches to target these pathways in order to rigorously establish their findings. Also, more mechanistic studies will provide additional evidence for the metabolic dependencies as the authors proposed. For example, do the two populations of MAIT cells express distinct transporters (e.g. CD36 and Glut1) for fatty acids and glucose? Do Etomoxir and 2-DG indeed cause the expected metabolic changes?*

In our study, we used two metabolic inhibitors that are widely used in the field to interfere with glycolytic and fatty acid/mitochondrial metabolism, respectively: 2-DG to target early glycolysis and etomoxir to target Cpt1-mediated fatty acid transport and mitochondrial OXPHOS. To validate this, we also used three genetic animal models that modulate these same pathways, *Atg5* deletion, *Vhl* deletion and *Hif1a* deletion, all targeted to mature T cells. We show that these genetic deletions are acting “on-target” by modulating glycolysis and mitochondrial function and they elicit the expected functional impact on *aa*MAIT cells (**Figs. 7B, 7I**). We appreciate the suggestion to also measure relevant transporters, CD36 and Glut1. CD36 protein expression was significantly higher in CD127⁺ *aa*MAIT cells than their Klr1⁺ counterparts (**Fig 4 C**). We also show a small but significant increase in the level of Glut1 protein expression in Klr1⁺ *aa*MAIT cells (**new Fig. S5A**), which is in line with their higher dependence on glycolysis to produce their signature cytokine, IFN γ . Furthermore, to demonstrate that metabolic inhibitors indeed caused the expected metabolic changes, we now demonstrate a significant decrease in extracellular acidification in response to the same dose and duration of 2-DG treatment used in our functional assays (**new Fig. S5**). Because of limited cell numbers, we carried out these assays on CD8⁺ splenic T cells from day 40 infected mice. Similarly, we show a significant impairment of oxygen consumption rate in response to etomoxir treatment under the same treatment conditions used in prior assays (**new Fig. S4B**), which elicited no evident cytotoxicity. We also point the reviewer’s attention to our dose response study for 2-DG in **Fig. 5D**, which shows linear impact on IFN- γ production even at lower doses.

2. In Fig. 2, the authors showed that the two populations of MAIT cells conferred differential protection. However, it is unclear if such functional differences are due to differential expansion *in vivo*, or due to cell-autonomous functional differences on a per-cell basis. It will be important for the authors to explore such mechanistic differences.

The reviewer raises an important point. Despite the well-established literature that implicates IL-17 production by innate cells as critical protective mechanism in the context of *S. pneumoniae* infection (Murray et al., ref. 37,39,40), differential expansion of subsets may quantitatively impact their contribution to protection. We now address this with new data that quantifies absolute numbers of *aa*MAIT cells resolved by subset in the context of secondary infection with *S. pneumoniae* (**new Fig. S3B, right**). These data indicate that the ratio of subsets at day 40 following *Salmonella* vaccination is not changed rapidly following a secondary infection with *S. pneumoniae*. Equal numbers of Klr1⁺ or CD127⁺ *aa*MAIT cells were adoptively transferred into naïve, immune-competent host mice to evaluate their anti-bacterial and anti-viral protective capacities. Further, IFN γ production by MAIT cells has been reported to protect against influenza infection (Van Wilgenburg, ref. 22). The fact that IL-17 producing cells protect from *S. pneumoniae*, while either IL-17 or IFN γ producing cells protect from influenza virus, is consistent with the literature showing protective capacity for IL-17 (PMID 31777067) in this infection. Therefore, we consider the described cell-intrinsic functional differences are most likely providing this differential protection.

3. *Given the expression of similar markers of antigen-adapted MAIT cells as effector and memory CD8 cells (also distinguished by KLRG1 and CD127 expression), the authors should compare their results with other T cell subsets, e.g. by GSEA using C7 immunological datasets.*

We extended our sc-RNAseq analysis with new data from the effector phase and then applied Gene Set Enrichment Analysis of several datasets from the C7 database. Interestingly, a gene list that describes effector CD8⁺ T cell upregulated genes vs memory T cells (GSE1000002_1582_200_UP) was evident in MAIT cells from the effector phase (day 6) while the corresponding CD8⁺ memory signature (GSE1000002_1582_200_DN) was equally evident in either day 40 aaMAIT cell subset. We further found a gene set that separates terminal effector memory T cells from conventional effector/central memory T cells that distinguished Klrp1⁺ aaMAIT cells, enriched for the terminal-T_{EM} genes, from their CD127⁺ counterparts (Milner et al, PNAS 2020) (**new Fig. S1M**).

Decision Letter, first revision:

Our ref: NCB-A49273A

7th March 2023

Dear Dr. Kronenberg,

Thank you for submitting your revised manuscript "Divergent metabolic programs control two populations of MAIT cells that protect the lung" (NCB-A49273A). It has now been seen by two of the original referees and their comments are below. The original MAIT cell expert Rev#2 was not available to re-review, unfortunately. However, the reviewers' points had strong overlap, and I asked Rev#1 and Rev#3 to please take a close look at how you addressed Rev#2's comments, and both kindly agreed. The reviewers find that the paper has improved in revision, with Reviewer #1 sharing with us that this applies also to how you have addressed Rev#2's points. Therefore, we'll be happy in principle to publish the study in Nature Cell Biology, pending minor revisions to comply with our editorial and formatting guidelines.

Please note that the current version of your manuscript is in a PDF format. Could you please email us a copy of the file in an editable format (Microsoft Word or LaTeX), as we cannot proceed with PDFs at this stage? Many thanks in advance for your attention to this point.

With the Word file in-hand, we will be performing detailed checks on your paper and will send you a checklist detailing our editorial and formatting requirements 1-2 weeks after receipt of the Word file. Please do not upload the final materials and make any revisions until you receive this additional information from us.

Thank you again for your interest in Nature Cell Biology. Please do not hesitate to contact me if you have any questions.

Sincerely,

Melina

Melina Casadio, PhD
Senior Editor, Nature Cell Biology
ORCID ID: <https://orcid.org/0000-0003-2389-2243>

Reviewer #1 (Remarks to the Author):

The authors have thoroughly addressed all my comments. Congratulations on a great work.

Reviewer #3 (Remarks to the Author):

The authors have successfully addressed my previous concerns.

ADDITIONAL COMMENTS ON RESPONSES TO REV#2:

I have now reviewed the authors' response to reviewer 2, and I think they did a good job in addressing the concerns. I don't have additional concerns.

Decision Letter, final checks:

Our ref: NCB-A49273A

16th March 2023

Dear Dr. Kronenberg,

Thank you for your patience as we've prepared the guidelines for final submission of your Nature Cell Biology manuscript, "Divergent metabolic programs control two populations of MAIT cells that protect the lung" (NCB-A49273A). Please carefully follow the step-by-step instructions provided in the attached file, and add a response in each row of the table to indicate the changes that you have made. Please also check and comment on any additional marked-up edits we have proposed within the text. Ensuring that each point is addressed will help to ensure that your revised manuscript can be swiftly handed over to our production team.

In recognition of the time and expertise our reviewers provide to Nature Cell Biology's editorial process, we would like to formally acknowledge their contribution to the external peer review of your manuscript entitled "Divergent metabolic programs control two populations of MAIT cells that protect the lung". For those reviewers who give their assent, we will be publishing their names alongside the published article.

Nature Cell Biology offers a Transparent Peer Review option for new original research manuscripts submitted after December 1st, 2019. As part of this initiative, we encourage our authors to support increased transparency into the peer review process by agreeing to have the reviewer comments, author rebuttal letters, and editorial decision letters published as a Supplementary item. When you

submit your final files please clearly state in your cover letter whether or not you would like to participate in this initiative. Please note that failure to state your preference will result in delays in accepting your manuscript for publication.

Cover suggestions

As you prepare your final files we encourage you to consider whether you have any images or illustrations that may be appropriate for use on the cover of Nature Cell Biology.

Nature Cell Biology has now transitioned to a unified Rights Collection system which will allow our Author Services team to quickly and easily collect the rights and permissions required to publish your work. Approximately 10 days after your paper is formally accepted, you will receive an email in providing you with a link to complete the grant of rights. If your paper is eligible for Open Access, our Author Services team will also be in touch regarding any additional information that may be required to arrange payment for your article.

Please note that *Nature Cell Biology* is a Transformative Journal (TJ). Authors may publish their research with us through the traditional subscription access route or make their paper immediately open access through payment of an article-processing charge (APC). Authors will not be required to make a final decision about access to their article until it has been accepted. Find out more about Transformative Journals

Authors may need to take specific actions to achieve compliance with funder and institutional open access mandates. If your research is supported by a funder that requires immediate open access (e.g. according to Plan S principles) then you should select the gold OA route, and we will direct you to the compliant route where possible. For authors selecting the subscription publication route, the journal's standard licensing terms will need to be accepted, including self-archiving policies. Those licensing terms will supersede any other terms that the author or any third party may assert apply to any version of the manuscript.

For information regarding our different publishing models please see our Transformative

Journals page. If you have any questions about costs, Open Access requirements, or our legal forms, please contact ASJournals@springernature.com.

Please use the following link for uploading these materials:
[Redacted]

Best regards,

Kendra Donahue
Staff
Nature Cell Biology

On behalf of

Melina Casadio, PhD
Senior Editor, Nature Cell Biology
ORCID ID: <https://orcid.org/0000-0003-2389-2243>

Reviewer #1:

Remarks to the Author:

The authors have thoroughly addressed all my comments. Congratulations on a great work.

Reviewer #3:

Remarks to the Author:

The authors have successfully addressed my previous concerns.

ADDITIONAL COMMENTS ON RESPONSES TO REV#2:

I have now reviewed the authors' response to reviewer 2, and I think they did a good job in addressing the concerns. I don't have additional concerns.

Final Decision Letter:

Dear Dr Kronenberg,

I am pleased to inform you that your manuscript, "Divergent metabolic programs control two populations of MAIT cells that protect the lung", has now been accepted for publication in Nature Cell Biology. Congratulations on this very nice study!

Please note that *Nature Cell Biology* is a Transformative Journal (TJ). Authors may publish their research with us through the traditional subscription access route or make their paper immediately open access through payment of an article-processing charge (APC). Authors will not be required to make a final decision about access to their article until it has been accepted. Find out more about Transformative Journals

To assist our authors in disseminating their research to the broader community, our SharedIt initiative

provides you with a unique shareable link that will allow anyone (with or without a subscription) to read the published article. Recipients of the link with a subscription will also be able to download and print the PDF.

If you have not already done so, we strongly recommend that you upload the step-by-step protocols used in this manuscript to the Protocol Exchange (www.nature.com/protocolexchange), an open online resource established by Nature Protocols that allows researchers to share their detailed experimental know-how. All uploaded protocols are made freely available, assigned DOIs for ease of citation and are fully searchable through nature.com. Protocols and Nature Portfolio journal papers in which they are used can be linked to one another, and this link is clearly and prominently visible in the online versions of both papers. Authors who performed the specific experiments can act as primary authors for the Protocol as they will be best placed to share the methodology details, but the Corresponding Author of the present research paper should be included as one of the authors. By uploading your Protocols to Protocol Exchange, you are enabling researchers to more readily reproduce or adapt the methodology you use, as well as increasing the visibility of your protocols and papers. You can also establish a dedicated page to collect your lab Protocols. Further information can be found at www.nature.com/protocolexchange/about

With kind regards,

Melina Casadio, PhD
Senior Editor, Nature Cell Biology
ORCID ID: <https://orcid.org/0000-0003-2389-2243>

** Visit the Springer Nature Editorial and Publishing website at www.springernature.com/editorial-and-publishing-jobs for more information about our career opportunities. If you have any questions please click here.**